# One Fits All:
# Power General Time Series Analysis by Pretrained LM

**Tian Zhou**[*]  **Peisong Niu**[*]  **Xue Wang**[*]  **Liang Sun**  **Rong Jin**[†]
{tian.zt,niupeisong.nps,xue.w,liang.sun,jinrong.jr}@alibaba-inc.com

## Abstract

Although we have witnessed great success of pre-trained models in natural language processing (NLP) and computer vision (CV), limited progress has been made for general time series analysis. Unlike NLP and CV where a unified model can be used to perform different tasks, specially designed approach still dominates in each time series analysis task such as classification, anomaly detection, forecasting, and few-shot learning. The main challenge that blocks the development of pre-trained model for time series analysis is the lack of a large amount of data for training. In this work, we address this challenge by leveraging language or CV models, pre-trained from billions of tokens, for time series analysis. Specifically, we refrain from altering the self-attention and feedforward layers of the residual blocks in the pre-trained language or image model. This model, known as the Frozen Pretrained Transformer (FPT), is evaluated through fine-tuning on all major types of tasks involving time series. Our results demonstrate that pre-trained models on natural language or images can lead to a comparable or state-of-the-art performance in all main time series analysis tasks, as illustrated in Figure 1. We also found both theoretically and empirically that the self-attention module behaviors similarly to principle component analysis (PCA), an observation that helps explains how transformer bridges the domain gap and a crucial step towards understanding the universality of a pre-trained transformer. The code is publicly available at `https://github.com/DAMO-DI-ML/One_Fits_All`.

## 1 Introduction

Time series analysis is a fundamental problem Hyndman & Athanasopoulos (2021) that has played an important role in many real-world applications Wen et al. (2022), such as retail sales forecasting Böse et al. (2017); Courty & Li (1999) , imputation of missing data for economic time series  Friedman (1962) anomaly detection for industrial maintenance Gao et al. (2020), and classification of time series from various domain  Ismail Fawaz et al. (2019). Numerous statistical and machine learning methods have been developed for time series analysis in the past. Inspired by its great success in natural language processing and computer vision Vaswani et al. (2017); Devlin et al. (2019); Dosovitskiy et al. (2021); Rao et al. (2021), transformer has been introduced to various time series tasks with promising results Wen et al. (2023), especially for time series forecasting Lim et al. (2021); Zhou et al. (2022, 2021); Wu et al. (2021); Nie et al. (2022).

We have recently witnessed the rapid development of foundation models in NLP. The key idea is to pre-train a large language model from billions of tokens to facilitate model training for downstream tasks, particularly when we have a few, sometimes even zero, labeled instances. Another advantage of

---

∗ Equal contribution
† Corresponding authors

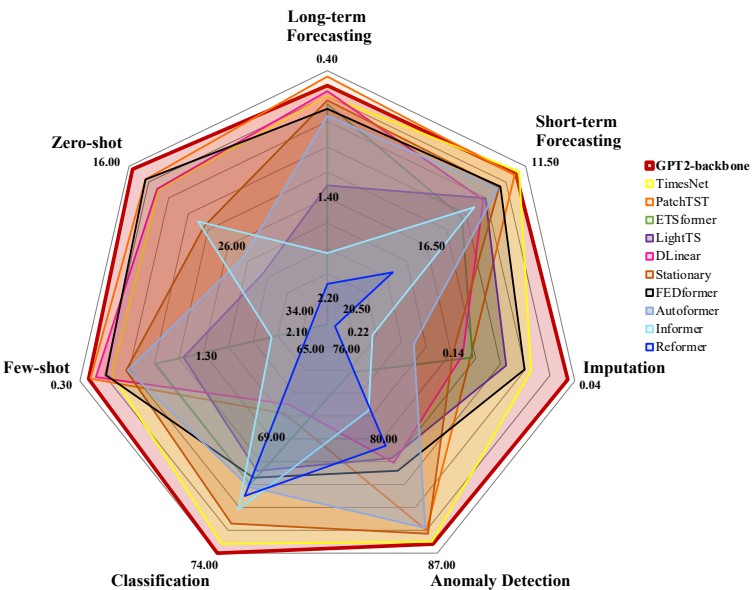

Figure 1: Model performance comparison on various tasks.

foundation models is that they provide a unified framework for handling diverse tasks, which contrasts conventional wisdom where each task requires a specially designed algorithm. However, so far, little progress has been made to exploit pre-trained or foundation models for time series analysis. One main challenge is the lack of the large amount of data to train a foundation model for time series analysis. The largest data sets for time series analysis is less than 10GB Godahewa et al. (2021), which is much smaller than that for NLP. To address this challenge, we propose to leverage pre-trained language models for general time series analysis. Our approach provides a **unified framework** for diverse time series tasks, such as classification, anomaly detection, forecasting, and few-shot or zero-shot learning. As shown in Figure 1, using the same backbone, our approach performs either on-par or better than the state-of-the-art methods for all main time series analysis tasks. Besides extensive empirical studies, we also investigate why a transformer model pre-trained from the language domain can be adapted to time series analysis with almost no change. Our analysis indicates that the self-attention modules in the pre-trained transformer acquire the ability to perform certain non-data-dependent operations through training. These operations are closely linked to principal component analysis over the input patterns. We believe it is this generic function performed by the self-attention module that allows trained transformer models to be so-called universal compute engine Lu et al. (2022) or general computation calculator Giannou et al. (2023). We support our claims by conducting an empirical investigation of the resemblance in model behaviors when self-attention is substituted with PCA, and by providing a theoretical analysis of their correlation.

Here we summarize our key contributions as follows:

1. We propose a unified framework that uses a frozen pre-trained language model to achieve a SOTA or comparable performance in all major types of time series analysis tasks supported by thorough and extensive experiments, including time series classification, short/long-term forecasting, imputation, anomaly detection, few-shot and zero-sample forecasting.

2. We found, both theoretically and empirically, that self-attention performs a function similar to PCA, which helps explain the universality of transformer models.

3. We demonstrate the universality of our approach by exploring a pre-trained transformer model from another backbond model (BERT) or modality (computer vision) to power the time series forecasting.

The remainder of this paper is structured as follows. Section 2 briefly summarizes the related work. Section 3 presents the proposed detailed model structure. In Section 4, we conduct a thorough and extensive evaluation of the performance of cross-modality time series analysis using our proposed method in seven main time series analysis tasks compared to various SOTA baseline models. Section 5

presents various ablation studies, and Section 6 demonstrates the universality of our proposed method using pre-trained models with another structure or pre-trained from another modality. In Section 7, we provide a theoretical explanation of the connection between self-attention and PCA. Finally, in Section 8, we discuss our results and future directions. Due to space limit, more extensive discussion of related work, experimental results, and theoretical analysis are provided in the Appendix.

## 2    Related Work

In this section, we provide short reviews of literature in the areas of time series analysis, in-modality transfer learning, and cross-modality knowledge transfer learning. We postpone the discussion of works for end-to-end time series analysis to Appendix B, due to the limited space.

**In-modality Transfer Learning through pre-trained models**    In recent years, a large number of research works have verified the effectiveness of the pre-trained model from NLP, CV to Vision-and-Language (VL). Latest studies for NLP focus on learning contextual word embeddings for downstream tasks. With the increase of computing power, the very deep transformer models have shown powerful representation ability in various language tasks. Among them, BERT Devlin et al. (2019) uses transformer encoders and employs masked language modeling task that aims to recover the random masked tokens within a text. OpenAI proposed GPT Radford & Narasimhan (2018) that trains transformer decoders on a large language corpus and then fine-tunes on task-specific data. GPT2 Radford et al. (2019) is trained on larger datasets with much more parameters and can be transferred to various downstream tasks. Since transformer models can adapt to various inputs, the idea of pre-training can also be well adapted to visual tasks. DEiT Touvron et al. (2021) proposed a teacher-student strategy for transformers with convolution neural networks (CNNs) as the teacher model and achieves competitive performance. BEiT Bao et al. (2022) converts images as visual tokens and successfully uses the BERT model in CV. However, because of the **insufficient training sample**, there is little research on pre-trained models on general time series analysis that cover all major tasks like CV or NLP domain.

**Cross-modality knowledge transfer**    Since transformers can handle different modal tasks through tokenizing the inputs to embeddings, it is also an interesting topic whether the transformers have universal representation ability and can be used for transferring between various domains. The VL pre-trained model VLMo Bao et al. (2021) proposed a stagewise pre-training strategy that utilizes frozen attention blocks pre-trained by image-only data to train the language expert. One of the most related works which transfer knowledge from a pre-trained language model to other domains is Lu et al. (2022), which studies the strong performance of a frozen pre-trained language model (LM) compared to an end-to-end transformer alternative learned from other domains' data. Another relative work for knowledge transfer to the time series is the Voice2series Yang et al. (2021), which leverages a pre-trained speech processing model for time series classification and achieves superior performance. To the best of our knowledge, no previous research has investigated cross-modality knowledge transfer for the time series forecasting task, let alone general time series analysis.

## 3    Methodology

### 3.1    Model Structure

The architecture we employ is depicted in Figure 2. We utilize parameters from NLP pretrained transformer models for time series analysis, with a focus on the GPT2 model Radford et al. (2019). We also experiment with other models, such as BERT Devlin et al. (2019) and BEiT Bao et al. (2022), to further demonstrate that the universal performance of cross-domain knowledge transfer exists in a wide range of pre-trained models.

**Frozen Pretrained Block** Our architecture retains the positional embedding layers and self-attention blocks from the pre-trained models. As self-attention layers and FFN (Feedforward Neural Networks) contain the majority of learned knowledge from pre-trained language models, we opt to freeze the self-attention blocks while fine-tuning.

**Positional Embeddings and Layer Normalization** To enhance downstream tasks with minimal effort, we fine-tune the positional embeddings and layer normalization layer, which is considered a

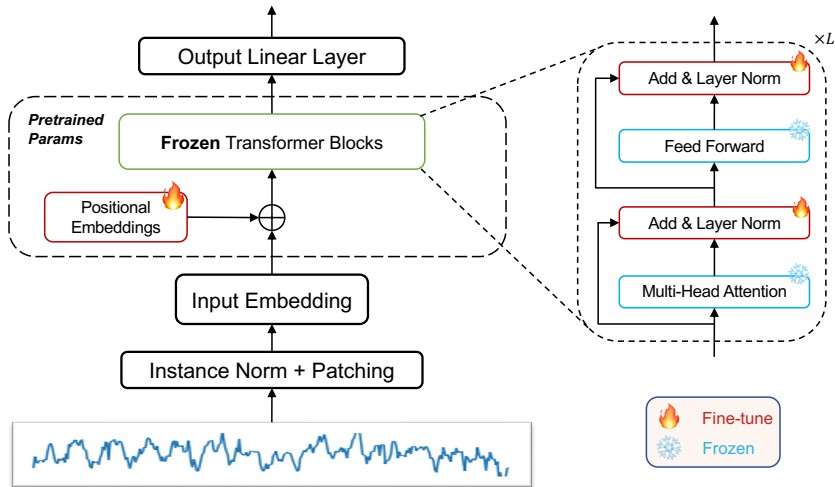

Figure 2: Model architecture. Pre-trained parameters are transferred to the time series forecasting tasks. Self-attention and Feedforward layers in the transformer blocks are frozen while only the embedding layer, normalization layers, and output layer require training.

standard practiceLu et al. (2022); Houlsby et al. (2019). As a result, we retrain these components during fine-tuning.

**Input Embedding** Given our goal of applying the NLP pre-trained model to various tasks and a new modality, we must redesign and train the input embedding layer. This layer is responsible for projecting the time-series data to the required dimensions of the specific pre-trained model. To accomplish this, we use linear probing, which also reduces the number of parameters required for training.

**Normalization** Data normalization is crucial for pre-trained models across various modalities. In addition to the layer norm utilized in pre-trained LM, we also incorporate a simple data normalization block, reverse instance norm Kim et al. (2022), to further facilitate knowledge transfer. This normalization block simply normalizes the input time series using mean and variance, and then adds them back to the output.

**Patching** To extract local semantic information, we utilize patching Nie et al. (2022) by aggregating adjacent time steps to form a single patch-based token. Patching enables a significant increase in the input historical time horizon while maintaining the same token length and reducing information redundancy for transformer models. In our architecture, we apply patching after instance normalization.

## 4 Main Time Series Analysis Tasks

Our proposed method excels in various downstream time series analysis tasks through fine-tuning. To demonstrate the effectiveness of our approach, we conduct extensive experiments on major types of downstream tasks, including time series classification, anomaly detection, imputation, short/long-term forecasting and few-shot/zero-shot forecasting. To ensure a fair comparison, we use GPT2-backbone FPT and adhere to the experimental settings of TimesNet Wu et al. (2023). Due to the space limit, only the summarized results are presented below except zero-shot forecasting. Full experimental results of the other six downstream tasks can be found in Appendix D.3, D.2, D.7, H.6, H.7, H.8, H.9 respectively.

**Baselines** We select representative baselines and cite their results from Wu et al. (2023), which includes the most recent and quite extensive empirical studies of time series. The baselines include CNN-based models: TimesNet Wu et al. (2023); MLP-based models: LightTS Zhang et al. (2022) and DLinear Zeng et al. (2023); Transformer-based models: Reformer Kitaev et al. (2020), Informer Zhou et al. (2021), Autoformer Wu et al. (2021), FEDformer Zhou et al. (2022), Non-stationary Transformer Liu et al. (2022), ETSformer Woo et al. (2022), PatchTST Nie et al. (2022). Besides, N-HiTS Challu et al. (2022) and N-BEATS Oreshkin et al. (2019) are used for short-term forecasting. Anomaly Transformer Xu et al. (2021) is used for anomaly detection. XGBoost Chen & Guestrin (2016), Rocket Dempster et al. (2020), LSTNet Lai et al. (2018), LSSL Gu et al. (2021),

Pyraformer Liu et al. (2021), TCN Franceschi et al. (2019) and Flowformer Huang et al. (2022) are used for classification.

## 4.1 Main Results

Overall, as shown in Figure 1, GPT2-backbone FPT outperforms other models in most tasks, including long/short-term forecasting, classification, anomaly detection, imputation, and fow-shot/zero-short forecasting. This confirms that time series tasks can also take advantage of cross-modality transferred knowledge. In the following, we use GPT2(K) to represent GPT2-backbone with first K Layers.

## 4.2 Imputation

**Setups** We conduct experiments on six popular real-world datasets, including 4 ETT datasets Zhou et al. (2021) (ETTh1, ETTh2, ETTm1, ETTm2), Electricity and Weather, where the data-missing is common. Following the settings of TimesNet, different random mask ratios ({12.5%, 25%, 37.5%, 50%}) of time points are selected for the evaluation on various proportions of missing data.

**Results** The results are shown in Table 1 that GPT2(3) FPT achieves the best performance on most datasets. Particularly, compared to the previous SOTA TimesNet, GPT2(3) FPT yields a relative **11.5%** MSE reduction on ETTh1,and a **4.1%** MSE reduction on average on six benchmark datasets. It verifies that the proposed method can also effectively mine temporal patterns of incomplete time series.

Table 1: Imputation task. We randomly mask {12.5%, 25%, 37.5%, 50%} time points of 96-length time series. The results are averaged from 4 different mask ratios. **Black**: best, Red: second best. Appendix H.8 shows the full results.

| Methods | GPT2(3) MSE | GPT2(3) MAE | TimesNet MSE | TimesNet MAE | PatchTST MSE | PatchTST MAE | ETSformer MSE | ETSformer MAE | LightTS MSE | LightTS MAE | DLinear MSE | DLinear MAE | FEDformer MSE | FEDformer MAE | Stationary MSE | Stationary MAE | Autoformer MSE | Autoformer MAE | Informer MSE | Informer MAE | Reformer MSE | Reformer MAE |
|---|---|---|---|---|---|---|---|---|---|---|---|---|---|---|---|---|---|---|---|---|---|---|
| ETTm1 | 0.028 | 0.105 | 0.027 | 0.107 | 0.047 | 0.140 | 0.120 | 0.253 | 0.104 | 0.218 | 0.093 | 0.206 | 0.062 | 0.177 | 0.036 | 0.126 | 0.051 | 0.150 | 0.071 | 0.188 | 0.055 | 0.166 |
| ETTm2 | 0.021 | 0.084 | 0.022 | 0.088 | 0.029 | 0.102 | 0.208 | 0.327 | 0.046 | 0.151 | 0.096 | 0.208 | 0.101 | 0.215 | 0.026 | 0.099 | 0.029 | 0.105 | 0.156 | 0.292 | 0.157 | 0.280 |
| ETTh1 | 0.069 | 0.173 | 0.078 | 0.187 | 0.115 | 0.224 | 0.202 | 0.329 | 0.284 | 0.373 | 0.201 | 0.306 | 0.117 | 0.246 | 0.094 | 0.201 | 0.103 | 0.214 | 0.161 | 0.279 | 0.122 | 0.245 |
| ETTh2 | 0.048 | 0.141 | 0.049 | 0.146 | 0.065 | 0.163 | 0.367 | 0.436 | 0.119 | 0.250 | 0.142 | 0.259 | 0.163 | 0.279 | 0.053 | 0.152 | 0.055 | 0.156 | 0.337 | 0.452 | 0.234 | 0.352 |
| ECL | 0.090 | 0.207 | 0.092 | 0.210 | 0.072 | 0.183 | 0.214 | 0.339 | 0.131 | 0.262 | 0.132 | 0.260 | 0.130 | 0.259 | 0.100 | 0.218 | 0.101 | 0.225 | 0.222 | 0.328 | 0.200 | 0.313 |
| Weather | 0.031 | 0.056 | 0.030 | 0.054 | 0.034 | 0.055 | 0.076 | 0.171 | 0.055 | 0.117 | 0.052 | 0.110 | 0.099 | 0.203 | 0.032 | 0.059 | 0.031 | 0.057 | 0.045 | 0.104 | 0.038 | 0.087 |
| Average | 0.047 | 0.127 | 0.049 | 0.132 | 0.060 | 0.144 | 0.197 | 0.309 | 0.123 | 0.228 | 0.119 | 0.224 | 0.112 | 0.229 | 0.056 | 0.142 | 0.061 | 0.151 | 0.165 | 0.273 | 0.134 | 0.240 |

## 4.3 Time Series Classification

**Setups** To evaluate the model's capacity for high-level representation learning, we employ sequence-level classification. Specifically, we follow the same setting as TimesNet: For classification, 10 multivariate UEA classification datasets Bagnall et al. (2018) are selected for evaluation, including gesture, action, audio recognition medical diagnosis and other practical tasks.

**Results** As shown in Figure 3, GPT2(6) FPT achieves an average accuracy of 74.00%, surpassing all baselines including TimesNet (73.60%). Specifically, compared to recent published patch-transformer-based models Nie et al. (2022) , GPT2(6) FPT surpasses it by a large margin **9.0%** which shows the prior NLP transfer knowledge can indeed help in time series representation.

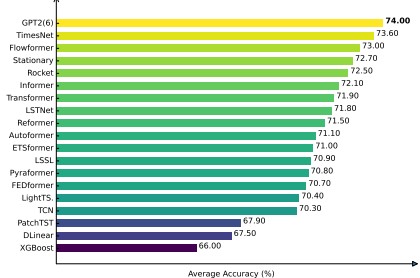

Figure 3: Model comparison in classification. The results are averaged from 10 subsets of UEA. Appendix H.6 shows the full results.

## 4.4 Time Series Anomaly Detection

**Setups** Detecting anomalies in time series is vital in industrial applications, ranging from health monitoring to space & earth exploration. We compare models on five commonly used datasets, including

---

https://archive.ics.uci.edu/ml/datasets/ElectricityLoadDiagrams 20112014
https://www.bgc-jena.mpg.de/wetter/

SMDSu et al. (2019), MSLHundman et al. (2018), SMAPHundman et al. (2018), SWaTMathur & Tippenhauer (2016) and PSMAbdulaal et al. (2021). To perform a fair comparison, only the classical reconstruction error is used for all baseline models to the make the setting the same as TimesNet.

**Results** Table 2 demonstrates that GPT2(6) FPT also achieves the best performance with the averaged F1-score **86.72%**, surpassing previous SOTA method TimesNet by **1.7%**. Thus, in addition to its proficiency in representing complete sequences for classification purposes, GPT2(6) FPT is capable of detecting infrequent anomalies within time series.

Table 2: Anomaly detection task. We calculate the F1-score (as %) for each dataset. ∗. in the Transformers indicates the name of ∗former. **Black**: best, Red: second best. Appendix H.7 shows the full results.

| Methods | GPT2(6) Ours | TimesNet* | PatchTS. | ETS. | FED. | LightTS | DLinear | Stationary | Auto. | Pyra. | Anomaly.** | In. | Re. | LogTrans. | Trans. |
|---|---|---|---|---|---|---|---|---|---|---|---|---|---|---|---|
| SMD | 86.89 | 84.61 | 84.62 | 83.13 | 85.08 | 82.53 | 77.10 | 84.72 | 85.11 | 83.04 | 85.49 | 81.65 | 75.32 | 76.21 | 79.56 |
| MSL | 82.45 | 81.84 | 78.70 | 85.03 | 78.57 | 78.95 | 84.88 | 77.50 | 79.05 | 84.86 | 83.31 | 84.06 | 84.40 | 79.57 | 78.68 |
| SMAP | 72.88 | 69.39 | 68.82 | 69.50 | 70.76 | 69.21 | 69.26 | 71.09 | 71.12 | 71.09 | 71.18 | 69.92 | 70.40 | 69.97 | 69.70 |
| SWaT | 94.23 | 93.02 | 85.72 | 84.91 | 93.19 | 93.33 | 87.52 | 79.88 | 92.74 | 91.78 | 83.10 | 81.43 | 82.80 | 80.52 | 80.37 |
| PSM | 97.13 | 97.34 | 96.08 | 91.76 | 97.23 | 97.15 | 93.55 | 97.29 | 93.29 | 82.08 | 79.40 | 77.10 | 73.61 | 76.74 | 76.07 |
| Average | **86.72** | 85.24 | 82.79 | 82.87 | 84.97 | 84.23 | 82.46 | 82.08 | 84.26 | 82.57 | 80.50 | 78.83 | 77.31 | 76.60 | 76.88 |

* We reproduce the results of TimesNet by https://github.com/thuml/Time-Series-Library.
** We replace the joint criterion in Anomaly Transformer with reconstruction error for fair comparison.

## 4.5 Long-term Forecasting

**Setups** Eight popular real-world benchmark datasets Wu et al. (2023), including Weather, Traffic , Electricity, ILI , and 4 ETT datasets (ETTh1, ETTh2, ETTm1, ETTm2), are used for long-term forecasting evaluation. Additional information regarding the discussion on the input length setting can be found in the appendix H.10.

**Results** As shown in Table 3, GPT2(6) FPT achieves comparable performance with PatchTST and outperforms other baselines. Specifically, compared with recent published SOTA method TimesNet, GPT2(6) FPT yields a relative **9.3%** average MSE reduction.

Table 3: Long-term forecasting task. All the results are averaged from 4 different prediction lengths, that is {24, 36, 48, 60} for ILI and {96, 192, 336, 720} for the others. **Black**: best, Red: second best. Appendix D.3 shows the full results.

| Methods | GPT2(6) MSE | MAE | TimesNet MSE | MAE | ETSformer MSE | MAE | LightTS MSE | MAE | DLinear MSE | MAE | FEDformer MSE | MAE | PatchTST MSE | MAE | Stationary MSE | MAE | Autoformer MSE | MAE | Informer MSE | MAE | Reformer MSE | MAE |
|---|---|---|---|---|---|---|---|---|---|---|---|---|---|---|---|---|---|---|---|---|---|---|
| Weather | 0.237 | 0.270 | 0.259 | 0.287 | 0.271 | 0.334 | 0.261 | 0.312 | 0.249 | 0.300 | 0.309 | 0.360 | **0.225** | **0.264** | 0.288 | 0.314 | 0.338 | 0.382 | 0.634 | 0.548 | 0.803 | 0.656 |
| ETTh1 | 0.427 | 0.426 | 0.458 | 0.450 | 0.542 | 0.510 | 0.491 | 0.479 | 0.423 | 0.437 | 0.440 | 0.460 | **0.413** | 0.430 | 0.570 | 0.537 | 0.496 | 0.487 | 1.040 | 0.795 | 1.029 | 0.915 |
| ETTh2 | 0.346 | 0.394 | 0.414 | 0.427 | 0.439 | 0.452 | 0.602 | 0.543 | 0.431 | 0.447 | 0.437 | 0.449 | **0.330** | **0.379** | 0.526 | 0.516 | 0.450 | 0.459 | 4.431 | 1.729 | 6.736 | 2.191 |
| ETTm1 | 0.352 | 0.383 | 0.400 | 0.406 | 0.429 | 0.425 | 0.435 | 0.437 | 0.357 | **0.378** | 0.448 | 0.452 | **0.351** | 0.387 | 0.481 | 0.456 | 0.588 | 0.517 | 0.961 | 0.734 | 0.799 | 0.671 |
| ETTm2 | 0.266 | 0.326 | 0.291 | 0.333 | 0.293 | 0.342 | 0.409 | 0.436 | 0.267 | 0.334 | 0.305 | 0.349 | **0.255** | **0.315** | 0.306 | 0.347 | 0.327 | 0.371 | 1.410 | 0.810 | 1.479 | 0.915 |
| ILI | 1.925 | 0.903 | 2.139 | 0.931 | 2.497 | 1.004 | 7.382 | 2.003 | 2.169 | 1.041 | 2.847 | 1.144 | **1.443** | **0.798** | 2.077 | 0.914 | 3.006 | 1.161 | 5.137 | 1.544 | 4.724 | 1.445 |
| ECL | 0.167 | 0.263 | 0.192 | 0.295 | 0.208 | 0.323 | 0.229 | 0.329 | 0.166 | 0.263 | 0.214 | 0.327 | **0.161** | **0.253** | 0.193 | 0.296 | 0.227 | 0.338 | 0.311 | 0.397 | 0.338 | 0.422 |
| Traffic | 0.414 | 0.294 | 0.620 | 0.336 | 0.621 | 0.396 | 0.622 | 0.392 | 0.434 | 0.295 | 0.610 | 0.376 | **0.390** | **0.264** | 0.624 | 0.340 | 0.628 | 0.379 | 0.764 | 0.416 | 0.741 | 0.422 |
| Average | 0.516 | 0.407 | 0.596 | 0.433 | 0.662 | 0.473 | 1.303 | 0.616 | 0.562 | 0.436 | 0.701 | 0.489 | **0.446** | **0.386** | 0.633 | 0.465 | 0.757 | 0.511 | 1.836 | 0.871 | 2.081 | 0.954 |

## 4.6 Short-term Forecasting

**Setups** To fully evaluate different algorithms in forecasting tasks, we also conduct short-term forecasting (with relatively short forecasting horizon) experiments on M4 Makridakis et al. (2018), contains marketing data of various frequencies.

**Results** The results in Table 4 show that the performance of GPT2-backbone (6) FPT is superior to advanced Transformer-based and MLP-based models, and comparable to TimesNet and N-BEATS.

## 4.7 Few-shot Forecasting

The large language model (LLM) has demonstrated remarkable performance in both few-shot and zero-shot learning settings Brown et al. (2020); OpenAI (2023). It can be argued that few-shot and

http://pems.dot.ca.gov
https://gis.cdc.gov/grasp/fluview/fluportaldashboard.html

Table 4: Short-term forecasting task on M4. The prediction lengths are in [6, 48] and results are weighted averaged from several datasets under different sample intervals. **Black**: best, **Red**: second best. Appendix H.9 shows the full results.

| Methods | GPT2(6) | TimesNet | PatchTST | N-HiTS | N-BEATS | ETSformer | LightTS | DLinear | FEDformer | Stationary | Autoformer | Informer | Reformer |
|---|---|---|---|---|---|---|---|---|---|---|---|---|---|
| SMAPE | 11.991 | **11.829** | 12.059 | 11.927 | **11.851** | 14.718 | 13.525 | 13.639 | 12.840 | 12.780 | 12.909 | 14.086 | 18.200 |
| MASE | 1.600 | **1.585** | 1.623 | 1.613 | **1.599** | 2.408 | 2.111 | 2.095 | 1.701 | 1.756 | 1.771 | 2.718 | 4.223 |
| OWA | 0.861 | **0.851** | 0.869 | 0.861 | **0.855** | 1.172 | 1.051 | 1.051 | 0.918 | 0.930 | 0.939 | 1.230 | 1.775 |

zero-shot learning also represent the ultimate tasks for a universal time series forecasting model. To extensively evaluate the representation power of the GPT2(6) for time series analysis, we conduct experiments under few-shot and zero-shot learning settings.

Similar to traditional experimental settings, each time series is split into three parts: training data, validation data, and test data. For few-shot learning, only a certain percentage (10%, 5%) timesteps of training data are used.

The results of 10% few-shot learning are shown in Table 5. Compared to TimesNet, DLinear, PatchTST and other methods, GPT2(6) FPT achieves the best performance. Traditionally, CNN-based and single MLP-based models are considered more data-efficient for training and suitable for few-shot learning methods. In comparison to convolution-based TimesNet and MLP-based DLinear models, GPT2(6) FPT demonstrates a relative average MSE reduction of **33.3%** and **13.5%** respectively. We add a comparison with traditional algorithms (ETS, ARIMA, NaiveDrift) in the Appendix D.5 as well, and GTP2(6)FPT also surpass all those traditional methods.

Table 5: Few-shot learning task on 10% data. All the results are averaged from 4 different prediction lengths ({96, 192, 336, 720}). **Black**: best, **Red**: second best. Appendix D.2 shows the detailed results of 10% and 5% data.

| Methods | GPT2(6) | | TimesNet | | DLinear | | FEDformer | | PatchTST | | Autoformer | | Stationary | | ETSformer | | LightTS | | Informer | | Reformer | |
|---|---|---|---|---|---|---|---|---|---|---|---|---|---|---|---|---|---|---|---|---|---|---|
| | MSE | MAE | MSE | MAE | MSE | MAE | MSE | MAE | MSE | MAE | MSE | MAE | MSE | MAE | MSE | MAE | MSE | MAE | MSE | MAE | MSE | MAE |
| Weather | 0.238 | 0.275 | 0.279 | 0.301 | 0.301 | 0.283 | 0.284 | 0.324 | 0.241 | 0.279 | 0.300 | 0.342 | 0.318 | 0.322 | 0.317 | 0.359 | 0.289 | 0.322 | 0.597 | 0.494 | 0.545 | 0.469 |
| ETTh1 | 0.590 | 0.524 | 0.869 | 0.628 | 0.691 | 0.599 | 0.638 | 0.561 | 0.633 | 0.542 | 0.701 | 0.596 | 0.914 | 0.639 | 1.179 | 0.833 | 1.375 | 0.877 | 1.199 | 0.808 | 1.249 | 0.833 |
| ETTh2 | 0.397 | 0.421 | 0.479 | 0.465 | 0.608 | 0.538 | 0.466 | 0.475 | 0.415 | 0.431 | 0.488 | 0.499 | 0.461 | 0.454 | 0.893 | 0.713 | 2.655 | 1.159 | 3.871 | 1.512 | 3.485 | 1.485 |
| ETTm1 | 0.464 | 0.441 | 0.676 | 0.537 | 0.411 | 0.429 | 0.721 | 0.605 | 0.501 | 0.466 | 0.802 | 0.628 | 0.797 | 0.577 | 0.979 | 0.714 | 0.970 | 0.704 | 1.192 | 0.820 | 1.425 | 0.856 |
| ETTm2 | 0.293 | 0.335 | 0.319 | 0.353 | 0.316 | 0.368 | 0.463 | 0.488 | 0.296 | 0.343 | 1.341 | 0.930 | 0.332 | 0.366 | 0.447 | 0.487 | 0.987 | 0.755 | 3.369 | 1.439 | 3.977 | 1.586 |
| ECL | 0.176 | 0.269 | 0.323 | 0.392 | 0.180 | 0.280 | 0.346 | 0.428 | 0.180 | 0.269 | 0.431 | 0.478 | 0.443 | 0.479 | 0.659 | 0.617 | 0.441 | 0.488 | 1.194 | 0.890 | 0.965 | 0.768 |
| Traffic | 0.440 | 0.309 | 0.951 | 0.535 | 0.496 | 0.371 | 0.663 | 0.425 | 0.430 | 0.305 | 0.749 | 0.446 | 1.453 | 0.815 | 1.913 | 0.936 | 1.247 | 0.684 | 1.534 | 0.811 | 1.550 | 0.821 |
| Average | **0.371** | **0.367** | 0.556 | 0.458 | 0.429 | 0.409 | 0.511 | 0.472 | **0.385** | **0.376** | 0.687 | 0.559 | 0.674 | 0.522 | 0.912 | 0.665 | 1.137 | 0.712 | 1.850 | 0.967 | 1.888 | 0.974 |

## 4.8 Zero-shot forecasting

This task is used to evaluate the cross datasets adaption ability of our proposed algorithm, i.e. how well a model is able to perform on dataset $A$ (without any training data from $A$) when it is trained from dataset $B$.

The results are summarized in Table 6. The GPT2(6) FPT model consistently outperforms all recent state-of-the-art transformer and MLP-based time series forecasting methods. Compared to recently published state-of-the-art MLP-based method Dlinear, convolution-based method Timesnet, and transformer-based method Patchtst, GPT2(6)FPT demonstrates a relative average metric reduction of **13.1%**, **13.6%** and **7.3%**, respectively. Also, the proposed method is comparable to N-BEATS without any meta-learning design and outperforms N-BEATS in the ELECTR dataset. We attribute this to the knowledge transfer capability from the FPT model.

## 5 Ablations

In this section, we conduct several ablations on model selection and effectiveness of pre-training. The detailed results are shown in Appendix H. We introduce several variants, GPT2(0) FPT, GPT2(6) without freezing and GPT2(6) without pre-training.

**Model Selection** We separately analyze the number of GPT2 layers and the fine-tuning parameters selection. The results in Appendix H show that GPT2 with 6-layers is a sound choice compared to full or few layers and partially freezing can avoid catastrophic forgetting, enabling fine-tuning without overfitting.

Table 6: Zero-shot learning results. Dataset-specific metrics aggregated over each dataset. A lower value indicates better performance. The source dataset of M3, Tourism, Electricity are M4. For M4, the source data for N-BEATS is FRED, and M3 for other models. **Black**: best, **Red**: second best, **Violet**: third best. Appendix D.7 shows full results.

| Methods
Metric | M4
sMAPE | M3
sMAPE | TOURISM
MAPE | ELECTR
$ND \times 100$ | Average |
|---|---|---|---|---|---|
| N-BEATS | **11.70** | **12.44** | **18.82** | 17.8 | **15.19** |
| DLinear | 15.33 | 14.03 | 28.51 | 17.6 | 18.86 |
| TimesNet | 13.55 | 14.17 | 28.84 | 19.3 | 18.96 |
| PatchTST | 13.22 | 13.06 | 27.10 | 17.3 | 17.67 |
| ETSformer | 27.74 | 16.03 | 180.40 | 44.2 | 67.09 |
| LightTS | 13.62 | 17.90 | 66.99 | 19.6 | 29.52 |
| Stationary | 13.32 | 15.29 | 43.75 | 22.0 | 23.59 |
| FEDformer | 15.04 | 13.53 | 31.55 | 18.4 | 19.63 |
| Autoformer | 20.02 | 15.87 | 40.39 | 33.9 | 27.54 |
| Informer | 19.04 | 15.82 | 35.82 | 21.2 | 22.97 |
| Reformer | 14.09 | 13.37 | 25.48 | 21.6 | 18.63 |
| GPT2(6) | 13.12 | 13.06 | 22.14 | 17.2 | 16.38 |

**Effectiveness of Pre-training** The results are shown in Table 7, GPT2(6) FPT outperforms both GPT2(0) FPT and GPT2-random-initialized, suggesting that GPT2 with pre-training parameters can achieve improvement on times series tasks. Besides, GPT2(6) FPT performs better than GPT2-unfrozen, demonstrating that partially freezing also helps. Also, results in Appendix H.2 show that random initialized GPT2(6) with freezing performs poorly and the pre-trained knowledge is instrumental for time series tasks.

Table 7: Ablation study on 10% data. All the results are averaged from 4 different prediction lengths. **No Freeze** represents GPT2(6) without freezing, **No Pretrain** represents GPT2(6) without pre-training. **Black**: best.

| Methods | GPT2(6) | | GPT2(0) | | No Freeze | | No Pretrain | |
|---|---|---|---|---|---|---|---|---|
| | MSE | MAE | MSE | MAE | MSE | MAE | MSE | MAE |
| Weather | **0.237** | **0.270** | 0.263 | 0.297 | 0.273 | 0.302 | 0.277 | 0.305 |
| ETTh1 | **0.427** | **0.426** | 0.874 | 0.647 | 0.753 | 0.596 | 1.326 | 0.743 |
| ETTh2 | **0.346** | **0.394** | 0.666 | 0.559 | 0.447 | 0.451 | 0.502 | 0.479 |

# 6 Exploring Transfer Learning from others: The Unexceptional Nature of GPT2-based-FPT

We also present experiments on BERT-backbond FPT Devlin et al. (2019) model and the image-pretrained BEiT-backbone FPT model Bao et al. (2022) to illustrate the generality of pre-trained models for cross-domain knowledge transferring. The results in Table 8 demonstrate that the ability of knowledge transfer is not exclusive to GPT2-based pre-trained language models. Subsequently, our theoretical analysis will shed light on the universality of this phenomenon.

Table 8: Results of frozen pretrained transformer variants on 5% ETTh2 and ETTm2. All the results are averaged from 4 different prediction lengths. **Black**: best. Appendix H.5 shows the full results.

| Methods | GPT2(6) | | BERT(6) | | BEiT(6) | | DLinear | | PatchTST | | FEDformer | | Autoformer | |
|---|---|---|---|---|---|---|---|---|---|---|---|---|---|---|
| | MSE | MAE | MSE | MAE | MSE | MAE | MSE | MAE | MSE | MAE | MSE | MAE | MSE | MAE |
| ETTh2 | **0.400** | **0.433** | 0.452 | 0.451 | 0.459 | 0.454 | 0.827 | 0.615 | 0.439 | 0.448 | 0.441 | 0.457 | 0.470 | 0.489 |
| ETTm2 | **0.308** | **0.346** | 0.318 | 0.357 | 0.315 | 0.357 | 0.399 | 0.426 | 0.314 | 0.352 | 0.381 | 0.404 | 0.388 | 0.433 |

# 7 Training/Inferencing Cost

Analysis of computational cost is helpful for investigating the practicality of the LLM-based model. The results can be found in table 9. Each baseline model comes in two variants, featuring model

Table 9: Training parameters and Training/Inference Cost Comparison

| Model | Training Params | Training Params Percentages | Training Time for 1 step(s) | Inference Time for 1 Batch(s) |
|---|---|---|---|---|
| FEDformer-32 | 44k | 100 | 0.889 | 0.170 |
| TimesNet-32 | 2M | 100 | 0.747 | 0.302 |
| PatchTST-32 | 543K | 100 | 0.043 | 0.022 |
| FEDformer-768 | 33M | 100 | 0.208 | 0.056 |
| TimesNet-768 | 42M | 100 | 5.723 | 2.162 |
| PatchTST-768 | 20M | 100 | 0.457 | 0.123 |
| GPT-2(3)-768 | 4M | 6.12 | 0.093 | 0.032 |
| GPT-2(6)-768 | 4M | 4.6 | 0.104 | 0.054 |

hidden dimensions of 32 and 768, which align with GPT-2's specifications. Furthermore, the majority of the baseline models consist of three layers. We assessed the computational cost using a batch from ETTh2 (with a batch size of 128) on a 32G V100 GPU.

The results indicate that GPT-2(3) has substantially enhanced time efficiency and reduced parameter quantity compared to baselines with the same model dimension. This was a surprise since we initially anticipated that this large language model might be slower. However, we surmise that the efficient optimization of huggingface's GPT model implementation primarily accounts for such a significant improvement in time costs. Furthermore, GPT-2(3) and GPT-2(6) demonstrate a mere 6.12% and 4.60% proportion of learnable parameters among the overall parameter size, respectively.

# 8 Towards Understanding the Universality of Transformer: Connecting Self-Attention with PCA

The observation, i.e. we can directly use a trained LM for time series forecasting without having to modify its model, makes us believe that the underlying model is doing something very generic and independent from texts despite it being trained from text data. Our analysis aims to show that part of this generic function can be related to PCA, as minimizing the gradient with respect to the self-attention layer seems to do something similar to PCA. In this section, we take the first step towards revealing the generality of self-attention by connecting the self-attention with principal component analysis (PCA). Moreover, when coming the question of why fine-tuning is restricted to the embedding layer and layer norm, following our hypothesis that the pre-trained LM as a whole performs something generic, partially fine-tuning any of its components may break the generic function and lead to relatively poor performance for time series analysis.

For each layer, we calculate and perform statistical analysis of the pairwise token similarity values. Specifically, we denote each output feature map with shape of $(b, n, d)$, where $b$ is the batch size, $n$ is the number of tokens, and $d$ is the dimension of each token feature. We calculate the cosine similarity, and the resulting pairwise similarity matrix of shape $(b, n, n)$. Next we count the number of occurrences of similarity values within each interval as a simple statistical analysis.

Our analysis is motivated by the observation that the within-layer token similarity increases with deeper layers in transformer. We report the layer-wise average token cosine similarity on ETTh2 dataset in Figure 4 (a, c), where we mix weights from pre-trained LM with weights randomly sampled from Gaussian distribution. Here we summarize our observations: a) in a randomly initialed GPT2 (6) model, the token similarity is low among all layers $(0.1 - 0.2)$; b) when gradually switched to the pretrained GPT2 model, the token similarity significantly increases in the deep layers and eventually reaches more than 0.9 in the last layer. One potential explanation for the increasing token similarity is that all the token vectors are projected into the low-dimensional top eigenvector space of input patterns. To verify this idea, we further conduct experiments where we replace the self-attention module with PCA and find token similarity patterns remain unchanged according to Figure 4 (b), which further justifies the potential connection between PCA and self-attention.

To build the theoretical connection between PCA and self-attention, we first analyze the gradient structure of self-attention. Let $X = (x_1, \ldots, x_N)^\top \in \mathbb{R}^{N \times D}$ be the input pattern, and let $f(X) = (f_1(X), \ldots, f_N(x))^\top : \mathbb{R}^{N \times D} \mapsto \mathbb{R}^{N \times D}$ be the function for self-attention, i.e.,
$$f_i(X) = \text{softmax}(XAX^\top)X \quad \text{where } A = W_Q W_K^\top \in \mathbb{R}^{D \times D}.$$

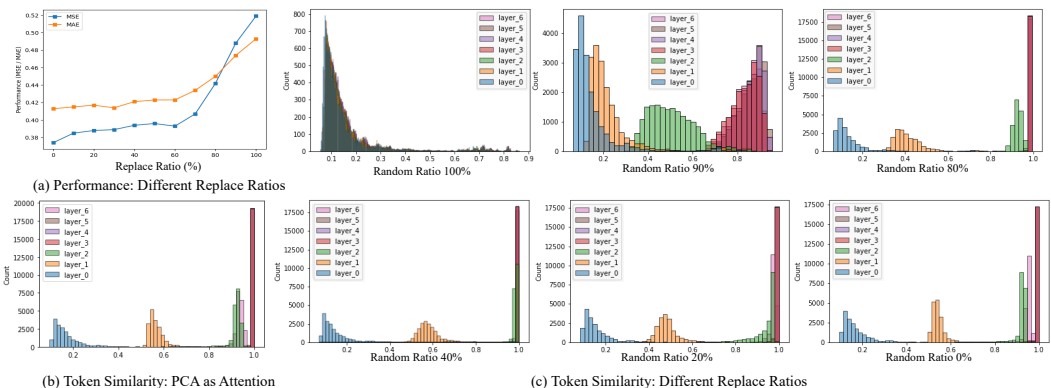

Figure 4: (a, c) The performance and token similarity within samples with respect to each layer with different random mixed ratio. Pre-trained parameters are mixed with random initial parameters according to certain proportions. (b) Token similarity within samples when replacing the attention with PCA.

*Lemma* 8.1. Let the Jacobian $J = \left[ \frac{\partial f_i(X)}{\partial x_j} \right]_{i,j=1}^N$ represent the gradient $f(X)$ w.r.t the input pattern, then we have $|J|_2 \leq |A|_2 \sum_{i=1}^N \left( P_{i,i} + \frac{1}{2} \right) \left| x_i - \sum_{j=1}^N P_{i,j} x_j \right|^2 + \Delta$ where $\Delta = |A|_2 \sum_{i \neq j}^N P_{i,j} \left| x_j - \sum_{k=1}^N P_{i,k} x_k \right|^2 + \frac{|A|_2}{2} \sum_{j=1}^N |x_i|^2$ and $P_{i,j} = \frac{\exp(x_i^\top A x_j)}{\sum_{k=1}^N \exp(x_i^\top A x_k)}$ .

This lemma reveals an important gradient structure of $J$. The proof of essentially follows the analysis in Kim et al. (2021), and we include it in Appendix G for completeness.

Using the gradient structure revealed in Lemma 8.1, we can connect self-attention with PCA. In order to minimize the norm of gradient $|J|_2$, we essentially need to make $\sum_{i=1}^N |x_i - \sum_{j=1}^N P_{i,j} x_j|^2$ small. When $A$ is small and all the input patterns are centered at 0 (i.e. $\sum_{i=1}^N x_i = 0$), we have $\sum_{i=1}^N |x_i - X^\top P_{i,:}|^2 \approx \sum_{i=1}^N |x_i - X^\top X A x_i|^2$.

The theorem below shows that $A$ minimizing the objective $\sum_{i=1}^N |x_i - X^\top X A x_i|^2$ contains the largest $m$ eigenvectors of $X^\top X$ where $m$ is the rank of $A$.

*Theorem* 1. Let $W_Q$ and $W_K$ be matrices of size $D \times m$. Let $\lambda_1 \geq \lambda_2 \geq ... \geq \lambda_D$ be the eigenvalues of $X^\top X$ ranked in descending order, and let $v_i \in \mathbb{R}^D, i = 1, \ldots, D$ be the corresponding eigenvectors. The optimal solution $A^*$ that minimizes $\sum_{i=1}^N |x_i - X^\top X A x_i|^2$ is given by $A = \sum_{i=1}^m \frac{1}{\lambda_i} v_i v_i^\top$.

The proof of Theorem 1 can be found in Appendix G. Following Theorem 1, through the training of pushing gradient to zero, self-attention learns to perform a function closely related to PCA.

## 9    Conclusions

In this paper, we developed a foundation model for time series analysis, based on pre-trained model from NLP or CV, that can (a) facilitate the model training for downstream tasks, and (b) provide unified framework for diverse time series analysis tasks. Our empirical studies show that the proposed method performs on par or better than the state-of-the-art approaches on almost all time series tasks. We also examine the universality of transformer by connecting self-attention with PCA, an important step towards understanding how generative models work in practice. On the other hand, we do recognize some limitations of our work: the zero-shot performance of our approach is still behind N-beat on several datasets, and our analysis of the generality of transformer is still in the early stage. Moving forward, we plan to improve the performance of our approach by exploiting the parameter efficient fine-tuning approaches which usually introduce additional structures into the pre-trained model for better adaption. To better understand the universality of transformer, we also plan to examine it from the viewpoint of n-gram language model, an approach that is taken by Elhage et al. (2021); Olsson et al. (2022). In Appendix F, we include our initial analysis along this direction.

## Acknowledgement

We would like to express our sincere gratitude to Ziqing Ma, Qingsong Wen, Mengni Ye, and Tao Yao for their valuable suggestions and proofreading assistance throughout the development of this paper. Their insightful feedback and attention to detail greatly improved the quality and clarity of our work.

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

# A    Visualization

In order to clarify the representation ability more clearly, Figure 5 provides showcases of imputation, long-term forecasting and few-shot forecasting. Especially for few-shot learning, GPT2(6) can accurately forecast, while TimesNet and DLinear fail in this task.

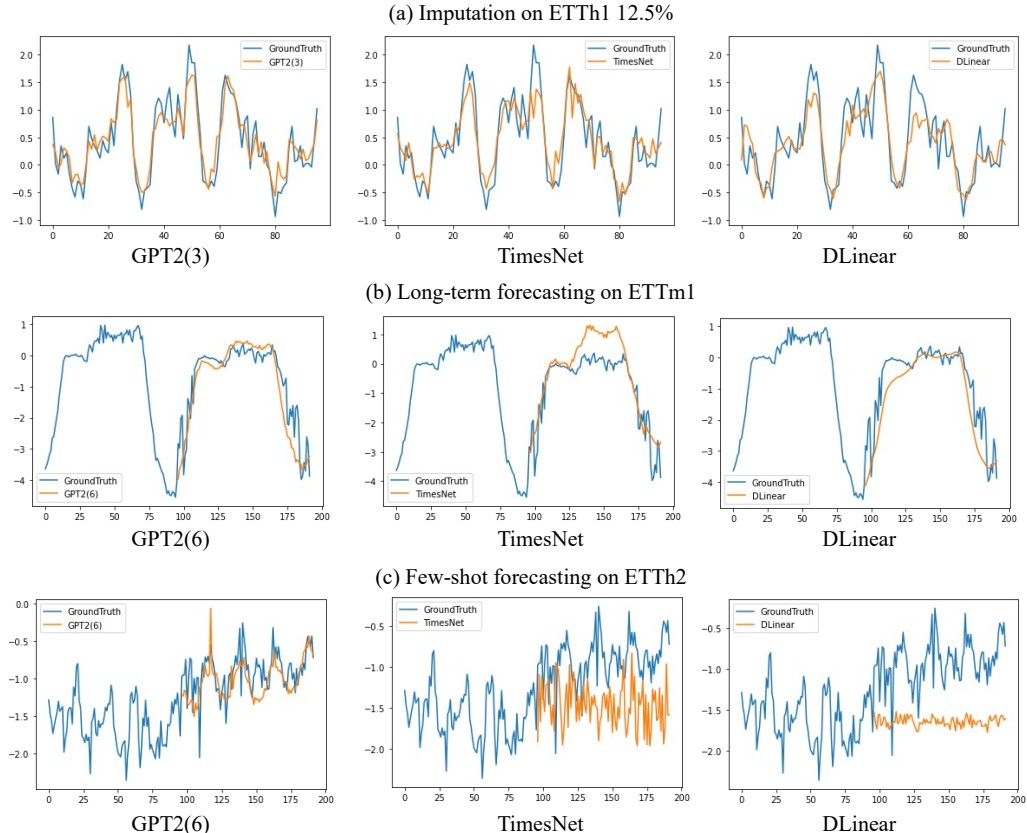

Figure 5: Visualization of imputation, long-term forecasting and few-shot forecasting.

# B    Related Works

We have presented a novel general time series analysis model in this paper, and to the best of our knowledge, there has been limited work on similar comprehensive methods for time series analysis. The most closely related field is time series forecasting, where transformer models have gained widespread popularity. Therefore, our focus in this related work will primarily be on introducing the end-to-end time series forecasting method.

Time series forecasting models can be roughly divided into three categories, ranging from the classic ARIMA models to the most recent transformer models. The first generation of well-discussed models can be dated back to auto-regressive family, such as ARIMA Box & Jenkins (1968); Box & Pierce (1970) that follows the Markov process and recursively execute sequential forecasting. However, it is limited to stationary sequences while most time series is non-stationary. Additionally, with the bloom of deep neural networks, recurrent neural networks (RNNs), such as LSTM Hochreiter & Schmidhuber (1997) and GRU Chung et al. (2014), were designed for sequential tasks. Yet the recurrent model is inefficient for training and long-term dependencies are still under resolved.

Recently, transformer models have achieve great progress in NLP Vaswani et al. (2017); Devlin et al. (2019); Radford et al. (2019) and CV Dosovitskiy et al. (2021); Bao et al. (2022) tasks. Also, a large amount of transformer models are proposed to apply to time series forecasting Wen et al. (2023). In the following, we briefly introduce several representative algorithms. Informer Zhou et al. (2021)

proposes a probability sparse attention mechanism to deal with long-term dependencies. Autoformer Wu et al. (2021) introduces a decomposition transformer architecture and replaces the attention module with an Auto-Correlation mechanism. FEDformer Zhou et al. (2022) uses Fourier enhanced structure to improve computational efficiency and achieves linear complexity. Similar to patching in ViT Dosovitskiy et al. (2021), PatchTST Nie et al. (2022) employs segmentation of time series that divide a sequence into patches to increase input length and reduce information redundancy. Besides, a simple MLP-based model DLinear Zeng et al. (2023) outperforms most transformer models and it validates channel-independence works well in time series forecasting. Recently, TimesNet Wu et al. (2023) has treated time series as a 2D signal and utilized a convolution-based inception net backbone to function as a comprehensive time series analysis model. This work is closely related to our tasks in this paper.

## C  Dataset Details

In this section, we separately summarize dataset details long/short-term forecasting and few-shot/zero-shot forecasting.

**Datasets of Long-term Forecasting and Few-shot Learning**  The details of datasets are shown as follows: 1) ETT datasets Zhou et al. (2021) contain electricity load of various resolutions (ETTh & ETTm) from two electricity stations. 2) Weather contains 21 meteorological indicators of Germany within 1 year; 3) Illness contains the influenza-like illness patients in the United States; 4) Electricity dataset contains the electricity consumption; 5) Traffic dataset contains the occupation rate of freeway system across the State of California. Table 10 summarizes details of feature statistics.

Similar to PatchTST Nie et al. (2022), Exchange is not contained. Zeng et al. (2023) shows that simply repeating the last value in the look-back window can outperform or be comparable to the best results. Also, ILI is not used for few-shot learning for the limited quantity that is hard to follow the definition of few-shot.

Table 10: Dataset details of few-shot learning.

| Dataset | Length | Dimension | Frequency |
|---|---|---|---|
| ETTh | 17420 | 7 | 1 hour |
| ETTm | 69680 | 7 | 15 min |
| Weather | 52696 | 22 | 10 min |
| ILI | 966 | 7 | 7 days |
| Electricity | 26304 | 321 | 1 hour |
| Traffic | 17544 | 862 | 1 hour |

**Datasets of Short-term Forecasting and Zero-shot Learning**  The details of short-term forecasting and zero-shot learning datasets are shown as follows: 1) M4 is a large and diverse dataset that contains time series of various frequencies and fields, including business, financial and economic forecasting; 2) M3 is smaller than M4, but also contains time series from diverse domains and frequencies; 3) TOURISM is the dataset of tourism activities with different frequencies and contains a much higher fraction of erratic series compared with M4; 4) ELECTR represents the electricity usage monitoring of 370 customers over three years. Table 6 summarizes details of the datasets and zero-shot mapping between source and target.

## D  Experimental Details

All the deep learning networks are implemented in PyTorch and trained on NVIDIA V100 32GB GPUs. We use the pre-trained models from Wolf et al. (2020) for experiments. For few-shot learning, an early stopping counter is employed to stop the training process after three epochs if no loss degradation on the valid set is observed. Plus, we convert the multivariate data into univariate data. Specifically, we treat each feature of the sequence as a single time series. This is mainly for memory efficiency after patching of GPT2(6) and previous works, DLinear and PatchTST, have proved the effectiveness of channel-independence.

Table 11: Datasets and mapping details of zero-shot learning.

| | Dataset | | Mapping | |
| | Length | Horizon | M4 | M3 |
|---|---|---|---|---|
| M3 Yearly | 645 | 6 | Yearly | - |
| M3 Quarterly | 756 | 8 | Quarterly | - |
| M3 Monthly | 1428 | 18 | Monthly | - |
| M3 Others | 174 | 8 | Monthly | - |
| M4 Yearly | 23000 | 6 | - | Yearly |
| M4 Quarterly | 24000 | 8 | - | Quarterly |
| M4 Monthly | 48000 | 18 | - | Monthly |
| M4 Weekly | 359 | 13 | - | Monthly |
| M4 Daily | 4227 | 14 | - | Monthly |
| M4 Hourly | 414 | 48 | - | Monthly |
| TOURISM Yearly | 518 | 4 | Yearly | Yearly |
| TOURISM Quarterly | 427 | 8 | Quarterly | Quarterly |
| TOURISM Monthly | 366 | 24 | Monthly | Monthly |
| ELECTR | 1311 | 168 | Hourly | Monthly |

## D.1 Accuracy Metrics

For long-term/short-term forecasting and few-shot forecasting, we use mean square error (MSE) and mean absolute error (MAE) as metrics. For zero-shot learning, mean absolute percentage error (MAPE) is used for TOURISM; symmetric MAPE (sMAPE) is used for M3 and M4; normalized deviation (ND) is used for ELECTR. All experiments are repeated 3 times and the mean of the metrics is used in the final results.

## D.2 Detailed Definition and Results for Few-shot and Long-term Forecasting

**Task Definition** Since Zeng et al. (2023) and Nie et al. (2022) have verified that channel-independence works well for time series datasets, we treat each multivariate series as multiple independent univariate series. Similar to traditional experimental settings, each time series is split into three parts: training data, validation data, and test data. For the few-shot forecasting task, only a certain percentage (5%, 10%) timesteps of training data are used, and the other two parts remain unchanged. The evaluation metrics remain the same as for classic multivariate time series forecasting. We repeat this experiment 3 times and report the average metrics in the following experiments.

**Detail Experiment Tables for Few-shot Time-Series Forecasting** in Table 12 and Table 13

Table 12: Few-shot learning results on 5% data. We use prediction length $O \in \{96, 192, 336, 720\}$. A lower MSE indicates better performance, and the best results are highlighted in bold. '-' means that 5% time series is not sufficient to constitute a training set.

| Methods | | GPT2(6) | | GPT2(0) | | DLinear | | PatchTST | | TimesNet | | FEDformer | | Autoformer | | Stationary | | ETSformer | | LightTS | | Informer | | Reformer | |
|---|---|---|---|---|---|---|---|---|---|---|---|---|---|---|---|---|---|---|---|---|---|---|---|---|---|
| Metric | | MSE | MAE | MSE | MAE | MSE | MAE | MSE | MAE | MSE | MAE | MSE | MAE | MSE | MAE | MSE | MAE | MSE | MAE | MSE | MAE | MSE | MAE | MSE | MAE |
| Weather | 96 | 0.175 | 0.230 | 0.191 | 0.243 | 0.184 | 0.242 | 0.171 | 0.224 | 0.207 | 0.253 | 0.229 | 0.309 | 0.227 | 0.299 | 0.215 | 0.252 | 0.218 | 0.295 | 0.230 | 0.285 | 0.497 | 0.497 | 0.406 | 0.435 |
| | 192 | 0.227 | 0.276 | 0.244 | 0.289 | 0.228 | 0.283 | 0.230 | 0.277 | 0.272 | 0.307 | 0.265 | 0.317 | 0.278 | 0.333 | 0.290 | 0.307 | 0.294 | 0.331 | 0.274 | 0.323 | 0.620 | 0.545 | 0.446 | 0.450 |
| | 336 | 0.286 | 0.322 | 0.303 | 0.332 | 0.279 | 0.322 | 0.294 | 0.326 | 0.313 | 0.328 | 0.353 | 0.392 | 0.351 | 0.393 | 0.353 | 0.348 | 0.359 | 0.398 | 0.318 | 0.355 | 0.649 | 0.547 | 0.465 | 0.459 |
| | 720 | 0.366 | 0.379 | 0.391 | 0.393 | 0.364 | 0.388 | 0.384 | 0.387 | 0.400 | 0.385 | 0.391 | 0.394 | 0.387 | 0.389 | 0.452 | 0.407 | 0.461 | 0.461 | 0.401 | 0.418 | 0.570 | 0.522 | 0.471 | 0.468 |
| | Avg. | **0.263** | **0.301** | 0.282 | 0.314 | **0.263** | 0.308 | 0.269 | 0.303 | 0.298 | 0.318 | 0.309 | 0.353 | 0.310 | 0.353 | 0.327 | 0.328 | 0.333 | 0.371 | 0.305 | 0.345 | 0.584 | 0.527 | 0.447 | 0.453 |
| ETTh1 | 96 | 0.543 | 0.506 | 0.825 | 0.638 | 0.547 | 0.503 | 0.557 | 0.519 | 0.892 | 0.625 | 0.593 | 0.529 | 0.681 | 0.570 | 0.952 | 0.650 | 1.169 | 0.832 | 1.483 | 0.91 | 1.225 | 0.812 | 1.198 | 0.795 |
| | 192 | 0.748 | 0.580 | 1.220 | 0.778 | 0.720 | 0.604 | 0.711 | 0.570 | 0.940 | 0.665 | 0.652 | 0.563 | 0.725 | 0.602 | 0.943 | 0.645 | 1.221 | 0.853 | 1.525 | 0.93 | 1.249 | 0.828 | 1.273 | 0.853 |
| | 336 | 0.754 | 0.595 | 1.852 | 0.965 | 0.984 | 0.727 | 0.816 | 0.619 | 0.945 | 0.653 | 0.731 | 0.594 | 0.761 | 0.624 | 0.935 | 0.644 | 1.179 | 0.832 | 1.347 | 0.87 | 1.202 | 0.811 | 1.254 | 0.857 |
| | 720 | - | - | - | - | - | - | - | - | - | - | - | - | - | - | - | - | - | - | - | - | - | - | - | - |
| | Avg. | 0.681 | **0.560** | 1.299 | 0.793 | 0.750 | 0.611 | 0.694 | 0.569 | 0.925 | 0.647 | **0.658** | 0.562 | 0.722 | 0.598 | 0.943 | 0.646 | 1.189 | 0.839 | 1.451 | 0.903 | 1.225 | 0.817 | 1.241 | 0.835 |
| ETTh2 | 96 | 0.376 | 0.421 | 0.551 | 0.507 | 0.442 | 0.456 | 0.401 | 0.421 | 0.409 | 0.420 | 0.390 | 0.424 | 0.428 | 0.468 | 0.408 | 0.423 | 0.678 | 0.619 | 2.022 | 1.006 | 3.837 | 1.508 | 3.753 | 1.518 |
| | 192 | 0.418 | 0.441 | 0.765 | 0.610 | 0.617 | 0.542 | 0.452 | 0.455 | 0.483 | 0.464 | 0.457 | 0.465 | 0.496 | 0.504 | 0.497 | 0.468 | 0.845 | 0.697 | 3.534 | 1.348 | 3.975 | 1.933 | 3.516 | 1.473 |
| | 336 | 0.408 | 0.439 | 0.767 | 0.614 | 1.424 | 0.849 | 0.464 | 0.469 | 0.499 | 0.479 | 0.477 | 0.483 | 0.486 | 0.496 | 0.507 | 0.481 | 0.905 | 0.727 | 4.063 | 1.451 | 3.956 | 1.520 | 3.312 | 1.427 |
| | 720 | - | - | - | - | - | - | - | - | - | - | - | - | - | - | - | - | - | - | - | - | - | - | - | - |
| | Avg. | **0.400** | **0.433** | 0.694 | 0.577 | 0.827 | 0.615 | 0.439 | 0.448 | 0.463 | 0.454 | 0.441 | 0.457 | 0.47 | 0.489 | 0.470 | 0.457 | 0.809 | 0.681 | 3.206 | 1.268 | 3.922 | 1.653 | 3.527 | 1.472 |
| ETTm1 | 96 | 0.386 | 0.405 | 0.582 | 0.512 | 0.332 | 0.374 | 0.399 | 0.414 | 0.606 | 0.518 | 0.628 | 0.544 | 0.726 | 0.578 | 0.823 | 0.587 | 1.031 | 0.747 | 1.048 | 0.733 | 1.130 | 0.775 | 1.234 | 0.798 |
| | 192 | 0.440 | 0.438 | 0.632 | 0.536 | 0.358 | 0.390 | 0.441 | 0.436 | 0.681 | 0.539 | 0.666 | 0.566 | 0.750 | 0.591 | 0.844 | 0.591 | 1.087 | 0.766 | 1.097 | 0.756 | 1.150 | 0.788 | 1.287 | 0.839 |
| | 336 | 0.485 | 0.459 | 0.767 | 0.584 | 0.402 | 0.416 | 0.499 | 0.467 | 0.786 | 0.597 | 0.807 | 0.628 | 0.851 | 0.659 | 0.870 | 0.603 | 1.138 | 0.787 | 1.147 | 0.775 | 1.198 | 0.809 | 1.288 | 0.842 |
| | 720 | 0.577 | 0.499 | 1.334 | 0.742 | 0.511 | 0.489 | 0.767 | 0.587 | 0.796 | 0.593 | 0.822 | 0.633 | 0.857 | 0.655 | 0.893 | 0.611 | 1.245 | 0.831 | 1.200 | 0.799 | 1.175 | 0.794 | 1.247 | 0.828 |
| | Avg. | 0.472 | 0.450 | 0.828 | 0.593 | **0.400** | **0.417** | 0.526 | 0.476 | 0.717 | 0.561 | 0.730 | 0.592 | 0.796 | 0.620 | 0.857 | 0.598 | 1.125 | 0.782 | 1.123 | 0.765 | 1.163 | 0.791 | 1.264 | 0.826 |
| ETTm2 | 96 | 0.199 | 0.280 | 0.282 | 0.347 | 0.236 | 0.326 | 0.206 | 0.288 | 0.220 | 0.299 | 0.229 | 0.320 | 0.232 | 0.322 | 0.238 | 0.316 | 0.404 | 0.485 | 1.108 | 0.772 | 3.599 | 1.478 | 3.883 | 1.545 |
| | 192 | 0.256 | 0.316 | 0.346 | 0.383 | 0.306 | 0.373 | 0.264 | 0.324 | 0.311 | 0.361 | 0.394 | 0.361 | 0.291 | 0.357 | 0.298 | 0.349 | 0.479 | 0.521 | 1.317 | 0.850 | 3.578 | 1.475 | 3.553 | 1.484 |
| | 336 | 0.318 | 0.353 | 0.429 | 0.427 | 0.380 | 0.423 | 0.334 | 0.367 | 0.338 | 0.366 | 0.378 | 0.427 | 0.478 | 0.517 | 0.353 | 0.380 | 0.552 | 0.555 | 1.415 | 0.879 | 3.561 | 1.473 | 3.446 | 1.460 |
| | 720 | 0.460 | 0.436 | 0.751 | 0.568 | 0.674 | 0.583 | 0.454 | 0.432 | 0.509 | 0.465 | 0.523 | 0.510 | 0.553 | 0.538 | 0.475 | 0.445 | 0.701 | 0.627 | 1.822 | 0.984 | 3.896 | 1.533 | 3.445 | 1.460 |
| | Avg. | **0.308** | **0.346** | 0.452 | 0.431 | 0.399 | 0.426 | 0.314 | 0.352 | 0.344 | 0.372 | 0.381 | 0.404 | 0.388 | 0.433 | 0.341 | 0.372 | 0.534 | 0.547 | 1.415 | 0.871 | 3.658 | 1.489 | 3.581 | 1.487 |
| ECL | 96 | 0.143 | 0.241 | 0.147 | 0.246 | 0.150 | 0.251 | 0.145 | 0.244 | 0.315 | 0.389 | 0.235 | 0.322 | 0.297 | 0.367 | 0.484 | 0.518 | 0.697 | 0.638 | 0.639 | 0.609 | 1.265 | 0.919 | 1.414 | 0.855 |
| | 192 | 0.159 | 0.255 | 0.163 | 0.260 | 0.163 | 0.263 | 0.163 | 0.260 | 0.318 | 0.396 | 0.247 | 0.341 | 0.308 | 0.375 | 0.501 | 0.531 | 0.718 | 0.648 | 0.772 | 0.678 | 1.298 | 0.939 | 1.240 | 0.919 |
| | 336 | 0.179 | 0.274 | 0.182 | 0.278 | 0.175 | 0.278 | 0.183 | 0.281 | 0.340 | 0.415 | 0.267 | 0.356 | 0.354 | 0.411 | 0.574 | 0.578 | 0.758 | 0.667 | 0.901 | 0.745 | 1.302 | 0.942 | 1.253 | 0.921 |
| | 720 | 0.233 | 0.323 | 0.239 | 0.329 | 0.219 | 0.311 | 0.233 | 0.323 | 0.635 | 0.613 | 0.318 | 0.394 | 0.426 | 0.466 | 0.952 | 0.786 | 1.028 | 0.788 | 1.200 | 0.871 | 1.259 | 0.919 | 1.249 | 0.921 |
| | Avg. | 0.178 | **0.273** | 0.182 | 0.278 | **0.176** | 0.275 | 0.181 | 0.277 | 0.402 | 0.453 | 0.266 | 0.353 | 0.346 | 0.404 | 0.627 | 0.603 | 0.800 | 0.685 | 0.878 | 0.725 | 1.281 | 0.929 | 1.289 | 0.904 |
| Traffic | 96 | 0.419 | 0.298 | 0.468 | 0.354 | 0.427 | 0.304 | 0.404 | 0.286 | 0.854 | 0.492 | 0.670 | 0.421 | 0.795 | 0.481 | 1.468 | 0.821 | 1.643 | 0.855 | 1.157 | 0.636 | 1.557 | 0.821 | 1.586 | 0.841 |
| | 192 | 0.434 | 0.305 | 0.479 | 0.352 | 0.447 | 0.315 | 0.412 | 0.294 | 0.894 | 0.517 | 0.653 | 0.405 | 0.837 | 0.503 | 1.509 | 0.838 | 1.856 | 0.928 | 1.688 | 0.848 | 1.596 | 0.834 | 1.602 | 0.844 |
| | 336 | 0.449 | 0.313 | 0.477 | 0.345 | 0.478 | 0.333 | 0.439 | 0.310 | 0.853 | 0.471 | 0.707 | 0.445 | 0.867 | 0.523 | 1.602 | 0.860 | 2.080 | 0.999 | 1.826 | 0.903 | 1.621 | 0.841 | 1.668 | 0.868 |
| | 720 | - | - | - | - | - | - | - | - | - | - | - | - | - | - | - | - | - | - | - | - | - | - | - | - |
| | Avg. | 0.434 | 0.305 | 0.474 | 0.350 | 0.450 | 0.317 | **0.418** | **0.296** | 0.867 | 0.493 | 0.676 | 0.423 | 0.833 | 0.502 | 1.526 | 0.839 | 1.859 | 0.927 | 1.557 | 0.795 | 1.591 | 0.832 | 1.618 | 0.851 |
| Average | | **0.377** | **0.375** | 0.575 | 0.465 | 0.441 | 0.413 | **0.392** | **0.383** | 0.552 | 0.464 | 0.483 | 0.445 | 0.537 | 0.480 | 0.697 | 0.537 | 0.909 | 0.675 | 1.341 | 0.789 | 1.878 | 0.994 | 1.819 | 0.966 |

Table 13: Few-shot learning results on 10% data. We use prediction length $O \in \{96, 192, 336, 720\}$. A lower MSE indicates better performance, and the best results are highlighted in bold. '-' means that 10% time series is not sufficient to constitute a training set.

| Methods | | GPT2(6) | | GPT2(0) | | DLinear | | PatchTST | | TimesNet | | FEDformer | | Autoformer | | Stationary | | ETSformer | | LightTS | | Informer | | Reformer | |
|---|---|---|---|---|---|---|---|---|---|---|---|---|---|---|---|---|---|---|---|---|---|---|---|---|---|
| Metric | | MSE | MAE | MSE | MAE | MSE | MAE | MSE | MAE | MSE | MAE | MSE | MAE | MSE | MAE | MSE | MAE | MSE | MAE | MSE | MAE | MSE | MAE | MSE | MAE |
| Weather | 96 | 0.163 | 0.215 | 0.190 | 0.240 | 0.171 | 0.224 | 0.165 | 0.215 | 0.184 | 0.230 | 0.188 | 0.253 | 0.221 | 0.297 | 0.192 | 0.234 | 0.199 | 0.272 | 0.217 | 0.269 | 0.374 | 0.401 | 0.335 | 0.380 |
| | 192 | 0.210 | 0.254 | 0.243 | 0.284 | 0.215 | 0.263 | 0.210 | 0.257 | 0.245 | 0.283 | 0.250 | 0.304 | 0.270 | 0.322 | 0.269 | 0.295 | 0.279 | 0.332 | 0.259 | 0.304 | 0.552 | 0.478 | 0.522 | 0.462 |
| | 336 | 0.256 | 0.292 | 0.270 | 0.305 | 0.258 | 0.299 | 0.259 | 0.297 | 0.305 | 0.321 | 0.312 | 0.346 | 0.320 | 0.351 | 0.370 | 0.357 | 0.356 | 0.386 | 0.303 | 0.334 | 0.724 | 0.541 | 0.715 | 0.535 |
| | 720 | 0.321 | 0.339 | 0.348 | 0.359 | 0.320 | 0.346 | 0.332 | 0.346 | 0.381 | 0.371 | 0.387 | 0.393 | 0.390 | 0.396 | 0.441 | 0.405 | 0.437 | 0.448 | 0.377 | 0.382 | 0.739 | 0.558 | 0.611 | 0.500 |
| | Avg. | **0.238** | **0.275** | 0.263 | 0.297 | 0.241 | 0.283 | 0.242 | 0.279 | 0.279 | 0.301 | 0.284 | 0.324 | 0.300 | 0.342 | 0.318 | 0.323 | 0.318 | 0.360 | 0.289 | 0.322 | 0.597 | 0.495 | 0.546 | 0.469 |
| ETTh1 | 96 | 0.458 | 0.456 | 0.601 | 0.536 | 0.492 | 0.495 | 0.516 | 0.485 | 0.861 | 0.628 | 0.512 | 0.499 | 0.613 | 0.552 | 0.918 | 0.639 | 1.112 | 0.806 | 1.298 | 0.838 | 1.179 | 0.792 | 1.184 | 0.790 |
| | 192 | 0.570 | 0.516 | 0.709 | 0.587 | 0.565 | 0.538 | 0.598 | 0.524 | 0.797 | 0.593 | 0.624 | 0.555 | 0.722 | 0.598 | 0.915 | 0.629 | 1.155 | 0.823 | 1.322 | 0.854 | 1.199 | 0.806 | 1.295 | 0.850 |
| | 336 | 0.608 | 0.535 | 0.801 | 0.635 | 0.721 | 0.622 | 0.657 | 0.550 | 0.941 | 0.648 | 0.691 | 0.574 | 0.750 | 0.619 | 0.939 | 0.644 | 1.179 | 0.832 | 1.347 | 0.870 | 1.202 | 0.811 | 1.294 | 0.854 |
| | 720 | 0.725 | 0.591 | 1.385 | 0.831 | 0.986 | 0.743 | 0.762 | 0.610 | 0.877 | 0.641 | 0.728 | 0.614 | 0.721 | 0.616 | 0.887 | 0.645 | 1.273 | 0.874 | 1.534 | 0.947 | 1.217 | 0.825 | 1.223 | 0.838 |
| | Avg. | **0.590** | **0.525** | 0.874 | 0.647 | 0.691 | 0.600 | 0.633 | 0.542 | 0.869 | 0.628 | 0.639 | 0.561 | 0.702 | 0.596 | 0.915 | 0.639 | 1.180 | 0.834 | 1.375 | 0.877 | 1.199 | 0.809 | 1.249 | 0.833 |
| ETTh2 | 96 | 0.331 | 0.374 | 0.539 | 0.495 | 0.357 | 0.411 | 0.353 | 0.389 | 0.378 | 0.409 | 0.382 | 0.416 | 0.413 | 0.451 | 0.389 | 0.411 | 0.678 | 0.619 | 2.022 | 1.006 | 3.837 | 1.508 | 3.788 | 1.533 |
| | 192 | 0.402 | 0.411 | 0.675 | 0.555 | 0.569 | 0.519 | 0.403 | 0.414 | 0.490 | 0.467 | 0.478 | 0.474 | 0.474 | 0.477 | 0.473 | 0.455 | 0.785 | 0.666 | 2.329 | 1.104 | 3.856 | 1.513 | 3.552 | 1.483 |
| | 336 | 0.406 | 0.433 | 0.718 | 0.580 | 0.671 | 0.572 | 0.426 | 0.441 | 0.537 | 0.494 | 0.504 | 0.501 | 0.547 | 0.543 | 0.507 | 0.480 | 0.839 | 0.694 | 2.453 | 1.122 | 3.952 | 1.526 | 3.395 | 1.526 |
| | 720 | 0.449 | 0.464 | 0.732 | 0.605 | 0.824 | 0.648 | 0.477 | 0.480 | 0.510 | 0.491 | 0.499 | 0.509 | 0.516 | 0.523 | 0.477 | 0.472 | 1.273 | 0.874 | 3.816 | 1.407 | 3.842 | 1.503 | 3.205 | 1.401 |
| | Avg. | **0.397** | **0.421** | 0.666 | 0.559 | 0.605 | 0.538 | 0.415 | 0.431 | 0.479 | 0.465 | 0.466 | 0.475 | 0.488 | 0.499 | 0.462 | 0.455 | 0.894 | 0.713 | 2.655 | 1.160 | 3.872 | 1.513 | 3.485 | 1.486 |
| ETTm1 | 96 | 0.390 | 0.404 | 0.610 | 0.508 | 0.352 | 0.392 | 0.410 | 0.419 | 0.583 | 0.501 | 0.578 | 0.518 | 0.774 | 0.614 | 0.761 | 0.568 | 0.911 | 0.688 | 0.921 | 0.682 | 1.162 | 0.785 | 1.442 | 0.847 |
| | 192 | 0.429 | 0.423 | 0.666 | 0.540 | 0.382 | 0.412 | 0.437 | 0.434 | 0.630 | 0.528 | 0.617 | 0.546 | 0.754 | 0.592 | 0.781 | 0.574 | 0.955 | 0.703 | 0.957 | 0.701 | 1.172 | 0.793 | 1.444 | 0.862 |
| | 336 | 0.469 | 0.439 | 0.895 | 0.615 | 0.419 | 0.434 | 0.476 | 0.454 | 0.725 | 0.568 | 0.998 | 0.775 | 0.869 | 0.677 | 0.803 | 0.587 | 0.991 | 0.719 | 0.998 | 0.716 | 1.227 | 0.908 | 1.450 | 0.866 |
| | 720 | 0.569 | 0.498 | 0.916 | 0.646 | 0.490 | 0.477 | 0.681 | 0.556 | 0.769 | 0.549 | 0.693 | 0.579 | 0.810 | 0.630 | 0.844 | 0.581 | 1.062 | 0.747 | 1.007 | 0.719 | 1.207 | 0.797 | 1.366 | 0.850 |
| | Avg. | 0.464 | 0.441 | 0.772 | 0.577 | **0.411** | **0.429** | 0.501 | 0.466 | 0.677 | 0.537 | 0.722 | 0.605 | 0.802 | 0.628 | 0.797 | 0.578 | 0.980 | 0.714 | 0.971 | 0.705 | 1.192 | 0.821 | 1.426 | 0.856 |
| ETTm2 | 96 | 0.188 | 0.269 | 0.283 | 0.344 | 0.213 | 0.303 | 0.191 | 0.274 | 0.212 | 0.285 | 0.291 | 0.399 | 0.352 | 0.454 | 0.229 | 0.308 | 0.331 | 0.430 | 0.813 | 0.688 | 3.203 | 1.407 | 4.195 | 1.628 |
| | 192 | 0.251 | 0.309 | 0.353 | 0.384 | 0.278 | 0.345 | 0.252 | 0.317 | 0.270 | 0.323 | 0.307 | 0.379 | 0.694 | 0.691 | 0.291 | 0.343 | 0.400 | 0.464 | 1.008 | 0.768 | 3.112 | 1.387 | 4.042 | 1.601 |
| | 336 | 0.307 | 0.346 | 0.420 | 0.422 | 0.338 | 0.385 | 0.306 | 0.353 | 0.323 | 0.353 | 0.543 | 0.559 | 2.408 | 1.407 | 0.348 | 0.376 | 0.469 | 0.498 | 1.031 | 0.775 | 3.255 | 1.421 | 3.963 | 1.585 |
| | 720 | 0.426 | 0.417 | 0.553 | 0.491 | 0.436 | 0.440 | 0.433 | 0.427 | 0.474 | 0.449 | 0.712 | 0.614 | 1.913 | 1.166 | 0.461 | 0.438 | 0.589 | 0.557 | 1.096 | 0.791 | 3.909 | 1.543 | 3.711 | 1.532 |
| | Avg. | **0.293** | **0.335** | 0.402 | 0.410 | 0.316 | 0.368 | 0.296 | 0.343 | 0.320 | 0.353 | 0.463 | 0.488 | 1.342 | 0.930 | 0.332 | 0.366 | 0.447 | 0.487 | 0.987 | 0.756 | 3.370 | 1.440 | 3.978 | 1.587 |
| ECL | 96 | 0.139 | 0.237 | 0.142 | 0.240 | 0.150 | 0.253 | 0.140 | 0.238 | 0.299 | 0.373 | 0.231 | 0.323 | 0.261 | 0.348 | 0.420 | 0.466 | 0.599 | 0.587 | 0.350 | 0.425 | 1.259 | 0.919 | 0.993 | 0.784 |
| | 192 | 0.156 | 0.252 | 0.158 | 0.254 | 0.164 | 0.264 | 0.160 | 0.255 | 0.305 | 0.379 | 0.261 | 0.356 | 0.338 | 0.406 | 0.411 | 0.459 | 0.620 | 0.598 | 0.376 | 0.448 | 1.160 | 0.873 | 0.938 | 0.753 |
| | 336 | 0.175 | 0.270 | 0.175 | 0.271 | 0.181 | 0.282 | 0.180 | 0.276 | 0.319 | 0.391 | 0.360 | 0.445 | 0.410 | 0.474 | 0.434 | 0.473 | 0.662 | 0.619 | 0.428 | 0.485 | 1.157 | 0.872 | 0.925 | 0.745 |
| | 720 | 0.233 | 0.317 | 0.230 | 0.315 | 0.223 | 0.321 | 0.241 | 0.323 | 0.369 | 0.426 | 0.530 | 0.585 | 0.715 | 0.685 | 0.510 | 0.521 | 0.757 | 0.664 | 0.611 | 0.597 | 1.203 | 0.898 | 1.004 | 0.790 |
| | Avg. | **0.176** | **0.269** | 0.176 | 0.270 | 0.180 | 0.280 | 0.180 | 0.273 | 0.323 | 0.392 | 0.346 | 0.427 | 0.431 | 0.478 | 0.444 | 0.480 | 0.660 | 0.617 | 0.441 | 0.489 | 1.195 | 0.891 | 0.965 | 0.768 |
| Traffic | 96 | 0.414 | 0.297 | 0.478 | 0.368 | 0.419 | 0.298 | 0.403 | 0.289 | 0.719 | 0.416 | 0.639 | 0.400 | 0.672 | 0.405 | 1.412 | 0.802 | 1.643 | 0.855 | 1.157 | 0.636 | 1.557 | 0.821 | 1.527 | 0.815 |
| | 192 | 0.426 | 0.301 | 0.481 | 0.363 | 0.434 | 0.305 | 0.415 | 0.296 | 0.748 | 0.428 | 0.637 | 0.416 | 0.727 | 0.424 | 1.419 | 0.806 | 1.641 | 0.854 | 1.207 | 0.661 | 1.454 | 0.765 | 1.538 | 0.817 |
| | 336 | 0.434 | 0.303 | 0.488 | 0.365 | 0.449 | 0.313 | 0.426 | 0.304 | 0.853 | 0.471 | 0.655 | 0.427 | 0.749 | 0.454 | 1.443 | 0.815 | 1.711 | 0.878 | 1.334 | 0.713 | 1.521 | 0.812 | 1.550 | 0.819 |
| | 720 | 0.487 | 0.337 | 0.537 | 0.386 | 0.484 | 0.336 | 0.474 | 0.331 | 1.485 | 0.825 | 0.722 | 0.456 | 0.847 | 0.499 | 1.539 | 0.837 | 2.660 | 1.157 | 1.292 | 0.726 | 1.605 | 0.846 | 1.588 | 0.833 |
| | Avg. | 0.440 | 0.310 | 0.496 | 0.371 | 0.447 | 0.313 | **0.430** | **0.305** | 0.951 | 0.535 | 0.663 | 0.425 | 0.749 | 0.446 | 1.453 | 0.815 | 1.914 | 0.936 | 1.248 | 0.684 | 1.534 | 0.811 | 1.551 | 0.821 |
| Average | | **0.371** | **0.367** | 0.521 | 0.447 | 0.413 | 0.401 | 0.385 | 0.376 | 0.556 | 0.458 | 0.511 | 0.472 | 0.687 | 0.559 | 0.674 | 0.522 | 0.912 | 0.665 | 1.137 | 0.712 | 1.850 | 0.967 | 1.888 | 0.974 |

## D.3 Long-term Time-series Forecasting

Here we investigate whether our architecture performs consistently well with more training data. Thus, we follow the classical experiment settings of Nie et al. (2022) and conduct experiments on full data. The results are shown in Table 14. Overall, GPT2(6) FPT achieves comparable performance to PatchTST, Dlinear and outperforms other baselines by a large margin. Compared with the second best transformer-based baseline method FEDformer, GPT2(6) FPT yields an overall **18.7%** relatively MSE reduction. It verifies the effectiveness of NLP pretrained model in time series forecasting, not limited to the few-shot setting.

**Detail Experiment Table for Long-term Time-Series Forecasting** in table 14

Table 14: Full results on full data. We use prediction length $O \in \{96, 192, 336, 720\}$ for ILI and $O \in \{24, 36, 48, 60\}$ for others. A lower MSE indicates better performance. **Black**: best, Red: second best.

| Methods | | GPT2(6) | | GPT2(0) | | DLinear | | PatchTST | | TimesNet | | FEDformer | | Autoformer | | Stationary | | ETSformer | | LightTS | | Informer | | Reformer | |
|---|---|---|---|---|---|---|---|---|---|---|---|---|---|---|---|---|---|---|---|---|---|---|---|---|---|
| Metric | | MSE | MAE | MSE | MAE | MSE | MAE | MSE | MAE | MSE | MAE | MSE | MAE | MSE | MAE | MSE | MAE | MSE | MAE | MSE | MAE | MSE | MAE | MSE | MAE |
| Weather | 96 | 0.162 | 0.212 | 0.181 | 0.232 | 0.176 | 0.237 | 0.149 | 0.198 | 0.172 | 0.220 | 0.217 | 0.296 | 0.266 | 0.336 | 0.173 | 0.223 | 0.197 | 0.281 | 0.182 | 0.242 | 0.300 | 0.384 | 0.689 | 0.596 |
| | 192 | 0.204 | 0.248 | 0.222 | 0.266 | 0.220 | 0.282 | 0.194 | 0.241 | 0.219 | 0.261 | 0.276 | 0.336 | 0.307 | 0.367 | 0.245 | 0.285 | 0.237 | 0.312 | 0.227 | 0.287 | 0.598 | 0.544 | 0.752 | 0.638 |
| | 336 | 0.254 | 0.286 | 0.270 | 0.299 | 0.265 | 0.319 | 0.245 | 0.282 | 0.280 | 0.306 | 0.339 | 0.380 | 0.359 | 0.395 | 0.321 | 0.338 | 0.298 | 0.353 | 0.282 | 0.334 | 0.578 | 0.523 | 0.639 | 0.596 |
| | 720 | 0.326 | 0.337 | 0.338 | 0.345 | 0.333 | 0.362 | 0.314 | 0.334 | 0.365 | 0.359 | 0.403 | 0.428 | 0.419 | 0.428 | 0.414 | 0.410 | 0.352 | 0.288 | 0.352 | 0.386 | 1.059 | 0.741 | 1.130 | 0.792 |
| | Avg | 0.237 | 0.270 | 0.252 | 0.285 | 0.248 | 0.300 | 0.225 | 0.264 | 0.259 | 0.287 | 0.309 | 0.360 | 0.338 | 0.382 | 0.288 | 0.314 | 0.271 | 0.334 | 0.261 | 0.312 | 0.634 | 0.548 | 0.803 | 0.656 |
| ETTh1 | 96 | 0.376 | 0.397 | 0.422 | 0.428 | 0.375 | 0.399 | 0.370 | 0.399 | 0.384 | 0.402 | 0.376 | 0.419 | 0.449 | 0.459 | 0.513 | 0.491 | 0.494 | 0.479 | 0.424 | 0.432 | 0.865 | 0.713 | 0.837 | 0.728 |
| | 192 | 0.416 | 0.418 | 0.466 | 0.450 | 0.405 | 0.416 | 0.413 | 0.421 | 0.436 | 0.429 | 0.420 | 0.448 | 0.500 | 0.482 | 0.534 | 0.504 | 0.538 | 0.504 | 0.475 | 0.462 | 1.008 | 0.792 | 0.923 | 0.766 |
| | 336 | 0.442 | 0.433 | 0.488 | 0.464 | 0.439 | 0.443 | 0.422 | 0.436 | 0.491 | 0.469 | 0.459 | 0.465 | 0.521 | 0.496 | 0.588 | 0.535 | 0.574 | 0.521 | 0.518 | 0.488 | 1.107 | 0.809 | 1.097 | 0.835 |
| | 720 | 0.477 | 0.456 | 0.485 | 0.478 | 0.472 | 0.490 | 0.447 | 0.466 | 0.521 | 0.500 | 0.506 | 0.507 | 0.514 | 0.512 | 0.643 | 0.616 | 0.562 | 0.535 | 0.547 | 0.533 | 1.181 | 0.865 | 1.257 | 0.889 |
| | Avg | 0.427 | 0.426 | 0.465 | 0.455 | 0.422 | 0.437 | 0.413 | 0.430 | 0.458 | 0.450 | 0.440 | 0.460 | 0.496 | 0.487 | 0.570 | 0.537 | 0.542 | 0.510 | 0.491 | 0.479 | 1.040 | 0.795 | 1.029 | 0.805 |
| ETTh2 | 96 | 0.285 | 0.342 | 0.318 | 0.368 | 0.289 | 0.353 | 0.274 | 0.336 | 0.340 | 0.374 | 0.358 | 0.397 | 0.346 | 0.388 | 0.476 | 0.458 | 0.340 | 0.391 | 0.397 | 0.437 | 3.755 | 1.525 | 2.626 | 1.317 |
| | 192 | 0.354 | 0.389 | 0.383 | 0.407 | 0.383 | 0.418 | 0.339 | 0.379 | 0.402 | 0.414 | 0.429 | 0.439 | 0.456 | 0.452 | 0.512 | 0.493 | 0.430 | 0.439 | 0.520 | 0.504 | 5.602 | 1.931 | 11.12 | 2.979 |
| | 336 | 0.373 | 0.407 | 0.406 | 0.427 | 0.448 | 0.465 | 0.329 | 0.380 | 0.452 | 0.452 | 0.496 | 0.487 | 0.482 | 0.486 | 0.552 | 0.551 | 0.485 | 0.479 | 0.626 | 0.559 | 4.721 | 1.835 | 9.323 | 2.769 |
| | 720 | 0.406 | 0.441 | 0.420 | 0.446 | 0.605 | 0.551 | 0.379 | 0.422 | 0.462 | 0.468 | 0.463 | 0.474 | 0.515 | 0.511 | 0.562 | 0.560 | 0.500 | 0.497 | 0.863 | 0.672 | 3.647 | 1.625 | 3.874 | 1.697 |
| | Avg | 0.354 | 0.394 | 0.381 | 0.412 | 0.431 | 0.446 | 0.330 | 0.379 | 0.414 | 0.427 | 0.437 | 0.449 | 0.450 | 0.459 | 0.526 | 0.516 | 0.439 | 0.452 | 0.602 | 0.543 | 4.431 | 1.729 | 6.736 | 2.191 |
| ETTm1 | 96 | 0.292 | 0.346 | 0.330 | 0.372 | 0.299 | 0.343 | 0.290 | 0.342 | 0.338 | 0.375 | 0.379 | 0.419 | 0.505 | 0.475 | 0.386 | 0.398 | 0.375 | 0.398 | 0.374 | 0.400 | 0.672 | 0.571 | 0.538 | 0.528 |
| | 192 | 0.332 | 0.372 | 0.371 | 0.394 | 0.335 | 0.365 | 0.332 | 0.369 | 0.374 | 0.387 | 0.426 | 0.441 | 0.553 | 0.496 | 0.459 | 0.444 | 0.408 | 0.410 | 0.400 | 0.407 | 0.795 | 0.669 | 0.658 | 0.592 |
| | 336 | 0.366 | 0.394 | 0.398 | 0.409 | 0.369 | 0.386 | 0.366 | 0.392 | 0.410 | 0.411 | 0.445 | 0.459 | 0.621 | 0.537 | 0.495 | 0.464 | 0.435 | 0.428 | 0.438 | 0.438 | 1.212 | 0.871 | 0.898 | 0.721 |
| | 720 | 0.417 | 0.421 | 0.454 | 0.440 | 0.425 | 0.421 | 0.416 | 0.420 | 0.478 | 0.450 | 0.543 | 0.490 | 0.671 | 0.561 | 0.585 | 0.516 | 0.499 | 0.462 | 0.527 | 0.502 | 1.166 | 0.823 | 1.102 | 0.841 |
| | Avg | 0.352 | 0.383 | 0.388 | 0.403 | 0.357 | 0.378 | 0.351 | 0.380 | 0.400 | 0.406 | 0.448 | 0.452 | 0.588 | 0.517 | 0.481 | 0.456 | 0.429 | 0.425 | 0.435 | 0.437 | 0.961 | 0.734 | 0.799 | 0.671 |
| ETTm2 | 96 | 0.173 | 0.262 | 0.192 | 0.281 | 0.167 | 0.269 | 0.165 | 0.255 | 0.187 | 0.267 | 0.203 | 0.287 | 0.255 | 0.339 | 0.192 | 0.274 | 0.189 | 0.280 | 0.209 | 0.308 | 0.365 | 0.453 | 0.658 | 0.619 |
| | 192 | 0.229 | 0.301 | 0.245 | 0.317 | 0.224 | 0.303 | 0.220 | 0.292 | 0.249 | 0.309 | 0.269 | 0.328 | 0.281 | 0.340 | 0.280 | 0.339 | 0.253 | 0.319 | 0.311 | 0.382 | 0.533 | 0.563 | 1.078 | 0.827 |
| | 336 | 0.286 | 0.341 | 0.302 | 0.352 | 0.281 | 0.342 | 0.274 | 0.329 | 0.321 | 0.351 | 0.325 | 0.366 | 0.339 | 0.372 | 0.334 | 0.361 | 0.314 | 0.357 | 0.442 | 0.466 | 1.363 | 0.887 | 1.549 | 0.972 |
| | 720 | 0.378 | 0.401 | 0.399 | 0.408 | 0.397 | 0.421 | 0.362 | 0.385 | 0.408 | 0.403 | 0.421 | 0.415 | 0.433 | 0.432 | 0.417 | 0.413 | 0.414 | 0.413 | 0.675 | 0.587 | 3.379 | 1.338 | 2.631 | 1.242 |
| | Avg | 0.266 | 0.326 | 0.284 | 0.339 | 0.267 | 0.333 | 0.255 | 0.315 | 0.291 | 0.333 | 0.305 | 0.349 | 0.327 | 0.371 | 0.306 | 0.347 | 0.293 | 0.342 | 0.409 | 0.436 | 1.410 | 0.810 | 1.479 | 0.915 |
| ILI | 24 | 2.063 | 0.881 | 2.723 | 1.099 | 2.215 | 1.081 | 1.319 | 0.754 | 2.317 | 0.934 | 3.228 | 1.260 | 3.483 | 1.287 | 2.294 | 0.945 | 2.527 | 1.020 | 8.313 | 2.144 | 5.764 | 1.677 | 4.400 | 1.382 |
| | 36 | 1.868 | 0.892 | 2.027 | 0.966 | 1.963 | 0.963 | 1.430 | 0.834 | 1.972 | 0.920 | 2.679 | 1.080 | 3.103 | 1.148 | 1.825 | 0.848 | 2.615 | 1.007 | 6.631 | 1.902 | 4.755 | 1.467 | 4.783 | 1.448 |
| | 48 | 1.790 | 0.884 | 2.206 | 1.022 | 2.130 | 1.024 | 1.553 | 0.815 | 2.238 | 0.940 | 2.622 | 1.078 | 2.669 | 1.085 | 2.010 | 0.900 | 2.359 | 0.972 | 7.299 | 1.982 | 4.763 | 1.469 | 4.832 | 1.465 |
| | 60 | 1.979 | 0.957 | 1.976 | 0.983 | 2.368 | 1.096 | 1.470 | 0.788 | 2.027 | 0.928 | 2.857 | 1.157 | 2.770 | 1.125 | 2.178 | 0.963 | 2.487 | 1.016 | 7.283 | 1.985 | 5.264 | 1.564 | 4.882 | 1.483 |
| | Avg | 1.925 | 0.903 | 2.233 | 1.017 | 2.169 | 1.041 | 1.443 | 0.797 | 2.139 | 0.931 | 2.847 | 1.144 | 3.006 | 1.161 | 2.077 | 0.914 | 2.497 | 1.004 | 7.382 | 2.003 | 5.137 | 1.544 | 4.724 | 1.445 |
| ECL | 96 | 0.139 | 0.238 | 0.138 | 0.234 | 0.140 | 0.237 | 0.129 | 0.222 | 0.168 | 0.272 | 0.193 | 0.308 | 0.201 | 0.317 | 0.169 | 0.273 | 0.187 | 0.304 | 0.207 | 0.307 | 0.274 | 0.368 | 0.312 | 0.402 |
| | 192 | 0.153 | 0.251 | 0.152 | 0.247 | 0.153 | 0.249 | 0.157 | 0.240 | 0.184 | 0.289 | 0.201 | 0.315 | 0.222 | 0.334 | 0.182 | 0.286 | 0.199 | 0.315 | 0.213 | 0.316 | 0.296 | 0.386 | 0.348 | 0.433 |
| | 336 | 0.169 | 0.266 | 0.168 | 0.263 | 0.169 | 0.267 | 0.163 | 0.259 | 0.198 | 0.300 | 0.214 | 0.329 | 0.231 | 0.338 | 0.200 | 0.304 | 0.212 | 0.329 | 0.230 | 0.333 | 0.300 | 0.394 | 0.350 | 0.433 |
| | 720 | 0.206 | 0.297 | 0.207 | 0.295 | 0.203 | 0.301 | 0.197 | 0.290 | 0.220 | 0.320 | 0.246 | 0.355 | 0.254 | 0.361 | 0.222 | 0.321 | 0.233 | 0.345 | 0.265 | 0.360 | 0.373 | 0.439 | 0.340 | 0.420 |
| | Avg | 0.167 | 0.263 | 0.166 | 0.259 | 0.166 | 0.263 | 0.161 | 0.252 | 0.192 | 0.295 | 0.214 | 0.327 | 0.227 | 0.338 | 0.193 | 0.296 | 0.208 | 0.323 | 0.229 | 0.329 | 0.311 | 0.397 | 0.338 | 0.422 |
| Traffic | 96 | 0.388 | 0.282 | 0.390 | 0.272 | 0.410 | 0.282 | 0.360 | 0.249 | 0.593 | 0.321 | 0.587 | 0.366 | 0.613 | 0.388 | 0.612 | 0.338 | 0.607 | 0.392 | 0.615 | 0.391 | 0.719 | 0.391 | 0.732 | 0.423 |
| | 192 | 0.407 | 0.290 | 0.403 | 0.276 | 0.423 | 0.287 | 0.379 | 0.256 | 0.617 | 0.336 | 0.604 | 0.373 | 0.616 | 0.382 | 0.613 | 0.340 | 0.621 | 0.399 | 0.601 | 0.382 | 0.696 | 0.379 | 0.733 | 0.420 |
| | 336 | 0.412 | 0.294 | 0.413 | 0.280 | 0.436 | 0.296 | 0.392 | 0.264 | 0.629 | 0.336 | 0.621 | 0.383 | 0.622 | 0.337 | 0.618 | 0.328 | 0.622 | 0.396 | 0.613 | 0.386 | 0.777 | 0.420 | 0.742 | 0.420 |
| | 720 | 0.450 | 0.312 | 0.447 | 0.298 | 0.466 | 0.315 | 0.432 | 0.286 | 0.640 | 0.350 | 0.626 | 0.382 | 0.660 | 0.408 | 0.653 | 0.355 | 0.632 | 0.396 | 0.658 | 0.407 | 0.864 | 0.472 | 0.755 | 0.423 |
| | Avg | 0.414 | 0.294 | 0.413 | 0.281 | 0.433 | 0.295 | 0.390 | 0.263 | 0.620 | 0.336 | 0.610 | 0.376 | 0.628 | 0.379 | 0.624 | 0.340 | 0.621 | 0.396 | 0.622 | 0.392 | 0.764 | 0.416 | 0.741 | 0.422 |
| Average | | 0.516 | 0.407 | 0.573 | 0.0.431 | 0.562 | 0.436 | 0.446 | 0.386 | 0.596 | 0.433 | 0.701 | 0.489 | 0.757 | 0.511 | 0.633 | 0.465 | 0.662 | 0.473 | 1.303 | 0.616 | 1.836 | 0.871 | 2.081 | 0.954 |

### D.4 Mean and STD for Few-shot Learning

Table 15 lists both mean and STD for GPT2(6), DLinear and PatchTST with 3 runs on 5% ETTh2 and ETTm2. The results show a small variance in performance of GPT2(6) that represents the stability of GPT2(6).

Table 15: A subset of results showing both Mean and STD on 5% datasets.

| Methods | | GPT2-backbone(6 Layers) | |
|---|---|---|---|
| Metric | | MSE | MAE |
| ETTh2 | 96 | $0.376 \pm 0.0072$ | $0.421 \pm 0.0054$ |
| | 192 | $0.418 \pm 0.0013$ | $0.441 \pm 0.0014$ |
| | 336 | $0.408 \pm 0.0006$ | $0.439 \pm 0.0002$ |
| | 720 | - | - |
| ETTm2 | 96 | $0.199 \pm 0.0040$ | $0.280 \pm 0.0042$ |
| | 192 | $0.256 \pm 0.0030$ | $0.316 \pm 0.0017$ |
| | 336 | $0.318 \pm 0.0046$ | $0.353 \pm 0.0032$ |
| | 720 | $0.460 \pm 0.0132$ | $0.436 \pm 0.0066$ |

### D.5 Comparison with Traditional Methods on Few-shot Learning

Since deep learning methods are more advantageous than traditional methods when applied to large datasets. For few-shot learning, traditional methods should also consider. The results are shown in Table 16 that GPT2(6) also achieves best performance.

Table 16: Comparison with traditional methods.

| Methods | | GPT2(6) 5% | | GPT2(6) 10% | | ETS | | ARIMA | | NaiveDrift | |
|---|---|---|---|---|---|---|---|---|---|---|---|
| Metric | | MSE | MAE | MSE | MAE | MSE | MAE | MSE | MAE | MSE | MAE |
| ETTh2 | 96 | 0.376 | 0.421 | 0.331 | 0.374 | 2.954 | 0.742 | 0.481 | 0.443 | 0.764 | 0.561 |
| | 192 | 0.418 | 0.441 | 0.402 | 0.411 | 10.226 | 1.212 | 0.585 | 0.495 | 1.560 | 0.785 |
| ETTm1 | 96 | 0.386 | 0.405 | 0.390 | 0.404 | 52.237 | 2.689 | 0.693 | 0.547 | 1.539 | 0.913 |
| | 192 | 0.440 | 0.438 | 0.429 | 0.423 | 186.445 | 4.654 | 0.710 | 0.557 | 2.869 | 1.215 |

### D.6 Baselines with Instance Normalization

Instance normalization Kim et al. (2022) is a plug-in for time series for distribution shift. Most baselines, such as Autoformer and FEDformer are not equipped with instance normalization. Thus, for a fair comparison, we add the experiment, as in Table 17, for baselines w/o instance normalization and GPT(6) can also perform superior.

Table 17: Comparison on 5% data. Autoformer and FEDformer are equiped with instance normalization.

| Methods | | GPT2(6) | | PatchTST | | DLinear | | Autoformer | | Autoformer(Revin) | | FEDformer | | FEDformer(Revin) | |
|---|---|---|---|---|---|---|---|---|---|---|---|---|---|---|---|
| Metric | | MSE | MAE | MSE | MAE | MSE | MAE | MSE | MAE | MSE | MAE | MSE | MAE | MSE | MAE |
| ETTm2 | 96 | 0.199 | 0.280 | 0.206 | 0.288 | 0.236 | 0.326 | 0.232 | 0.322 | 0.224 | 0.300 | 0.229 | 0.320 | 0.223 | 0.298 |
| | 192 | 0.256 | 0.316 | 0.264 | 0.324 | 0.306 | 0.373 | 0.291 | 0.357 | 0.296 | 0.343 | 0.294 | 0.361 | 0.288 | 0.336 |

### D.7 Detailed Definition and Results of Zero-shot Learning

**Task Definition** Each experiment contains two distinct datasets, source, and target datasets. The source dataset is used to train the model and then forecasts without fine-tuning in the target dataset. The target dataset is split into non-overlapping historical and test sequences. We use the historical sequence as input to the model, and the obtained output is used to calculate errors with the test sequences. Besides meta-learning-based models like N-BEATS, evaluated models' parameters are not allowed any adjustment using the forecasting phase. Also, same as Oreshkin et al. (2021), each data set adopts a specific metric (M4: sMAPE; M3: sMAPE; TOURISM: MAPE; ELECTR: ND)

**Detailed Results** Here, we list detailed performance of zero-shot learning in Table 18, Table 19 and Table 20. For each dataset, we separately list the performance of models under diverse frequency. Compared to the most recent published method DLinear, GPT2(6) performs superior in most situations. Also, GPT2(6) does not use any information from the test data, but achieves a comparable performance of meta-leaning based N-BEATS.

Table 18: Zero-shot performance on M4 (sMAPE).

|  | Yearly (23k) | Quarterly (24k) | Monthly (48k) | Others (5k) | Average (100k) |
|---|---|---|---|---|---|
| N-BEATS-FR | 13.267 | 9.596 | 12.676 | 4.696 | 11.675 |
| DLinear-M3 | 14.193 | 18.856 | 14.765 | 9.194 | 15.337 |
| TimesNet-M3 | 15.655 | 11.877 | 16.165 | 6.863 | 14.553 |
| PatchTST-M3 | 13.966 | 10.929 | 14.664 | 7.087 | 13.228 |
| ETSformer-M3 | 27.846 | 36.134 | 25.114 | 12.338 | 27.748 |
| LightTS-M3 | 13.787 | 11.289 | 15.181 | 9.117 | 13.623 |
| Stationary-M3 | 14.988 | 11.686 | 16.098 | 6.977 | 14.327 |
| FEDformer-M3 | 13.887 | 11.513 | 18.154 | 7.529 | 15.047 |
| Autoformer-M3 | 14.552 | 17.341 | 25.063 | 9.666 | 20.022 |
| Informer-M3 | 18.542 | 16.907 | 23.454 | 7.348 | 19.047 |
| Reformer-M3 | 15.652 | 11.051 | 15.604 | 7.001 | 14.092 |
| GPT(6)-M3 | 13.740 | 10.787 | 14.630 | 7.081 | 13.125 |

Table 19: Zero-shot performance on M3 (sMAPE).

|  | Yearly (645) | Quarterly (756) | Monthly (1428) | Others (174) | Average (3003) |
|---|---|---|---|---|---|
| N-BEATS-M4 | 15.07 | 9.07 | 13.19 | 4.29 | 12.38 |
| N-BEATS-FR | 16.43 | 9.05 | 13.30 | 4.51 | 12.61 |
| DLinear-M4 | 17.43 | 9.74 | 15.65 | 6.81 | 14.03 |
| TimesNet-M4 | 18.75 | 12.26 | 14.01 | 6.88 | 14.17 |
| PatchTST-M4 | 15.99 | 9.62 | 14.71 | 9.44 | 13.39 |
| ETSformer-M4 | 20.56 | 11.65 | 16.97 | 10.57 | 16.03 |
| LightTS-M4 | 15.63 | 9.40 | 24.60 | 8.28 | 17.90 |
| Stationary-M4 | 17.05 | 12.56 | 16.82 | 8.13 | 15.29 |
| FEDformer-M4 | 16.00 | 9.48 | 15.12 | 8.94 | 13.53 |
| Autoformer-M4 | 16.18 | 13.92 | 16.91 | 14.68 | 15.87 |
| Informer-M4 | 19.70 | 13.00 | 15.91 | 13.03 | 15.82 |
| Reformer-M4 | 16.03 | 9.76 | 14.80 | 7.53 | 13.37 |
| GPT2(6)-M4 | 16.42 | 10.13 | 14.10 | 4.81 | 13.06 |

# E   Proof

In our numerical experiments, we obtain two interesting observations. First, the token similarity within a sample is larger in pretrained LM. We report the layer-wise average token cosine similarity in ETTh2 experiment in Figure 7. In particular, Figure 7 (a) shows that in a fine-tuned random initialed GPT2(6) model, the token similarity is around 0.1-0.2 among different layers. When switching to the frozen pre-trained GPT2-FPT model, the token similarity significantly increases in the deep layers and eventually reaches more than 0.9 in the last layer. The ETTh2 dataset contains high volatility hourly information related to the electricity transformer temperature. In this situation, higher token similarity implies the high-frequency noise in the data is eased and only low-frequency information will be reserved. In other words, after going through the pretrained GPT2-FPT model, the signal-noise ratio is enhanced. We use the following theorem to characterize this behavior.

Table 20: Zero-shot performance on Tourism (MAPE).

|  | Yearly (518) | Quarterly (427) | Monthly (366) | Average (1311) |
|---|---|---|---|---|
| N-BEATS-M4 | 23.57 | 14.66 | 19.32 | 18.82 |
| N-BEATS-FR | 23.43 | 14.45 | 20.47 | 19.46 |
| DLinear-M4 | 39.59 | 18.30 | 24.76 | 28.51 |
| TimesNet-M4 | 35.59 | 19.22 | 30.54 | 28.84 |
| PatchTST-M4 | 33.23 | 19.27 | 27.57 | 27.10 |
| ETSformer-M4 | 391.60 | 35.56 | 50.47 | 180.40 |
| LightTS-M4 | 138.22 | 16.28 | 25.34 | 66.99 |
| Stationary-M4 | 35.42 | 35.15 | 65.58 | 43.75 |
| FEDformer-M4 | 43.41 | 19.88 | 28.39 | 31.55 |
| Autoformer-M4 | 51.19 | 34.95 | 31.47 | 40.39 |
| Informer-M4 | 41.16 | 30.98 | 33.92 | 35.82 |
| Reformer-M4 | 33.86 | 16.85 | 23.71 | 25.48 |
| GPT2(6)-M4 | 27.17 | 16.21 | 21.92 | 22.14 |

## E.1 Theorem E.1

**Theorem E.1** (informal). *We consider the self-attention for $l$-th query token. Let's assume the input token $x_i$ are bounded with mean $\mu$ for $i = 1, 2, ..., n$. Under mild conditions, with high probability, the output value token $V_l$ converges to $\mu W_v$ on the order of $\mathcal{O}(n^{-1/2})$, where $W_v$ is the parameter matrix to compute the value token.*

The Theorem E.1 describes the self-attention structure can efficiently make output value token $V_l$ converge its mean value $\mu W_v$. In the time series forecasting task, each token represents several adjacent points in a time series. When the time series has some periodical or translation invariant structures, by comparing a given token with other tokens, one could have a higher chance to figure out those invariant structures. This phenomenon is especially important in few-shot forecasting tasks. Without enough token noise distillation ability, the model will more likely tend to overfit due to insufficient training data.

We denote $x_i$ as $i$-th element of vector $x$, $W_{ij}$ as the element at $i$-th row and $j$-th column of matrix $W$, and $W_j$ as the $j$-th row of matrix $W$. Moreover, we denote $x_i$ as the $i$-th patch (token) of the inputs with $x_i = X_i$.

Before given the formal statement of the Theorem E.1, we first show the assumptions.

1. The token $x_i$ is the sub-gaussian random vector with mean $\mu_i$ and variance $(\sigma^2/d)I$ for $i = 1, 2, ..., n$.

2. $\mu$ follows a discrete distribution with finite values $\mu \in \mathcal{V}$. Moreover, there exist $0 < \nu_1, 0 < \nu_2 < \nu_4$ such that a) $\|\mu_i\| = \nu_1$, and b) $\mu_i W_Q W_K^T \mu_i \in [\nu_2, \nu_4]$ for all $i$ and $|\mu_i W_Q W_K^\top \mu_j^\top| \leq \nu_2$ for all $\mu_i \neq \mu_j \in \mathcal{V}$.

3. $W_V$ and $W_Q W_K^\top$ are element-wise bounded with $\nu_5$ and $\nu_6$ respectively, that is, $|W_V^{(ij)}| \leq \nu_5$ and $|(W_Q W_K^\top)^{(ij)}| \leq \nu_6$, for all $i, j$ from 1 to $d$.

In the above assumptions, we ensure that for a given query patch, the difference between the clustering center and noises are large enough to be distinguished.

**Theorem E.2** (formal statement of Theorem E.1). *Let patch $x_i$ be $\sigma^2$-subgaussian random variable with mean $\mu_i$ and all $n$ patches follow the same clustering center of query $l$. Per Assumptions aforementioned, when $\sqrt{d} \geq 3(\psi(\delta, d) + \nu_2 + \nu_4)$, then with probability $1 - 5\delta$, we have*

$$\left\| \frac{\sum_{i=1}^{n} \exp\left(\frac{1}{\sqrt{d}} \boldsymbol{x}_l \boldsymbol{W_Q} \boldsymbol{W}_k^\top \boldsymbol{x}_i\right) \boldsymbol{x}_i \boldsymbol{W_V}}{\sum_{j=1}^{n} \exp\left(\frac{1}{\sqrt{d}} \boldsymbol{x}_l \boldsymbol{W_Q} \boldsymbol{W}_K^\top \boldsymbol{x}_j\right)} - \boldsymbol{\mu}_l \boldsymbol{W_V} \right\|_\infty \le 4 \exp\left(\frac{\psi(\delta, d)}{\sqrt{d}}\right) \sigma \nu_5 \sqrt{\frac{2}{dn} \log\left(\frac{2d}{\delta}\right)}$$

$$+ 7 \left[\exp\left(\frac{\nu_2 - \nu_4 + \psi(\delta, d)}{\sqrt{d}}\right) - 1\right] \|\boldsymbol{\mu}_l \boldsymbol{W_V}\|_\infty,$$

*where $\psi(\delta, d) = 2\sigma\nu_1\nu_6\sqrt{2\log\left(\frac{1}{\delta}\right)} + 2\sigma^2\nu_6 \log\left(\frac{d}{\delta}\right)$.*

*Proof.* See the proof of Lemma 2 in Wang et al. (2022) with $k_1 = k = n$. ∎

## E.2 Theorem E.4

We first give the formal statement of Theorem E.4.

**Theorem E.3** (formal statement of Theorem E.4). *Let $\boldsymbol{g}_i \in \mathbb{R}^d$ and $\boldsymbol{y}_i \in \mathbb{R}^T$ be the feature map vector and forecasting targets for the sample $i = 1, 2, ..., N$ respectively, and we assume $\frac{1}{N}\sum_{i=1}^{N} \boldsymbol{g}_i \boldsymbol{g}_i^\top \succeq \sigma I$ for some $\sigma > 0$. We want to learn a matrix $\boldsymbol{W} \in \mathbb{R}^{d \times T}$ from the following optimization problem:*

$$\boldsymbol{W} = \arg\min \frac{1}{2N} \sum_{i=1}^{N} \|\boldsymbol{W}\boldsymbol{g}_i - \boldsymbol{y}_i\|_2^2. \tag{1}$$

*If we apply stochastic gradient descent with diminishing step sizes $\eta_t = \frac{1}{\sigma t}$ at step $t$, we will need $t = \tilde{\mathcal{O}}(\epsilon^{-1}\sigma^{-1})$ steps to reach*

$$\frac{1}{t} \sum_{j=1}^{t} \left(\frac{1}{2N} \sum_{i=1}^{N} \|\boldsymbol{W}_j \boldsymbol{g}_i - \boldsymbol{y}_i\|_2^2\right) - \frac{1}{2N} \sum_{i=1}^{N} \|\boldsymbol{W}^* \boldsymbol{g}_i - \boldsymbol{y}_i\|_2^2 \le \epsilon, \tag{2}$$

*where $\boldsymbol{W}^*$ is the optimal solution and $\boldsymbol{W}_j$ is the $j$ step's solution and $\tilde{\mathcal{O}}$ we suppress the logarithmic dependence.*

*Proof.* As we assume $\frac{1}{N}\sum_{i=1}^{T} \boldsymbol{g}_i \boldsymbol{g}_i^\top \succeq \sigma I$, the hessian of optimization problem in (1) is also positive definite, which is equivalent to the optimization problem in (1) is strongly convex with parameter proportional to $\sigma$. Then via standard stochastic gradient decent analysis (e.g., section 3.1 in Lacoste-Julien et al. (2012)), we obtain:

$$\frac{1}{t} \sum_{j=1}^{t} \left(\frac{1}{2N} \sum_{i=1}^{N} \|\boldsymbol{W}_j \boldsymbol{g}_i - \boldsymbol{y}_i\|_2^2\right) - \frac{1}{2N} \sum_{i=1}^{N} \|\boldsymbol{W}^* \boldsymbol{g}_i - \boldsymbol{y}_i\|_2^2 \le \mathcal{O}\left(\frac{\log t}{\sigma t}\right) = \tilde{O}(\sigma^{-1} t^{-1}). \tag{3}$$

Therefore, to reach $\epsilon$ optimization gap, we just need to set $t = \tilde{\mathcal{O}}(\sigma^{-1}\epsilon^{-1})$. ∎

The second observation is that for the pretrained GPT2-FPT model, the last transformer layer's outputs, i.e., feature maps, are spread widely throughout the feature space. We report the t-SNE visualization of the feature maps for GPT2-FPT and an end-to-end model PatchTST in Figure 8. In Figure 8 (a) and (b), we color the samples chunked from the one single time series into the same color and the same configuration of the T-SNE is applied. One may observe that the feature maps of GPT2-FPT has less concentration compared to PatchTST. It implies the GPT2-FPT's feature maps corresponding to different samples are more distinctive which eventually facilitates the learning ability of the last MLP layer. Researchers Wang & Isola (2020) have found that contrastive learning-based representation learning may result in a uniform distribution of training data, and such behavior plays an important role in its good downstream task performance. We use the following theorem to justify it.

**Theorem E.4** (informal). *Let $\boldsymbol{g}_i$ and $\boldsymbol{y}_i$ be the feature map vector and forecasting targets for the sample $i = 1, 2, ..., N$ respectively, and we assume $\frac{1}{N}\sum_{i=1}^{N} \boldsymbol{g}_i \boldsymbol{g}_i^\top \succeq \sigma I$ for some $\sigma > 0$. Under mild conditions, if we train an MLP layer that maps feature maps to forecasting targets via the stochastic gradient descent, the total step to reach some optimization tolerance is on the order of $\mathcal{O}(\sigma^{-1})$.*

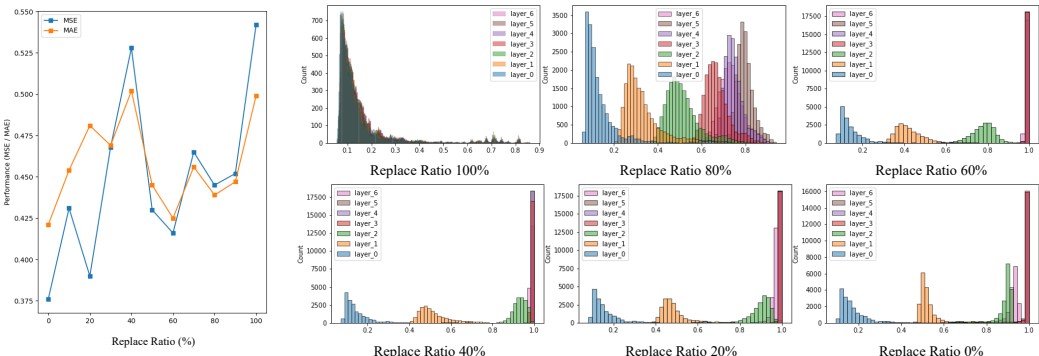

Figure 6: The performance and token similarity within samples with respect to each layer with different random replace ratios. Pretrained parameters are replaced by random initial parameters according to certain proportions.

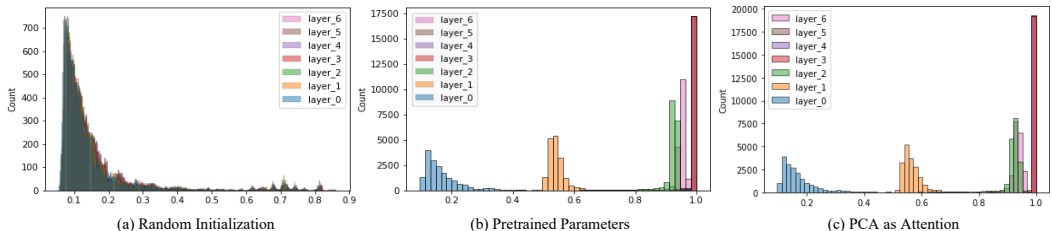

Figure 7: The token similarity within samples with respect to each layer. (a) GPT2-noPretrain-model; (b) GPT2-Pretrained-model; (c) Pretrained attention is replaced by PCA.

The Theorem E.4 considers the covariate matrix of feature maps being positive definite that indicates the set of all feature maps $\{g_i\}$ spans the whole feature spaces, and the higher spread level gives a larger $\sigma$. In this case, if we only want to learn an MLP layer, the problem reduces to a well-conditioned least-squared regression problem. Then the fast convergence rate is achieved.

Efficiently learning the last MLP layer plays a very important role in time series forecasting and can substantially impact the prediction performance. In Zeng et al. (2023), the authors show that learning a single MLP layer can also bring very promising performance. In few-shot forecasting, the pre-trained GPT2 model may still preserve highly diverse feature maps than end-to-end type models and eventually leads to fast learning speed on the last MLP layer.

Another possible benefit of wide spared feature maps is enhancing the model memorization ability when using a multi-layer decoder structure. In the literature on network memorization ability (e.g., Vardi et al. (2021); Yun et al. (2020)), the deep learning model tends to have better memorization ability when feature maps are well separated. In forecasting tasks, capturing extreme or rare behavior is very important. The pretrained GPT gains more capacity in the decoder to correctly forecast uncommon time series.

## F   N-gram Explanation for Universality

Why does the proposed pretrained-frozen-model work so effectively? We have achieved state-of-the-art performance in time series analysis using a language model that is mostly trained on natural language data. The answer lies in the universality of the frozen structure, which includes attention layers and Feed Forward layers. We can represent images and time series forecasting tasks as an n-gram estimation problem, akin to text analysis, by employing a patching approach. This method treats subsequences of time series or image patches as individual tokens. Central to sequential prediction is the $n$-order Markov process, and a simple way to capture the $n$-order Markov process is $n$-gram language model. To predict next token $w_0$, we need to compute $p(w_0|w_1, \ldots, w_{n-1})$, which can be further computed as $p(w_0 w_1 \ldots w_{n-1})/p(w_1 \ldots w_{n-1})$. Hence, the core of $n$-gram

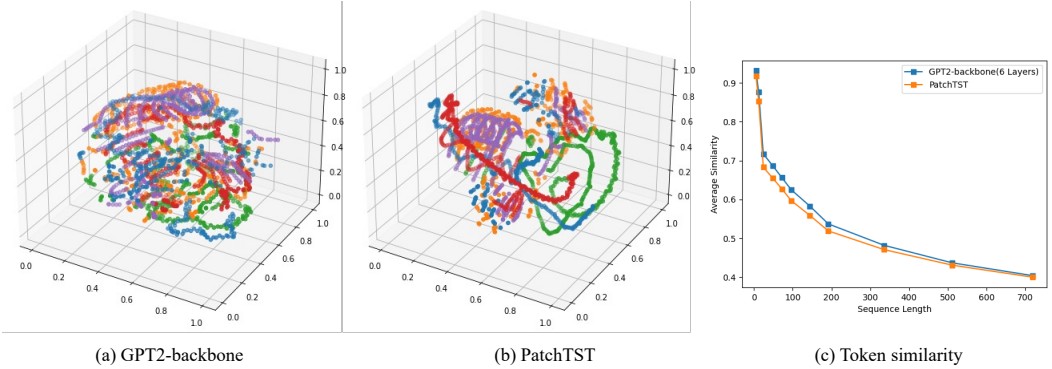

| (a) GPT2-backbone | (b) PatchTST | (c) Token similarity |

Figure 8: The t-SNE visualization of sample feature maps for (a) GPT-backbone, (b) end-to-end-PatchTST-model. (c) The token similarity within samples within different continuous sequence lengths.

language model is to estimate the probability of observing a sequence of $n$ tokens. When $n$ is large, most of $n$ token sequences will not be observed from data, leading to the sparse data problem, a common challenge faced by $n$-gram language model. As a result, a large body of research in $n$-gram language model is focused on how to effectively estimate probability of having $n$-token sequences even when they are NOT observed from data. We hypothesize that the transformer model pretrained by GPT-2 essentially allows us to estimate $p(w_0 w_1 \ldots w_{n-1})$ from observations of significantly shorter token sequences. In this section, we will show that the function of estimating probabilities of longer sequences from observation of shorter sequences is universal and is independent from domain as long as data exhibit a skew distribution (e.g., follows a power law). We note that our work is closely related to the discussion presented in Elhage et al. (2021); Olsson et al. (2022), where the authors also connect the function of transformer to compute of $n$-grams. We however note that our key result is to show the universality in computing probability of longer sequences from observations of shorter sequences, which can't be found in any existing studies. Although the discussion is restricted to discrete tokens, it should be generalized to continuous signals as we can always quantize continuous signals into a finite number of discrete tokens, similar to what BEiT Bao et al. (2022) did.

To gain a better understanding, let's start by examining a "zero-layer" Transformer model. This model operates by taking a token, embedding it, and transforming it back to produce logits that predict the subsequent token. Because it cannot transfer information from other tokens, it relies solely on the current token to predict the next one. Consequently, the optimal behavior of this model is to closely resemble the **bigram** log-likelihood.

Then we move on to the so-called "attention-only" transformer, which doesn't have MLP layers. As discussed in a recent work Elhage et al. (2021), one-layer attention-only Transformers can be comprehended as a combination of a **bigram** model and multiple **"skip-trigram"** models (impacting the probabilities of sequences "A... BC"). This can be intuitively understood as each attention head having the ability to selectively attend from the current token ("B") to a previous token ("A") and transfer relevant information to fine-tune the probability of potential subsequent tokens ("C"). Olsson et al. (2022) further discusses a multi-layer transformer can do more complex n-gram estimation using an induction heads mechanism. To be more precise, induction heads employ a straightforward principle: the '[A][B] ... [A] → [B]' rule, which elevates the likelihood of generating the subsequent token 'B' given the current token 'A' if there is a fuzzy match of the AB bigram in the historical context. This rule seems to largely decouple A and B, which means they do not memorize a fixed table of n-gram statistics. The rule [A][B] ... [A] → [B] applies regardless of what A and B are, which can abstract to new patterns.

Building upon these discussions, we are now prepared to substantiate the following argument: **For sequential data following a power law, there is a potentially universal solution to the final estimation of n-gram probabilities**. That's the reason behind the universality of pretrained LM's performance in cross-domain tasks. For simplicity, we assume that $n$ is so large that we are unable to observe any occurrence of $n$-gram from data, and we only observe the occurrence of $n'$-grams with $n' < n$. We denote by $s_i^n$ the $i$th unique $n$-gram, and by the notation $s_j^{n'} \in s_i^n$ if $n'$-gram $s_j^{n'}$ appears

in $s_i^n$, the $i$th $n$-gram. Let $m_n$ be the number of unique $n$-grams. According to the maximum entropy model, our estimation of n-gram probabilities can be cast into the following optimization problem:

$$\min \ \sum_{i=1}^{m_n} p(s_i^n) \log p(s_i^n) \quad \text{s. t.} \sum_{i:s_j^{n'} \in s_i^n} p(s_i^n) = \widehat{p}(s_j^{n'}) \quad \text{where } \widehat{p}(s_j^{n'}) \text{ represents the proba-}$$

bility of observing pattern $s_j^{n'}$ from the data and $j \in [m_{n'}], n' \in [n-1]$.

For each constraint for $\widehat{p}(s_j^{n'})$, we introduce a Lagrangian dual variable $\lambda_j^{n'}$, and rewrite the optimization problem as follows:

$$\min_\lambda \ \log \left( \sum_{i=1}^{m_n} \exp \left( \sum_{(n',j):s_j^{n'} \in s_i^n} \lambda_j^{n'} \right) \right) - \sum_{n'=1}^{n-1} \sum_{j=1}^{m_{n'}} \lambda_j^{n'} \widehat{p}(s_j^{n'}),$$

where n-gram probability $p(s_j^n)$ is given as $p(s_j^n) = \frac{1}{Z(\lambda)} \exp \left( \sum_{(n',j):s_j^{n'} \in s_i^n} \lambda_j^{n'} \right)$ and $Z(\lambda) = \sum_{i=1}^{m_n} \exp(\sum_{(n',j):s_j^{n'} \in s_i^n} \lambda_j^{n'})$

In the case that all n-grams follow a power law, for each $n' \in [n-1]$, we divide $n'$-gram into two groups: the group $\mathcal{V}_{n'}$ includes the high frequency $n'$-gram and the group $\mathcal{U}_{n'}$ including the low frequency of $n'$-gram. For simplicity, we assume that the probability for all the high frequency $n'$-grams are roughly $\alpha_{n'} \in [0,1]$ and the probability for all the low frequency $n'$-grams are roughly $\beta_{n'} \in [0,1]$. By assuming that all the patterns in $\mathcal{V}_{n'}$ and $\mathcal{U}_{n'}$ share similar appearance frequency, we simplify the optimization problem by only introducing two dual variables for each $n'$-gram, i.e. $\lambda_a^{n'}$ for high-frequency patterns and $\lambda_b^{n'}$ for low-frequency patterns as follow Using these notations, we have the optimization problem simplified as

$$\min_\lambda \quad \log(\sum_{i=1}^{m_n} \exp(\sum_{n'=1}^{n-1} \sum_{j:s_j^{n'} \in s_i^n} \lambda_a^{n'} I(s_j^{n'} \in \mathcal{V}_{n'})$$
$$+ \lambda_b^{n'} I(s_j^{n'} \in \mathcal{U}_{n'}))) - \sum_{n'=1}^{n-1} \left( \lambda_a^{n'} g_{n'} + \lambda_b^{n'} h_{n'} \right),$$

where $g_{n'} = \sum_{s_j^{n'} \in \mathcal{V}_{n'}} \widehat{p}(s_j^{n'})$ and $\quad h_{n'} = \sum_{s_j^{n'} \in \mathcal{U}_{n'}} \widehat{p}(s_j^{n'})$.

Furthermore, let $q_a^{n'}$ be the probability to observe a high frequency $n'$-gram appearing in any $n$-gram, and $q_b^{n'}$ be the probability to observe a low frequency $n'$-gram appearing in any $n$-gram, we have

$$\sum_{i=1}^{m_n} \exp(\sum_{n'=1}^{n-1} \sum_{j:s_j^{n'} \in s_i^n} \lambda_a^{n'} I(s_j^{n'} \in \mathcal{V}_{n'}) + \lambda_b^{n'} I(s_j^{n'} \in \mathcal{U}_{n'}))$$
$$= m_n \prod_{n'=1}^{n-1}(1 + q_a^{n'} \exp(\lambda_a^{n'}))(1 + q_b^{n'} \exp(\lambda_b^{n'})) + \mathcal{O}\left(\sqrt{m_n}\right).$$

By skipping the term $\mathcal{O}(\sqrt{m_n})$, we further simplify the optimization problem as

$$\min_\lambda \quad \sum_{n'=1}^{n-1} \log \left( 1 + q_a^{n'} \exp(\lambda_a^{n'}) \right) - + \quad \sum_{n'=1}^{n-1} \log \left( 1 + q_b^{n'} \exp(\lambda_b^{n'}) - \lambda_b^{n'} h_{n'}, \right.$$

which is equivalent to

$$\begin{aligned} \lambda_{n'}^a &= \min_\lambda \log \left( 1 + q_a^{n'} \exp(\lambda) \right) - \lambda g_n' \\ \lambda_{n'}^b &= \min_\lambda \log \left( 1 + q_b^{n'} \exp(\lambda) \right) - \lambda h_{n'}. \end{aligned} \quad \text{As illustrated by the above analysis, dual variables}$$

$\lambda_a^{n'}$ and $\lambda_b^{n'}$ will only depend on statistics $q_a^{n'}$, $q_b^{n'}$, $g_{n'}$ and $h_{n'}$. They are independent from the detailed statistics $\widehat{p}(s_j^{n'})$ and how each $n'$-gram appears in different $n$-gram. Thus, this simple analysis does indicate, to some degree, that the solution obtained from the maximum entropy model can be universal, as long as $n$-grams follow skewed distributions like power law.

We informally demonstrate that transformer models utilize attention mechanisms to perform a sophisticated form of n-gram estimation, and the generation rule for such n-gram distributions could be universal. This is how universality is achieved in our proposed cross-domain knowledge transfer. However, we currently lack a concrete metric to evaluate the performance of knowledge transfer between different domains, which requires further investigation. Nonetheless, in our experimental study, we demonstrate that a transformer model (beit) Bao et al. (2022) trained on images can perform well on cross-domain time series forecasting tasks.

# G   Connection between self-attention and Principle component analysis

**Understand the Gradient Structure of Self-Attention**

Let $X = (x_1, \ldots, x_N)^\top \in \mathbb{R}^{N \times D}$ be the input pattern, and let $f(X) = (f_1(X), \ldots, f_N(x))^\top : \mathbb{R}^{N \times D} \mapsto \mathbb{R}^{N \times D}$ be the function for self-attention, i.e.

$$f_i(X) = \text{softmax}(XAX^\top)X$$

where $A = W_Q W_K^\top \in \mathbb{R}^{D \times D}$. Let the Jacobian $J = \left[\frac{\partial f_i(X)}{\partial x_j}\right]_{i,j=1}^N$ represent the gradient $f(X)$ with respect to input pattern. The lemma below shows an important structure of $J$.

*Lemma* G.1.   $|J|_2 \le |A|_2 \sum_{i=1}^N \left(P_{i,i} + \frac{1}{2}\right) \left|x_i - \sum_{j=1}^N P_{i,j}x_j\right|^2 + \Delta$

where   $\Delta = |A|_2 \sum_{i \ne j}^N P_{i,j} \left|x_j - \sum_{k=1}^N P_{i,k}x_k\right|^2 + \frac{|A|_2}{2}\sum_{j=1}^N |x_i|^2$   and   $P_{i,j} = \frac{\exp(x_i^\top A x_j)}{\sum_{k=1}^N \exp(x_i^\top A x_k)}$

*Proof.* According to the analysis from the work, we have the gradient $J_{i,j} = \frac{\partial f_i(X)}{x_j}$ is given by

$$J_{i,j} = P_{i,j}I + X^\top Q^i \left(XA\delta_{i,j} + E_{j,i}XA^\top\right) \quad \text{where} \quad Q^i = \text{diag}(P_{i,:}) - P_{i,:}P_{i,:}^\top \quad \text{Here } P_{i,:} \in \mathbb{R}_+^N$$

represents the $i$-th row of matrix $P$. We thus have

$$
\begin{aligned}
|J|_2 &\le \sum_{i,j=1}^N |J_{i,j}|_2 \\
&\le \sum_{i,j=1}^N P_{i,j} + \sum_{i=1}^N |X^\top Q^i X|_2 |A|_2 + \sum_{i,j=1}^N |X^\top Q^i E_{j,i} X|_2 |A|_2 \\
&\le N + |A|_2 \sum_{i=1}^N \left(\sum_{j=1}^N P_{i,j}|x_j|^2 - \left|\sum_{j=1}^N P_{i,j}x_j\right|^2\right) + |A|_2 \sum_{i,j=1}^N |X^\top Q^i e_j x_i^\top| \\
&\le N + |A|_2 \sum_{i=1}^N \sum_{j=1}^N P_{i,j}\left|x_j - \sum_{k=1}^N P_{i,k}x_k\right|^2 + |A|_2 \sum_{i,j=1}^N P_{i,j}\left|x_i^\top\left(x_j - X^\top P_{i,:}\right)\right| \\
&\le |A|_2 \sum_{i=1}^N \left(P_{i,i} + \frac{1}{2}\right)|x_i - X^\top P_{i,:}|^2 + \underbrace{N + |A|_2 \sum_{i \ne j}^N P_{i,j}|x_j - X^\top P_{i,:}|^2 + \frac{|A|_2}{2}\sum_{j=1}^N |x_i|^2}_{:=\Delta}
\end{aligned}
$$
∎

As indicated by Lemma 1, one of the key components in the upper bound of Jacobian is $|x_i - \sum_{j=1}^N P_{i,j}x_j|^2$. Thus, through the optimization, we like to reduce the size of the gradient and therefore may prefer to reduce the quantity to $\sum_{i=1}^N |x_i - \sum_{j=1}^N P_{i,j}x_j|^2$. Hence, it will be interesting to understand the choice of $W^Q$ and $W^K$ that leads to the minimization of $\sum_{i=1}^N |x_i - \sum_{j=1}^N P_{i,j}x_j|^2$, i.e. the following optimization problem   $\min_{|A|_F \le \rho} \sum_{i=1}^N \left|x_i - \sum_{j=1}^N P_{i,j}x_j\right|^2$   where $\rho$ is introduced to control the size of $A$.

**Connection between Self-Attention and Principal Component Analysis**

Let consider the optimization problem in (G) when $\rho$ is small, we can approximate $P_{i,j}$ as   $P_{i,j} \approx \frac{1}{N} + \frac{1}{N}x_i^\top A x_j$   Define $\bar{x} = X^\top \mathbf{1}/N$.   We have $\sum_{i=1}^N |x_i - X^\top P_{i,:}|^2 = \sum_{i=1}^N |x_i - \bar{x} - X^\top X A x_i|^2$   By assuming that all the input patterns are zero centralized, we have $\bar{x} = 0$ and   $\sum_{i=1}^N |x_i - X^\top X A x_i|^2 = \text{tr}\left((I - X^\top X A)^2 X^\top X\right)$   The theorem below shows that $A$ minimizing the objective $\sum_{i=1}^N |x_i - X^\top X A x_i|^2$ contains the largest $m$ eigenvectors of $X^\top X$ where $m$ is the rank of $A$.

*Theorem* 2. Let $W_Q$ and $W_K$ be matrices of size $D \times m$. Let $\lambda_1 \ge \lambda_2 \ge \ldots \ge \lambda_D$ be the eigenvalues of $X^\top X$ ranked in descending order, and let $v_i \in \mathbb{R}^D, i = 1, \ldots, D$ be the corresponding eigenvectors. The optimal solution $A^*$ that minimizes $\sum_{i=1}^N |x_i - X^\top X A x_i|^2$ is given by $A = \sum_{i=1}^m \frac{1}{\lambda_i} v_i v_i^\top$

*Proof.* Since $W_Q, W_K \in \mathbb{R}^{D \times m}$ where $m < D$, we know that $A$ is a matrix of rank $m$. Hence, we know   $\min_A \sum_{i=1}^N |x_i - X^\top X A x_i|^2 \ge \sum_{k=m+1}^N \lambda_k$   We also know that by choosing $A$ as $A = \sum_{i=1}^m \frac{1}{\lambda_i} v_i v_i^\top$   we have

$$\sum_{i=1}^{N} |x_i - X^\top X A x_i|^2 = \text{tr}\left(\left(I - \sum_{i=1}^{m} v_i v_i^\top\right)^2 X^\top X\right) = \sum_{k=m+1}^{D} \lambda_k$$

Hence, the solution $A$ for minimizing $\sum_{i=1}^{N} |x_i - X^\top X A x_i|^2$ is essential a weighted combination of top eigenvectors of $X^\top X$. Since a small gradient will prefer a small quantity of $\sum_{i=1}^{N} |x_i - X^\top X A x_i|^2$, by minimizing through the self-attention layer, we essentially choose weight matrix $W_Q$ and $W_K$ to be aligned with the principal directions of $X^\top X$. ∎

## H Experiment Analysis and Other Key Results

### H.1 Experiment analysis of GPT2-FPT model

In this section, we conduct experiments to analyze whether the self-attention frozen pre-trained model improves performance compared with overall fine-tuning and random initialization. Firstly, we compare GPT2(6) FPT with the same model without freezing (No Freeze) and random initial model (No Pre-train). For the end-to-end paradigm No Pre-train GPT2-backbone (6 Layers), we directly train all parameters of the model. We summarize the results in Table 21 and Table 22. Then we analyze the performance of various layers to clarify our selection of GPT2(6) FPT.

Table 21: Model analysis results on 5% data. We use prediction length $O \in \{96, 192, 336, 720\}$ for ILI and $O \in \{24, 36, 48, 60\}$ for others.

| Methods | | GPT2(6) | | No Freeze | | No Pretrain | |
|---|---|---|---|---|---|---|---|
| Metric | | MSE | MAE | MSE | MAE | MSE | MAE |
| Weather | 96 | **0.175** | 0.230 | 0.183 | **0.229** | 0.199 | 0.254 |
| | 192 | **0.227** | **0.276** | 0.275 | 0.300 | 0.262 | 0.302 |
| | 336 | **0.286** | **0.322** | 0.297 | 0.331 | 0.326 | 0.345 |
| | 720 | **0.366** | **0.379** | 0.380 | 0.388 | 0.405 | 0.396 |
| ETTh1 | 96 | **0.543** | **0.506** | 0.671 | 0.564 | 0.882 | 0.643 |
| | 192 | **0.748** | **0.580** | 0.907 | 0.632 | 1.389 | 0.817 |
| | 336 | **0.754** | **0.595** | 0.931 | 0.655 | 2.968 | 1.149 |
| | 720 | - | - | - | - | - | - |
| ETTh2 | 96 | **0.376** | **0.421** | 0.440 | 0.449 | 0.465 | 0.457 |
| | 192 | **0.418** | **0.441** | 0.503 | 0.478 | 0.614 | 0.536 |
| | 336 | **0.408** | **0.439** | 0.691 | 0.572 | 0.596 | 0.529 |
| | 720 | - | - | - | - | - | - |
| ETTm1 | 96 | **0.386** | **0.405** | 0.429 | 0.432 | 0.394 | 0.410 |
| | 192 | 0.440 | 0.438 | 0.496 | 0.470 | **0.432** | **0.432** |
| | 336 | **0.485** | **0.459** | 0.535 | 0.489 | 0.491 | 0.464 |
| | 720 | **0.557** | **0.499** | 0.786 | 0.592 | 0.564 | 0.503 |
| ETTm2 | 96 | **0.199** | **0.280** | 0.217 | 0.293 | 0.301 | 0.353 |
| | 192 | **0.256** | **0.316** | 0.300 | 0.350 | 0.321 | 0.365 |
| | 336 | **0.318** | **0.353** | 0.331 | 0.368 | 0.371 | 0.398 |
| | 720 | **0.460** | 0.439 | **0.460** | **0.436** | 0.659 | 0.528 |

**Fine-tune More Parameters** Compared with fine-tuning all parameters, self-attention frozen pre-trained model GPT2(6) FPT achieves better performance on most datasets and yields an overall **12.7%** relative MSE reduction on 5% data and **11.5%** relative MSE reduction on 10% data. It verifies that frozen pre-trained attention layers are effective for time series forecasting.

**Parameters Initialization** Compared with the random initial model, self-attention frozen pre-trained model GPT2(6) FPT achieves better performance on most datasets and yields an overall **21.2%** relative MSE reduction on 5% data and **14.3%** relative MSE reduction on 10% data. It again suggests that a model pre-trained on cross-domain data can achieve significant performance improvement in time series forecasting.

**The Number of GPT2 Layers** For most transformer-based methods in time-series forecasting Zhou et al. (2022); Wu et al. (2021); Nie et al. (2022), no more than 3 encoder layers are included. However, most pre-trained models with at least 12 layers may suffer from overfitting in time series forecasting. To better balance performance and computational efficiency, we test using various numbers of layers on ETTh2. Additionally, we train a completely random initialized non-pretrained GPT2 as a comparison. The results are shown in Figure 9, for both 5% and 10% data, the pre-trained model is unable to do well with few layers but significantly outperforms non-pre-trained GPT2 with more attention blocks transferred from NLP. It indicates that pre-trained attention layers produce

Table 22: No Pretrain and No Freeze results on 10% data. We use prediction length $O \in \{96, 192, 336, 720\}$ for ILI and $O \in \{24, 36, 48, 60\}$ for others.

| Methods | | GPT2(6) | | No Freeze | | No Pretrain | |
|---|---|---|---|---|---|---|---|
| Metric | | MSE | MAE | MSE | MAE | MSE | MAE |
| *Weather* | 96 | **0.163** | **0.215** | 0.168 | 0.221 | 0.175 | 0.229 |
| | 192 | **0.210** | **0.254** | 0.238 | 0.286 | 0.244 | 0.287 |
| | 336 | **0.256** | **0.292** | 0.289 | 0.318 | 0.301 | 0.325 |
| | 720 | **0.321** | **0.339** | 0.398 | 0.383 | 0.390 | 0.378 |
| *ETTh1* | 96 | **0.458** | **0.456** | 0.605 | 0.532 | 0.680 | 0.560 |
| | 192 | **0.570** | **0.516** | 0.713 | 0.579 | 0.738 | 0.602 |
| | 336 | **0.608** | **0.535** | 0.747 | 0.586 | 0.893 | 0.641 |
| | 720 | **0.725** | **0.591** | 0.945 | 0.688 | 2.994 | 1.169 |
| *ETTh2* | 96 | **0.331** | **0.374** | 0.369 | 0.394 | 0.422 | 0.433 |
| | 192 | **0.402** | **0.411** | 0.464 | 0.455 | 0.482 | 0.466 |
| | 336 | **0.406** | **0.433** | 0.420 | 0.439 | 0.540 | 0.496 |
| | 720 | **0.449** | **0.464** | 0.535 | 0.515 | 0.564 | 0.519 |
| *ETTm1* | 96 | 0.390 | 0.404 | 0.429 | 0.430 | **0.385** | **0.401** |
| | 192 | 0.429 | 0.423 | 0.463 | 0.446 | **0.426** | **0.421** |
| | 336 | **0.469** | **0.439** | 0.510 | 0.470 | 0.506 | 0.455 |
| | 720 | **0.569** | **0.498** | 0.780 | 0.591 | 0.576 | 0.505 |
| *ETTm2* | 96 | **0.188** | **0.269** | 0.243 | 0.311 | 0.244 | 0.315 |
| | 192 | **0.251** | **0.309** | 0.307 | 0.352 | 0.318 | 0.363 |
| | 336 | **0.307** | **0.346** | 0.337 | 0.364 | 0.409 | 0.412 |
| | 720 | **0.426** | **0.417** | 0.471 | 0.440 | 0.473 | 0.450 |

a great benefit in time series forecasting. Also, the pre-trained model achieves better performance between 3 and 9 layers. Thus GPT2 with 6 layers is chosen as our default architecture.

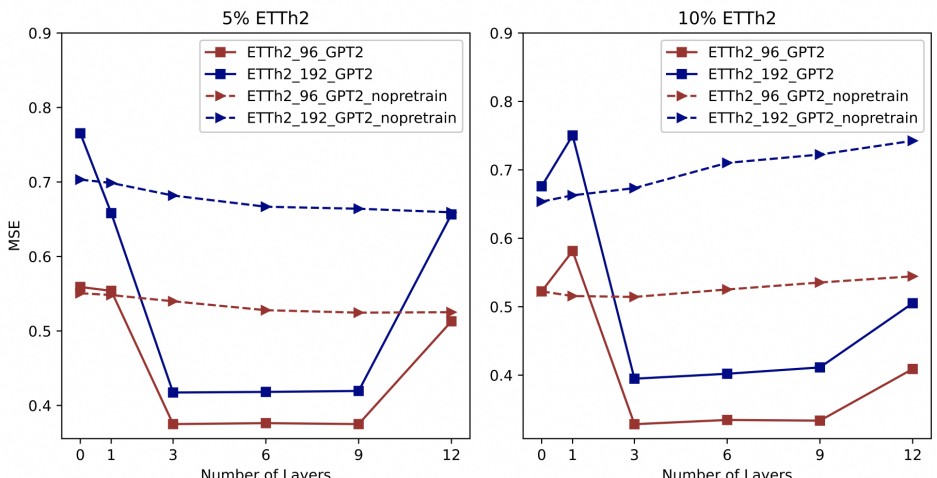

Figure 9: Comparison of pre-trained and non-pre-trained GPT2 with various layers on ETTh2. Color represents various prediction length $O \in \{96, 192\}$ and line style means different models .

## H.2    No Pre-training but Freezing

For comprehensively ablation on pre-training and freezing strategies, we also add experiment for random initialized GPT2(6) with freezing. The results in Table 23 shows that only input and output modules can not work and pre-trained knowledge play an importance part in time series tasks.

Table 23: Ablation on random initialized model with freezing.

| Methods | | GPT2(6) | | No Freeze | | No Pretrain | | No Pretrain + Freeze | |
|---|---|---|---|---|---|---|---|---|---|
| Metric | | MSE | MAE | MSE | MAE | MSE | MAE | MSE | MAE |
| *ETTh2* | 96 | 0.376 | 0.421 | 0.440 | 0.449 | 0.465 | 0.457 | 0.540 | 0.497 |
| | 192 | 0.418 | 0.441 | 0.503 | 0.478 | 0.614 | 0.536 | 0.721 | 0.580 |

### H.3 Fine-Tuning Parameters Selection

In this section, we conduct ablation experiments to study which parameters are important to fine-tune. Since the input embedding and output layers are randomly initialized for adapting to a new domain, they must be trained. Then, we study adding layer normalization and positional embeddings to the list of fine-tuning parameters. Table 24 shows the results that re-train parameters of layer normalization and positional embeddings can bring certain benefits, especially in longer prediction lengths. Thus, we follow the standard practice to re-train positional embeddings and layer normalization.

Table 24: Ablation by fixing positional embeddings or layer normalization on 5% ETTm1 and ETTm2. Parameters of GPT2(6) are successively added to the list of fine-tuned parameters.

| Methods | | Input & Output | | + LN | | + POS | |
|---|---|---|---|---|---|---|---|
| Metric | | MSE | MAE | MSE | MAE | MSE | MAE |
| ETTm1 | 96 | 0.395 | 0.410 | 0.392 | 0.409 | 0.386 | 0.405 |
| | 192 | 0.444 | 0.438 | 0.436 | 0.435 | 0.440 | 0.438 |
| | 336 | 0.510 | 0.472 | 0.495 | 0.467 | 0.485 | 0.459 |
| | 720 | 0.607 | 0.517 | 0.564 | 0.503 | 0.557 | 0.499 |
| ETTm2 | 96 | 0.198 | 0.282 | 0.198 | 0.279 | 0.199 | 0.280 |
| | 192 | 0.261 | 0.324 | 0.263 | 0.325 | 0.256 | 0.316 |
| | 336 | 0.336 | 0.377 | 0.322 | 0.356 | 0.318 | 0.353 |
| | 720 | 0.473 | 0.444 | 0.457 | 0.435 | 0.460 | 0.436 |

### H.4 Analysis of Data Volume

Results of few-shot learning show that GPT2(6) FPT shows SOTA performance in few-shot learning tasks in which the model is trained on 5% data and 10% data. Plus, it has comparable performance with the SOTA baselines PatchTST and Dlinear on full sample forecasting setting as well. This phenomenon raises a question that how performance changes with an increase in data sample size. Thus, we conduct experiments on various percentages $P \in \{5\%, 10\%, 20\%, 50\%, 80\%, 100\%\}$ of ETTh2. Figure 10 shows that the performance improvement for GPT2(6) FPT is almost flattened. These results illustrate that such a cross-domain FPT model is extremely efficient in few-shot time series forecasting and only requires a few fine-tuning samples to reach a SOTA performance. For more complete data, end-to-end training models start to catch up, but still, a GPT2(6) FPT model can be comparable to those SOTA end-to-end training algorithms.

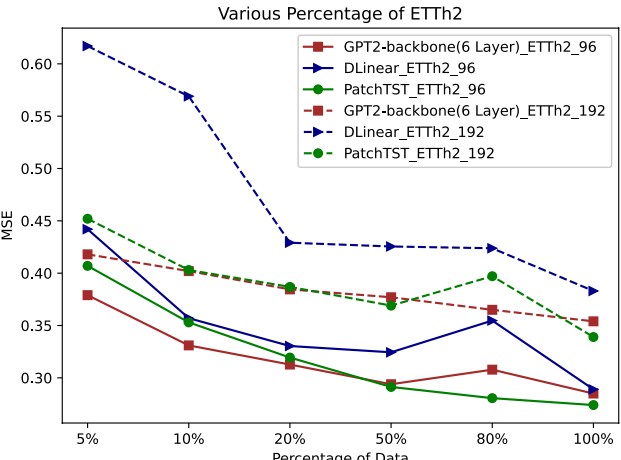

Figure 10: Results on various percentages of ETTh2. Line color represents different models and line style means various prediction lengths $O \in \{96, 192\}$.

## H.5 Knowledge transfer with other Pre-trained Transformer Models

We investigate how other pre-trained transformer models perform and whether other domains can also help. Another NLP pre-trained model BERT Devlin et al. (2019) and the CV pre-trained model BEiT Bao et al. (2022) are trained on 5% ETTh2 and 5% ETTm2. Similar to GPT2, we only reserve 6 layers and freeze attention blocks. Our results are shown in Table 25 that BERT(6) FPT and BEiT(6) FPT are comparable to PatchTST and remarkably surpass other baselines. We come to the conclusion that the universality of our proposed architecture holds across other pre-trained-transformer models. Moreover, the domain of successful knowledge transfer in time series forecasting is not limited to natural language. Knowledge from the CV domain can also help, supported by BEiT's experimental results.

Table 25: Results of frozen pretrained transformer variants on 5% ETTh2 and ETTm2. We use prediction length $O \in \{96, 192, 336, 720\}$. A lower MSE indicates better performance. **Black**: best, **Red**: second best, **Violet**: third best. '-' means that 5% time series is not sufficient to constitute a training set.

| Methods | Metric | ETTh2 | | | | ETTm2 | | | |
|---|---|---|---|---|---|---|---|---|---|
| | | 96 | 192 | 336 | 720 | 96 | 192 | 336 | 720 |
| GPT2-backbone(6 Layers) | MSE | **0.376** | **0.421** | **0.408** | - | **0.199** | **0.256** | **0.318** | 0.460 |
| | MAE | 0.419 | **0.441** | **0.439** | - | **0.280** | **0.316** | **0.353** | 0.436 |
| BERT-backbond(6 Layers) | MSE | 0.397 | 0.480 | 0.481 | - | 0.222 | 0.281 | 0.331 | 0.441 |
| | MAE | **0.418** | 0.465 | 0.472 | - | 0.300 | 0.335 | 0.367 | 0.428 |
| BEiT-backbond(6 Layers) | MSE | 0.405 | 0.448 | 0.524 | - | 0.208 | 0.272 | 0.331 | 0.452 |
| | MAE | **0.418** | 0.446 | 0.500 | - | 0.291 | 0.326 | 0.362 | 0.433 |
| DLinearZeng et al. (2023) | MSE | 0.442 | 0.617 | 1.424 | - | 0.236 | 0.306 | 0.380 | 0.674 |
| | MAE | 0.456 | 0.542 | 0.849 | - | 0.326 | 0.373 | 0.423 | 0.583 |
| PatchTSTNie et al. (2022) | MSE | 0.401 | 0.452 | 0.464 | - | 0.206 | 0.264 | 0.334 | 0.454 |
| | MAE | 0.421 | 0.455 | 0.469 | - | 0.288 | 0.324 | 0.367 | 0.483 |
| FEDformerZhou et al. (2022) | MSE | 0.390 | 0.457 | 0.477 | - | 0.299 | 0.290 | 0.378 | 0.523 |
| | MAE | 0.424 | 0.465 | 0.483 | - | 0.320 | 0.361 | 0.427 | 0.510 |
| AutoformerWu et al. (2021) | MSE | 0.428 | 0.496 | 0.486 | - | 0.232 | 0.291 | 0.478 | 0.533 |
| | MAE | 0.468 | 0.504 | 0.496 | - | 0.322 | 0.357 | 0.517 | 0.538 |

## H.6 Full Results of Classification

Table 26: Full results for the classification task. ∗. in the Transformers indicates the name of ∗former.

| Methods | Classical methods | | RNN | | TCN | Transformers | | | | | | | | | MLP | | TimesNet | GPT2(6) |
|---|---|---|---|---|---|---|---|---|---|---|---|---|---|---|---|---|---|---|
| | XGBoost | Rocket | LSTNet | LSSL | | Trans. | Re. | In. | Pyra. | Auto. | Station. | FED. | ETS. | Flow. | DLinear | LightTS. | | |
| EthanolConcentration | 43.7 | 45.2 | 39.9 | 31.1 | 28.9 | 32.7 | 31.9 | 31.6 | 30.8 | 31.6 | 32.7 | 31.2 | 28.1 | 33.8 | 32.6 | 29.7 | 35.7 | 34.2 |
| FaceDetection | 63.3 | 64.7 | 65.7 | 66.7 | 52.8 | 67.3 | 68.6 | 67.0 | 65.7 | 68.4 | 68.0 | 66.0 | 66.3 | 67.6 | 68.0 | 67.5 | 68.6 | 69.2 |
| Handwriting | 15.8 | 58.8 | 25.8 | 24.6 | 53.3 | 32.0 | 27.4 | 32.8 | 29.4 | 36.7 | 31.6 | 28.0 | 32.5 | 33.8 | 27.0 | 26.1 | 32.1 | 32.7 |
| Heartbeat | 73.2 | 75.6 | 77.1 | 72.7 | 75.6 | 76.1 | 77.1 | 80.5 | 75.6 | 74.6 | 73.7 | 73.7 | 71.2 | 77.6 | 75.1 | 75.1 | 78.0 | 77.2 |
| JapaneseVowels | 86.5 | 96.2 | 98.1 | 98.4 | 98.9 | 98.7 | 97.8 | 98.9 | 98.4 | 96.2 | 99.2 | 98.4 | 95.9 | 98.9 | 96.2 | 96.2 | 98.4 | 98.6 |
| PEMS-SF | 98.3 | 75.1 | 86.7 | 86.1 | 68.8 | 82.1 | 82.7 | 81.5 | 83.2 | 82.7 | 87.3 | 80.9 | 86.0 | 83.8 | 75.1 | 88.4 | 89.6 | 87.9 |
| SelfRegulationSCP1 | 84.6 | 90.8 | 84.0 | 90.8 | 84.6 | 92.2 | 90.4 | 90.1 | 88.1 | 84.0 | 89.4 | 88.7 | 89.6 | 92.5 | 87.3 | 89.8 | 91.8 | 93.2 |
| SelfRegulationSCP2 | 48.9 | 53.3 | 52.8 | 52.2 | 55.6 | 53.9 | 56.7 | 53.3 | 53.3 | 50.6 | 57.2 | 54.4 | 55.0 | 56.1 | 50.5 | 51.1 | 57.2 | 59.4 |
| SpokenArabicDigits | 69.6 | 71.2 | 100.0 | 100.0 | 95.6 | 98.4 | 97.0 | 100.0 | 99.6 | 100.0 | 100.0 | 100.0 | 100.0 | 98.8 | 81.4 | 100.0 | 99.0 | 99.2 |
| UWaveGestureLibrary | 75.9 | 94.4 | 87.8 | 85.9 | 88.4 | 85.6 | 85.6 | 85.6 | 83.4 | 85.9 | 87.5 | 85.3 | 85.0 | 86.6 | 82.1 | 80.3 | 85.3 | 88.1 |
| Average | 66.0 | 72.5 | 71.8 | 70.9 | 70.3 | 71.9 | 71.5 | 72.1 | 70.8 | 71.1 | 72.7 | 70.7 | 71.0 | 73.0 | 67.5 | 70.4 | **73.6** | **74.0** |

## H.7 Full Results of Anomaly Detection

## H.8 Full Results of Imputation

## H.9 Full Results of Short-term Forecasting

## H.10 Input Length Setting Discussion

The consideration of input length is of great importance. It is widely believed that longer input lengths have the potential to generate superior results. However, in practice, certain algorithms might fall short in effectively utilizing long input signals due to overfitting issues either. Here, we conducted long-term forecasting experiments by comparing our approach with the best reported values from different baseline papers. This was done to avoid any biases stemming from selective tuning of baseline parameters. While some may argue in favor of a fairer comparison using a fixed input length,

Table 27: Full results for the anomaly detection.

| Methods | SMD | | | MSL | | | SMAP | | | SWaT | | | PSM | | | Avg F1 |
| Metrics | P | R | F1 | P | R | F1 | P | R | F1 | P | R | F1 | P | R | F1 | % |
|---|---|---|---|---|---|---|---|---|---|---|---|---|---|---|---|---|
| GPT(6) | **88.89** | 84.98 | **86.89** | 82.00 | 82.91 | 82.45 | **90.60** | 60.95 | **72.88** | **92.20** | 96.34 | **94.23** | 98.62 | 95.68 | 97.13 | **86.72** |
| TimesNet[*] | 87.91 | 81.54 | 84.61 | **89.54** | 75.36 | 81.84 | 90.14 | 56.40 | 69.39 | 90.75 | 95.40 | 93.02 | 98.51 | **96.20** | **97.34** | 85.24 |
| PatchTST | 87.26 | 82.14 | 84.62 | 88.34 | 70.96 | 78.70 | 90.64 | 55.46 | 68.82 | 91.10 | 80.94 | 85.72 | 98.84 | 93.47 | 96.08 | 82.79 |
| ETSformer | 87.44 | 79.23 | 83.13 | 85.13 | 84.93 | 85.03 | 92.25 | 55.75 | 69.50 | 90.02 | 80.36 | 84.91 | 99.31 | 85.28 | 91.76 | 82.87 |
| FEDformer | 87.95 | 82.39 | 85.08 | 77.14 | 80.07 | 78.57 | 90.47 | 58.10 | 70.76 | 90.17 | 96.42 | 93.19 | 97.31 | 97.16 | 97.23 | 84.97 |
| LightTS | 87.10 | 78.42 | 82.53 | 82.40 | 75.78 | 78.95 | 92.58 | 55.27 | 69.21 | 91.98 | 94.72 | 93.33 | 98.37 | 95.97 | 97.15 | 84.23 |
| DLinear | 83.62 | 71.52 | 77.10 | 84.34 | 85.42 | 84.88 | 92.32 | 55.41 | 69.26 | 80.91 | 95.30 | 87.52 | 98.28 | 89.26 | 93.55 | 82.46 |
| Stationary | 88.33 | 81.21 | 84.62 | 68.55 | 89.14 | 77.50 | 89.37 | 59.02 | 71.09 | 68.03 | 96.75 | 79.88 | 97.82 | 96.76 | 97.29 | 82.08 |
| Autoformer | 88.06 | 82.35 | 85.11 | 77.27 | 80.92 | 79.05 | 90.40 | 58.62 | 71.12 | 89.85 | 95.81 | 92.74 | 99.08 | 88.15 | 93.29 | 84.26 |
| Pyraformer | 85.61 | 80.61 | 83.04 | 83.81 | 85.93 | 84.86 | 92.54 | 57.71 | 71.09 | 87.92 | 96.00 | 91.78 | 71.67 | 96.02 | 82.08 | 82.57 |
| Anomaly Transformer[**] | 88.91 | 82.23 | 85.49 | 79.61 | 87.37 | 83.31 | 91.85 | 58.11 | 71.18 | 72.51 | 97.32 | 83.10 | 68.35 | 94.72 | 79.40 | 80.50 |
| Informer | 86.60 | 77.23 | 81.65 | 81.77 | 86.48 | 84.06 | 90.11 | 57.13 | 69.92 | 70.29 | 96.75 | 81.43 | 64.27 | 96.33 | 77.10 | 78.83 |
| Reformer | 82.58 | 69.24 | 75.32 | 85.51 | 83.31 | 84.40 | 90.91 | 57.44 | 70.40 | 72.50 | 96.53 | 82.80 | 59.93 | 95.38 | 73.61 | 77.31 |
| LogTransformer | 83.46 | 70.13 | 76.21 | 73.05 | 87.37 | 79.57 | 89.15 | 57.59 | 69.97 | 68.67 | 97.32 | 80.52 | 63.06 | 98.00 | 76.74 | 76.60 |
| Transformer | 83.58 | 76.13 | 79.56 | 71.57 | 87.37 | 78.68 | 89.37 | 57.12 | 69.70 | 68.84 | 96.53 | 80.37 | 62.75 | 96.56 | 76.07 | 76.88 |

[*] We reproduce the results of TimesNet by https://github.com/thuml/Time-Series-Library.
[**] We replace the joint criterion in Anomaly Transformer with reconstruction error for fair comparison.

Table 28: Full results for the imputation task.

| Methods | | GPT2(3) | | TimesNet | | PatchTST | | ETSformer | | LightTS | | DLinear | | FEDformer | | Stationary | | Autoformer | | Informer | | Reformer | |
| Mask | Ratio | MSE | MAE | MSE | MAE | MSE | MAE | MSE | MAE | MSE | MAE | MSE | MAE | MSE | MAE | MSE | MAE | MSE | MAE | MSE | MAE | MSE | MAE |
|---|---|---|---|---|---|---|---|---|---|---|---|---|---|---|---|---|---|---|---|---|---|---|---|
| ETTm1 | 12.5% | **0.017** | **0.085** | 0.023 | 0.101 | 0.041 | 0.130 | 0.096 | 0.229 | 0.093 | 0.206 | 0.080 | 0.193 | 0.052 | 0.166 | 0.032 | 0.119 | 0.046 | 0.144 | 0.063 | 0.180 | 0.042 | 0.146 |
| | 25% | **0.022** | **0.096** | 0.023 | 0.101 | 0.044 | 0.135 | 0.096 | 0.229 | 0.093 | 0.206 | 0.080 | 0.193 | 0.052 | 0.166 | 0.032 | 0.119 | 0.046 | 0.144 | 0.063 | 0.180 | 0.042 | 0.146 |
| | 37.5% | **0.029** | **0.111** | 0.029 | 0.111 | 0.049 | 0.143 | 0.133 | 0.271 | 0.113 | 0.231 | 0.103 | 0.219 | 0.069 | 0.191 | 0.039 | 0.131 | 0.057 | 0.161 | 0.079 | 0.200 | 0.063 | 0.182 |
| | 50% | 0.040 | 0.128 | **0.036** | **0.124** | 0.055 | 0.151 | 0.186 | 0.323 | 0.134 | 0.255 | 0.132 | 0.248 | 0.089 | 0.218 | 0.047 | 0.145 | 0.067 | 0.174 | 0.093 | 0.218 | 0.082 | 0.208 |
| | Avg | 0.028 | **0.105** | **0.027** | 0.107 | 0.047 | 0.140 | 0.120 | 0.253 | 0.104 | 0.218 | 0.093 | 0.206 | 0.062 | 0.177 | 0.036 | 0.126 | 0.051 | 0.150 | 0.071 | 0.188 | 0.055 | 0.166 |
| ETTm2 | 12.5% | **0.017** | **0.076** | 0.018 | 0.080 | 0.026 | 0.094 | 0.108 | 0.239 | 0.034 | 0.127 | 0.062 | 0.166 | 0.056 | 0.159 | 0.021 | 0.088 | 0.023 | 0.092 | 0.133 | 0.270 | 0.108 | 0.228 |
| | 25% | **0.020** | **0.080** | 0.020 | 0.085 | 0.028 | 0.099 | 0.164 | 0.294 | 0.042 | 0.143 | 0.085 | 0.196 | 0.080 | 0.195 | 0.024 | 0.096 | 0.026 | 0.101 | 0.135 | 0.272 | 0.136 | 0.262 |
| | 37.5% | **0.022** | **0.087** | 0.023 | 0.091 | 0.030 | 0.104 | 0.237 | 0.356 | 0.051 | 0.159 | 0.106 | 0.222 | 0.110 | 0.231 | 0.027 | 0.103 | 0.030 | 0.108 | 0.155 | 0.293 | 0.175 | 0.300 |
| | 50% | **0.025** | **0.095** | 0.026 | 0.098 | 0.034 | 0.110 | 0.323 | 0.421 | 0.059 | 0.174 | 0.131 | 0.247 | 0.156 | 0.276 | 0.030 | 0.108 | 0.035 | 0.119 | 0.200 | 0.333 | 0.211 | 0.329 |
| | Avg | **0.021** | **0.084** | 0.022 | 0.088 | 0.029 | 0.102 | 0.208 | 0.327 | 0.046 | 0.151 | 0.096 | 0.208 | 0.101 | 0.215 | 0.026 | 0.099 | 0.029 | 0.105 | 0.156 | 0.292 | 0.157 | 0.280 |
| ETTh1 | 12.5% | **0.043** | **0.140** | 0.057 | 0.159 | 0.093 | 0.201 | 0.126 | 0.263 | 0.240 | 0.345 | 0.151 | 0.267 | 0.070 | 0.190 | 0.060 | 0.165 | 0.074 | 0.182 | 0.114 | 0.234 | 0.074 | 0.194 |
| | 25% | **0.054** | **0.156** | 0.069 | 0.178 | 0.107 | 0.217 | 0.169 | 0.304 | 0.265 | 0.364 | 0.180 | 0.292 | 0.106 | 0.236 | 0.080 | 0.189 | 0.090 | 0.203 | 0.140 | 0.262 | 0.102 | 0.227 |
| | 37.5% | **0.072** | **0.180** | 0.084 | 0.196 | 0.120 | 0.230 | 0.220 | 0.347 | 0.296 | 0.382 | 0.215 | 0.318 | 0.124 | 0.258 | 0.102 | 0.212 | 0.109 | 0.222 | 0.174 | 0.293 | 0.135 | 0.261 |
| | 50% | 0.107 | 0.216 | **0.102** | **0.215** | 0.141 | 0.248 | 0.293 | 0.402 | 0.334 | 0.404 | 0.257 | 0.347 | 0.165 | 0.299 | 0.133 | 0.240 | 0.137 | 0.248 | 0.215 | 0.325 | 0.179 | 0.298 |
| | Avg | **0.069** | **0.173** | 0.078 | 0.187 | 0.115 | 0.224 | 0.202 | 0.329 | 0.284 | 0.373 | 0.201 | 0.306 | 0.117 | 0.246 | 0.094 | 0.201 | 0.103 | 0.214 | 0.161 | 0.279 | 0.122 | 0.245 |
| ETTh2 | 12.5% | **0.039** | **0.125** | 0.040 | 0.130 | 0.057 | 0.152 | 0.187 | 0.319 | 0.101 | 0.231 | 0.100 | 0.216 | 0.095 | 0.212 | 0.042 | 0.133 | 0.044 | 0.138 | 0.305 | 0.431 | 0.163 | 0.289 |
| | 25% | **0.044** | **0.135** | 0.046 | 0.141 | 0.061 | 0.158 | 0.279 | 0.390 | 0.115 | 0.246 | 0.127 | 0.247 | 0.137 | 0.258 | 0.049 | 0.147 | 0.050 | 0.149 | 0.322 | 0.444 | 0.206 | 0.331 |
| | 37.5% | **0.051** | **0.147** | 0.052 | 0.151 | 0.067 | 0.166 | 0.400 | 0.465 | 0.126 | 0.257 | 0.158 | 0.276 | 0.187 | 0.304 | 0.056 | 0.158 | 0.060 | 0.163 | 0.353 | 0.462 | 0.252 | 0.370 |
| | 50% | **0.059** | **0.158** | 0.060 | 0.162 | 0.073 | 0.174 | 0.602 | 0.572 | 0.136 | 0.268 | 0.183 | 0.299 | 0.232 | 0.341 | 0.065 | 0.170 | 0.068 | 0.173 | 0.369 | 0.472 | 0.316 | 0.419 |
| | Avg | **0.048** | **0.141** | 0.049 | 0.146 | 0.065 | 0.163 | 0.367 | 0.436 | 0.119 | 0.250 | 0.142 | 0.259 | 0.163 | 0.279 | 0.053 | 0.152 | 0.055 | 0.156 | 0.337 | 0.452 | 0.234 | 0.352 |
| ECL | 12.5% | **0.080** | **0.194** | 0.085 | 0.202 | 0.055 | 0.160 | 0.196 | 0.321 | 0.102 | 0.229 | 0.092 | 0.214 | 0.107 | 0.237 | 0.093 | 0.210 | 0.089 | 0.210 | 0.218 | 0.326 | 0.190 | 0.308 |
| | 25% | **0.087** | **0.203** | 0.089 | 0.206 | 0.065 | 0.175 | 0.207 | 0.332 | 0.121 | 0.252 | 0.118 | 0.247 | 0.120 | 0.251 | 0.097 | 0.214 | 0.096 | 0.220 | 0.219 | 0.326 | 0.197 | 0.312 |
| | 37.5% | **0.094** | **0.211** | 0.094 | 0.213 | 0.076 | 0.189 | 0.219 | 0.344 | 0.141 | 0.273 | 0.144 | 0.276 | 0.136 | 0.266 | 0.102 | 0.220 | 0.104 | 0.229 | 0.222 | 0.328 | 0.203 | 0.315 |
| | 50% | **0.101** | **0.220** | 0.100 | 0.221 | 0.091 | 0.208 | 0.235 | 0.357 | 0.160 | 0.293 | 0.175 | 0.305 | 0.158 | 0.284 | 0.108 | 0.228 | 0.113 | 0.239 | 0.228 | 0.331 | 0.210 | 0.319 |
| | Avg | **0.090** | **0.207** | 0.092 | 0.210 | 0.072 | 0.183 | 0.214 | 0.339 | 0.131 | 0.262 | 0.132 | 0.260 | 0.130 | 0.259 | 0.100 | 0.218 | 0.101 | 0.225 | 0.222 | 0.328 | 0.200 | 0.313 |
| Weather | 12.5% | 0.026 | 0.049 | **0.025** | **0.045** | 0.029 | 0.049 | 0.057 | 0.141 | 0.047 | 0.101 | 0.039 | 0.084 | 0.041 | 0.107 | 0.027 | 0.051 | 0.026 | 0.047 | 0.037 | 0.093 | 0.031 | 0.076 |
| | 25% | **0.028** | **0.052** | 0.029 | 0.052 | 0.031 | 0.053 | 0.065 | 0.155 | 0.052 | 0.111 | 0.048 | 0.103 | 0.064 | 0.163 | 0.029 | 0.056 | 0.030 | 0.054 | 0.042 | 0.100 | 0.035 | 0.082 |
| | 37.5% | 0.033 | 0.060 | **0.031** | **0.057** | 0.035 | 0.058 | 0.081 | 0.180 | 0.058 | 0.121 | 0.057 | 0.117 | 0.107 | 0.229 | 0.033 | 0.062 | 0.032 | 0.060 | 0.049 | 0.111 | 0.040 | 0.091 |
| | 50% | 0.037 | 0.065 | **0.034** | **0.062** | 0.038 | 0.063 | 0.102 | 0.207 | 0.065 | 0.133 | 0.066 | 0.134 | 0.183 | 0.312 | 0.037 | 0.068 | 0.037 | 0.067 | 0.053 | 0.114 | 0.046 | 0.099 |
| | Avg | 0.031 | 0.056 | **0.030** | **0.054** | 0.060 | 0.144 | 0.076 | 0.171 | 0.055 | 0.117 | 0.052 | 0.110 | 0.099 | 0.203 | 0.032 | 0.059 | 0.031 | 0.057 | 0.045 | 0.104 | 0.038 | 0.087 |

we are starting to shift our focus towards pursuing more accurate algorithms that possess enhanced capabilities for handling longer inputs. Instead of restricting the input length to a fixed small value, it is pragmatic to tune both the input length and model parameters based on performance, as it is often the primary concern in practical usage. Exploring the utilization of extremely long inputs, such as Chatgpt or LLM, is among our future research directions.

Table 29: Full results of short-term forecasting.

| | Methods | GPT2(6) | TimesNet | PatchTST | N-HiTS | N-BEATS | ETSformer | LightTS | DLinear | FEDformer | Stationary | Autoformer | Informer | Reformer |
|---|---|---|---|---|---|---|---|---|---|---|---|---|---|---|
| *Yearly* | SMAPE | 13.531 | **13.387** | 13.477 | 13.418 | 13.436 | 18.009 | 14.247 | 16.965 | 13.728 | 13.717 | 13.974 | 14.727 | 16.169 |
| | MASE | 3.015 | **2.996** | 3.019 | 3.045 | 3.043 | 4.487 | 3.109 | 4.283 | 3.048 | 3.078 | 3.134 | 3.418 | 3.800 |
| | OWA | 0.793 | **0.786** | 0.792 | 0.793 | 0.794 | 1.115 | 0.827 | 1.058 | 0.803 | 0.807 | 0.822 | 0.881 | 0.973 |
| *Quarterly* | SMAPE | 10.177 | **10.100** | 10.38 | 10.202 | 10.124 | 13.376 | 11.364 | 12.145 | 10.792 | 10.958 | 11.338 | 11.360 | 13.313 |
| | MASE | 1.194 | **1.182** | 1.233 | 1.194 | 1.169 | 1.906 | 1.328 | 1.520 | 1.283 | 1.325 | 1.365 | 1.401 | 1.775 |
| | OWA | 0.898 | **0.890** | 0.921 | 0.899 | 0.886 | 1.302 | 1.000 | 1.106 | 0.958 | 0.981 | 1.012 | 1.027 | 1.252 |
| *Monthly* | SMAPE | 12.894 | **12.670** | 12.959 | 12.791 | 12.677 | 14.588 | 14.014 | 13.514 | 14.260 | 13.917 | 13.958 | 14.062 | 20.128 |
| | MASE | 0.956 | **0.933** | 0.970 | 0.969 | 0.937 | 1.368 | 1.053 | 1.037 | 1.102 | 1.097 | 1.103 | 1.141 | 2.614 |
| | OWA | 0.897 | **0.878** | 0.905 | 0.899 | 0.880 | 1.149 | 0.981 | 0.956 | 1.012 | 0.998 | 1.002 | 1.024 | 1.927 |
| *Others* | SMAPE | 4.940 | **4.891** | 4.952 | 5.061 | 4.925 | 7.267 | 15.880 | 6.709 | 4.954 | 6.302 | 5.485 | 24.460 | 32.491 |
| | MASE | 3.228 | 3.302 | 3.347 | **3.216** | 3.391 | 5.240 | 11.434 | 4.953 | 3.264 | 4.064 | 3.865 | 20.960 | 33.355 |
| | OWA | 1.029 | **1.035** | 1.049 | 1.040 | 1.053 | 1.591 | 3.474 | 1.487 | 1.036 | 1.304 | 1.187 | 5.879 | 8.679 |
| *Average* | SMAPE | 11.991 | **11.829** | 12.059 | 11.927 | 11.851 | 14.718 | 13.525 | 13.639 | 12.840 | 12.780 | 12.909 | 14.086 | 18.200 |
| | MASE | 1.600 | **1.585** | 1.623 | 1.613 | 1.599 | 2.408 | 2.111 | 2.095 | 1.701 | 1.756 | 1.771 | 2.718 | 4.223 |
| | OWA | 0.861 | **0.851** | 0.869 | 0.861 | 0.855 | 1.172 | 1.051 | 1.051 | 0.918 | 0.930 | 0.939 | 1.230 | 1.775 |

