| **0.418** | **0.446** | 0.500 | - | **0.291** | **0.326** | **0.362** | **0.433** |
| DLinear Zeng et al. (2023) | MSE | 0.442 | 0.617 | 1.424 | - | 0.236 | 0.306 | 0.380 | 0.674 |
| | MAE | 0.456 | 0.542 | 0.849 | - | 0.326 | 0.373 | 0.423 | 0.583 |
| PatchTST Nie et al. (2022) | MSE | 0.401 | **0.452** | **0.464** | - | **0.206** | **0.264** | **0.334** | **0.454** |
| | MAE | 0.421 | **0.455** | **0.469** | - | **0.288** | **0.324** | **0.367** | 0.483 |
| FEDformer Zhou et al. (2022) | MSE | **0.390** | 0.457 | **0.477** | - | 0.299 | 0.290 | 0.378 | 0.523 |
| | MAE | 0.424 | 0.465 | 0.483 | - | 0.320 | 0.361 | 0.427 | 0.510 |
| Autoformer Wu et al. (2021) | MSE | 0.428 | 0.496 | 0.486 | - | 0.232 | 0.291 | 0.478 | 0.533 |
| | MAE | 0.468 | 0.504 | 0.496 | - | 0.322 | 0.357 | 0.517 | 0.538 |

## H.6 Full Results of Classification

Table 25: Full results for the classification task. ∗. in the Transformers indicates the name of ∗former.

| Methods | Classical methods | | RNN | | TCN | Transformers | | | | | | | | | MLP | | TimesNet | GPT2(6) |
|---|---|---|---|---|---|---|---|---|---|---|---|---|---|---|---|---|---|---|
| | XGBoost | Rocket | LSTNet | LSSL | | Trans. | Re. | In. | Pyra. | Auto. | Station. | FED. | ETS. | Flow. | DLinear | LightTS. | | |
| EthanolConcentration | 43.7 | 45.2 | 39.9 | 31.1 | 28.9 | 32.7 | 31.9 | 31.6 | 30.8 | 31.6 | 32.7 | 31.2 | 28.1 | 33.8 | 32.6 | 29.7 | 35.7 | 34.2 |
| FaceDetection | 63.3 | 64.7 | 65.7 | 66.7 | 52.8 | 67.3 | 68.6 | 67.0 | 65.7 | 68.4 | 68.0 | 66.0 | 66.3 | 67.6 | 68.0 | 67.5 | 68.6 | 69.2 |
| Handwriting | 15.8 | 58.8 | 25.8 | 24.6 | 53.3 | 32.0 | 27.4 | 32.8 | 29.4 | 36.7 | 31.6 | 28.0 | 32.5 | 33.8 | 27.0 | 26.1 | 32.1 | 32.7 |
| Heartbeat | 73.2 | 75.6 | 77.1 | 72.7 | 75.6 | 76.1 | 77.1 | 80.5 | 75.6 | 74.6 | 73.7 | 73.7 | 71.2 | 77.6 | 75.1 | 75.1 | 78.0 | 77.2 |
| JapaneseVowels | 86.5 | 96.2 | 98.1 | 98.4 | 98.9 | 98.7 | 97.8 | 98.9 | 98.4 | 96.2 | 99.2 | 98.4 | 95.9 | 98.9 | 96.2 | 96.2 | 98.4 | 98.6 |
| PEMS-SF | 98.3 | 75.1 | 86.7 | 86.1 | 68.8 | 82.1 | 82.7 | 81.5 | 83.2 | 82.7 | 87.3 | 80.9 | 86.0 | 83.8 | 75.1 | 88.4 | 89.6 | 87.9 |
| SelfRegulationSCP1 | 84.6 | 90.8 | 84.0 | 90.8 | 84.6 | 92.2 | 90.4 | 90.1 | 88.1 | 84.0 | 89.4 | 88.7 | 89.6 | 92.5 | 87.3 | 89.8 | 91.8 | 93.2 |
| SelfRegulationSCP2 | 48.9 | 53.3 | 52.8 | 52.2 | 55.6 | 53.9 | 56.7 | 53.3 | 53.3 | 50.6 | 57.2 | 54.4 | 55.0 | 56.1 | 50.5 | 51.1 | 57.2 | 59.4 |
| SpokenArabicDigits | 69.6 | 71.2 | 100.0 | 100.0 | 95.6 | 98.4 | 97.0 | 100.0 | 99.6 | 100.0 | 100.0 | 100.0 | 100.0 | 98.8 | 81.4 | 100.0 | 99.0 | 99.2 |
| UWaveGestureLibrary | 75.9 | 94.4 | 87.8 | 85.9 | 88.4 | 85.6 | 85.6 | 85.6 | 83.4 | 85.9 | 87.5 | 85.3 | 85.0 | 86.6 | 82.1 | 80.3 | 85.3 | 88.1 |
| Average | 66.0 | 72.5 | 71.8 | 70.9 | 70.3 | 71.9 | 71.5 | 72.1 | 70.8 | 71.1 | 72.7 | 70.7 | 71.0 | 73.0 | 67.5 | 70.4 | **73.6** | **74.0** |

## H.7 Full Results of Anomaly Detection

## H.8 Full Results of Imputation

## H.9 Full Results of Short-term Forecasting

Table 26: Full results for the anomaly detection.

| Methods | SMD P | SMD R | SMD F1 | MSL P | MSL R | MSL F1 | SMAP P | SMAP R | SMAP F1 | SWaT P | SWaT R | SWaT F1 | PSM P | PSM R | PSM F1 | Avg F1 % |
|---|---|---|---|---|---|---|---|---|---|---|---|---|---|---|---|---|
| GPT(6) | **88.89** | 84.98 | **86.89** | 82.00 | **82.91** | **82.45** | 90.60 | 60.95 | 72.88 | 92.20 | 96.34 | 94.23 | 98.62 | 95.68 | 97.13 | **86.72** |
| TimesNet[*] | 87.91 | 81.54 | 84.61 | **89.54** | 75.36 | 81.84 | 90.14 | 56.40 | 69.39 | 90.75 | 95.40 | 93.02 | 98.51 | **96.20** | **97.34** | 85.24 |
| PatchTST | 87.26 | 82.14 | 84.62 | 88.34 | 70.96 | 78.70 | 90.64 | 55.46 | 68.82 | 91.10 | 80.94 | 85.72 | 98.84 | 93.47 | 96.08 | 82.79 |
| ETSformer | 87.44 | 79.23 | 83.13 | 85.13 | 84.93 | 85.03 | 92.25 | 55.75 | 69.50 | 90.02 | 80.36 | 84.91 | 99.31 | 85.28 | 91.76 | 82.87 |
| FEDformer | 87.95 | 82.39 | 85.08 | 77.14 | 80.07 | 78.57 | 90.47 | 58.10 | 70.76 | 90.17 | 96.42 | 93.19 | 97.31 | 97.16 | 97.23 | 84.97 |
| LightTS | 87.10 | 78.42 | 82.53 | 82.40 | 75.78 | 78.95 | 92.58 | 55.27 | 69.21 | 91.98 | 94.72 | 93.33 | 98.37 | 95.97 | 97.15 | 84.23 |
| DLinear | 83.62 | 71.52 | 77.10 | 84.34 | 85.42 | 84.88 | 92.32 | 55.41 | 69.26 | 80.91 | 95.30 | 87.52 | 98.28 | 89.26 | 93.55 | 82.46 |
| Stationary | 88.33 | 81.21 | 84.62 | 68.55 | 89.14 | 77.50 | 89.37 | 59.02 | 71.09 | 68.03 | 96.75 | 79.88 | 97.82 | 96.76 | 97.29 | 82.08 |
| Autoformer | 88.06 | 82.35 | 85.11 | 77.27 | 80.92 | 79.05 | 90.40 | 58.62 | 71.12 | 89.85 | 95.81 | 92.74 | 99.08 | 88.15 | 93.29 | 84.26 |
| Pyraformer | 85.61 | 80.61 | 83.04 | 83.81 | 85.93 | 84.86 | 92.54 | 57.71 | 71.09 | 87.92 | 96.00 | 91.78 | 71.67 | 96.02 | 82.08 | 82.57 |
| Anomaly Transformer[**] | 88.91 | 82.23 | 85.49 | 79.61 | 87.37 | 83.31 | 91.85 | 58.11 | 71.18 | 72.51 | 97.32 | 83.10 | 68.35 | 94.72 | 79.40 | 80.50 |
| Informer | 86.60 | 77.23 | 81.65 | 81.77 | 86.48 | 84.06 | 90.11 | 57.13 | 69.92 | 70.29 | 96.75 | 81.43 | 64.27 | 96.33 | 77.10 | 78.83 |
| Reformer | 82.58 | 69.24 | 75.32 | 85.51 | 83.31 | 84.40 | 90.91 | 57.44 | 70.40 | 72.50 | 96.53 | 82.80 | 59.93 | 95.38 | 73.61 | 77.31 |
| LogTransformer | 83.46 | 70.13 | 76.21 | 73.05 | 87.37 | 79.57 | 89.15 | 57.59 | 69.97 | 68.67 | 97.32 | 80.52 | 63.06 | 98.00 | 76.74 | 76.60 |
| Transformer | 83.58 | 76.13 | 79.56 | 71.57 | 87.37 | 78.68 | 89.37 | 57.12 | 69.70 | 68.84 | 96.53 | 80.37 | 62.75 | 96.56 | 76.07 | 76.88 |

[*] We reproduce the results of TimesNet by https://github.com/thuml/Time-Series-Library.
[**] We replace the joint criterion in Anomaly Transformer with reconstruction error for fair comparison.

Table 27: Full results for the imputation task.

| Methods Mask Ratio | GPT2(3) MSE | MAE | TimesNet MSE | MAE | PatchTST MSE | MAE | ETSformer MSE | MAE | LightTS MSE | MAE | DLinear MSE | MAE | FEDformer MSE | MAE | Stationary MSE | MAE | Autoformer MSE | MAE | Informer MSE | MAE | Reformer MSE | MAE |
|---|---|---|---|---|---|---|---|---|---|---|---|---|---|---|---|---|---|---|---|---|---|---|
| ETTm1 12.5% | **0.017** | **0.085** | 0.023 | 0.101 | 0.041 | 0.130 | 0.096 | 0.229 | 0.093 | 0.206 | 0.080 | 0.193 | 0.052 | 0.166 | 0.032 | 0.119 | 0.046 | 0.144 | 0.063 | 0.180 | 0.042 | 0.146 |
| ETTm1 25% | **0.022** | **0.096** | 0.023 | 0.101 | 0.044 | 0.135 | 0.096 | 0.229 | 0.093 | 0.206 | 0.080 | 0.193 | 0.052 | 0.166 | 0.032 | 0.119 | 0.046 | 0.144 | 0.063 | 0.180 | 0.042 | 0.146 |
| ETTm1 37.5% | **0.029** | **0.111** | 0.029 | 0.111 | 0.049 | 0.143 | 0.133 | 0.271 | 0.113 | 0.231 | 0.103 | 0.219 | 0.069 | 0.191 | 0.039 | 0.131 | 0.057 | 0.161 | 0.079 | 0.200 | 0.063 | 0.182 |
| ETTm1 50% | 0.040 | 0.128 | **0.036** | **0.124** | 0.055 | 0.151 | 0.186 | 0.323 | 0.134 | 0.255 | 0.132 | 0.248 | 0.089 | 0.218 | 0.047 | 0.145 | 0.067 | 0.174 | 0.093 | 0.218 | 0.082 | 0.208 |
| ETTm1 Avg | 0.028 | **0.105** | 0.027 | 0.107 | 0.047 | 0.140 | 0.120 | 0.253 | 0.104 | 0.218 | 0.093 | 0.206 | 0.062 | 0.177 | 0.036 | 0.126 | 0.051 | 0.150 | 0.071 | 0.188 | 0.055 | 0.166 |
| ETTm2 12.5% | **0.017** | **0.076** | 0.018 | 0.080 | 0.026 | 0.094 | 0.108 | 0.239 | 0.034 | 0.127 | 0.062 | 0.166 | 0.056 | 0.159 | 0.021 | 0.088 | 0.023 | 0.092 | 0.133 | 0.270 | 0.108 | 0.228 |
| ETTm2 25% | **0.020** | **0.080** | 0.020 | 0.085 | 0.028 | 0.099 | 0.164 | 0.294 | 0.042 | 0.143 | 0.085 | 0.196 | 0.080 | 0.195 | 0.024 | 0.096 | 0.026 | 0.101 | 0.135 | 0.272 | 0.136 | 0.262 |
| ETTm2 37.5% | **0.022** | **0.087** | 0.023 | 0.091 | 0.030 | 0.104 | 0.237 | 0.356 | 0.051 | 0.159 | 0.106 | 0.222 | 0.110 | 0.231 | 0.027 | 0.103 | 0.030 | 0.108 | 0.155 | 0.293 | 0.175 | 0.300 |
| ETTm2 50% | **0.025** | **0.095** | 0.026 | 0.098 | 0.034 | 0.110 | 0.323 | 0.421 | 0.059 | 0.174 | 0.131 | 0.247 | 0.156 | 0.276 | 0.030 | 0.108 | 0.035 | 0.119 | 0.200 | 0.333 | 0.211 | 0.329 |
| ETTm2 Avg | **0.021** | **0.084** | 0.022 | 0.088 | 0.029 | 0.102 | 0.208 | 0.327 | 0.046 | 0.151 | 0.096 | 0.208 | 0.101 | 0.215 | 0.026 | 0.099 | 0.029 | 0.105 | 0.156 | 0.292 | 0.157 | 0.280 |
| ETTh1 12.5% | **0.043** | **0.140** | 0.057 | 0.159 | 0.093 | 0.201 | 0.126 | 0.263 | 0.240 | 0.345 | 0.151 | 0.267 | 0.070 | 0.190 | 0.060 | 0.165 | 0.074 | 0.182 | 0.114 | 0.234 | 0.074 | 0.194 |
| ETTh1 25% | **0.054** | **0.156** | 0.069 | 0.178 | 0.107 | 0.217 | 0.169 | 0.304 | 0.265 | 0.364 | 0.180 | 0.292 | 0.106 | 0.236 | 0.080 | 0.189 | 0.090 | 0.203 | 0.140 | 0.262 | 0.102 | 0.227 |
| ETTh1 37.5% | **0.072** | **0.180** | 0.084 | 0.196 | 0.120 | 0.230 | 0.220 | 0.347 | 0.296 | 0.382 | 0.215 | 0.318 | 0.124 | 0.258 | 0.102 | 0.212 | 0.109 | 0.222 | 0.174 | 0.293 | 0.135 | 0.261 |
| ETTh1 50% | 0.107 | 0.216 | **0.102** | **0.215** | 0.141 | 0.248 | 0.293 | 0.402 | 0.334 | 0.404 | 0.257 | 0.347 | 0.165 | 0.299 | 0.133 | 0.240 | 0.137 | 0.248 | 0.215 | 0.325 | 0.179 | 0.298 |
| ETTh1 Avg | **0.069** | **0.173** | 0.078 | 0.187 | 0.115 | 0.224 | 0.202 | 0.329 | 0.284 | 0.373 | 0.201 | 0.306 | 0.117 | 0.246 | 0.094 | 0.201 | 0.103 | 0.214 | 0.161 | 0.279 | 0.122 | 0.245 |
| ETTh2 12.5% | **0.039** | **0.125** | 0.040 | 0.130 | 0.057 | 0.152 | 0.187 | 0.319 | 0.101 | 0.231 | 0.100 | 0.216 | 0.095 | 0.212 | 0.042 | 0.133 | 0.044 | 0.138 | 0.305 | 0.431 | 0.163 | 0.289 |
| ETTh2 25% | **0.044** | **0.135** | 0.046 | 0.141 | 0.061 | 0.158 | 0.279 | 0.390 | 0.115 | 0.246 | 0.127 | 0.247 | 0.137 | 0.258 | 0.049 | 0.147 | 0.050 | 0.149 | 0.322 | 0.444 | 0.206 | 0.331 |
| ETTh2 37.5% | **0.051** | **0.147** | 0.052 | 0.151 | 0.067 | 0.166 | 0.400 | 0.465 | 0.126 | 0.257 | 0.158 | 0.276 | 0.187 | 0.304 | 0.056 | 0.158 | 0.060 | 0.163 | 0.353 | 0.462 | 0.252 | 0.370 |
| ETTh2 50% | **0.059** | **0.158** | 0.060 | 0.162 | 0.073 | 0.174 | 0.602 | 0.572 | 0.136 | 0.268 | 0.183 | 0.299 | 0.232 | 0.341 | 0.065 | 0.170 | 0.068 | 0.173 | 0.369 | 0.472 | 0.316 | 0.419 |
| ETTh2 Avg | **0.048** | **0.141** | 0.049 | 0.146 | 0.065 | 0.163 | 0.367 | 0.436 | 0.119 | 0.250 | 0.142 | 0.259 | 0.163 | 0.279 | 0.053 | 0.152 | 0.055 | 0.156 | 0.337 | 0.452 | 0.234 | 0.352 |
| ECL 12.5% | **0.080** | **0.194** | 0.085 | 0.202 | 0.055 | 0.160 | 0.196 | 0.321 | 0.102 | 0.229 | 0.092 | 0.214 | 0.107 | 0.237 | 0.093 | 0.210 | 0.089 | 0.210 | 0.218 | 0.326 | 0.190 | 0.308 |
| ECL 25% | **0.087** | **0.203** | 0.089 | 0.206 | 0.065 | 0.175 | 0.207 | 0.332 | 0.121 | 0.252 | 0.118 | 0.247 | 0.120 | 0.251 | 0.097 | 0.214 | 0.096 | 0.220 | 0.219 | 0.326 | 0.197 | 0.312 |
| ECL 37.5% | 0.094 | **0.211** | **0.094** | 0.213 | 0.076 | 0.189 | 0.219 | 0.344 | 0.141 | 0.273 | 0.144 | 0.276 | 0.136 | 0.266 | 0.102 | 0.220 | 0.104 | 0.229 | 0.222 | 0.328 | 0.203 | 0.315 |
| ECL 50% | 0.101 | **0.220** | **0.100** | 0.221 | 0.091 | 0.208 | 0.235 | 0.357 | 0.160 | 0.293 | 0.175 | 0.305 | 0.158 | 0.284 | 0.108 | 0.228 | 0.113 | 0.239 | 0.228 | 0.331 | 0.210 | 0.319 |
| ECL Avg | **0.090** | **0.207** | 0.092 | 0.210 | 0.072 | 0.183 | 0.214 | 0.339 | 0.131 | 0.262 | 0.132 | 0.260 | 0.130 | 0.259 | 0.100 | 0.218 | 0.101 | 0.225 | 0.222 | 0.328 | 0.200 | 0.313 |
| Weather 12.5% | 0.026 | 0.049 | **0.025** | **0.045** | 0.029 | 0.049 | 0.057 | 0.141 | 0.047 | 0.101 | 0.039 | 0.084 | 0.041 | 0.107 | 0.027 | 0.051 | 0.026 | 0.047 | 0.037 | 0.093 | 0.031 | 0.076 |
| Weather 25% | **0.028** | **0.052** | 0.029 | 0.052 | 0.031 | 0.053 | 0.065 | 0.155 | 0.052 | 0.111 | 0.048 | 0.103 | 0.064 | 0.163 | 0.029 | 0.056 | 0.030 | 0.054 | 0.042 | 0.100 | 0.035 | 0.082 |
| Weather 37.5% | 0.033 | 0.060 | **0.031** | **0.057** | 0.035 | 0.058 | 0.081 | 0.180 | 0.058 | 0.121 | 0.057 | 0.117 | 0.107 | 0.229 | 0.033 | 0.062 | 0.032 | 0.060 | 0.049 | 0.111 | 0.040 | 0.091 |
| Weather 50% | 0.037 | 0.065 | **0.034** | **0.062** | 0.038 | 0.063 | 0.102 | 0.207 | 0.065 | 0.133 | 0.066 | 0.134 | 0.183 | 0.312 | 0.037 | 0.068 | 0.037 | 0.067 | 0.053 | 0.114 | 0.046 | 0.099 |
| Weather Avg | 0.031 | 0.056 | **0.030** | **0.054** | 0.060 | 0.144 | 0.076 | 0.171 | 0.055 | 0.117 | 0.052 | 0.110 | 0.099 | 0.203 | 0.032 | 0.059 | 0.031 | 0.057 | 0.045 | 0.104 | 0.038 | 0.087 |

Table 28: Full results of short-term forecasting.

| Methods | | GPT2(6) | TimesNet | PatchTST | N-HiTS | N-BEATS | ETSformer | LightTS | DLinear | FEDformer | Stationary | Autoformer | Informer | Reformer |
|---|---|---|---|---|---|---|---|---|---|---|---|---|---|---|
| Yearly | SMAPE | 13.531 | **13.387** | 13.477 | 13.418 | 13.436 | 18.009 | 14.247 | 16.965 | 13.728 | 13.717 | 13.974 | 14.727 | 16.169 |
| Yearly | MASE | 3.015 | **2.996** | 3.019 | 3.045 | 3.043 | 4.487 | 3.109 | 4.283 | 3.048 | 3.078 | 3.134 | 3.418 | 3.800 |
| Yearly | OWA | 0.793 | **0.786** | 0.792 | 0.793 | 0.794 | 1.115 | 0.827 | 1.058 | 0.803 | 0.807 | 0.822 | 0.881 | 0.973 |
| Quarterly | SMAPE | 10.177 | **10.100** | 10.38 | 10.202 | 10.124 | 13.376 | 11.364 | 12.145 | 10.792 | 10.958 | 11.338 | 11.360 | 13.313 |
| Quarterly | MASE | 1.194 | **1.182** | 1.233 | 1.194 | 1.169 | 1.906 | 1.328 | 1.520 | 1.283 | 1.325 | 1.365 | 1.401 | 1.775 |
| Quarterly | OWA | 0.898 | **0.890** | 0.921 | 0.899 | 0.886 | 1.302 | 1.000 | 1.106 | 0.958 | 0.981 | 1.012 | 1.027 | 1.252 |
| Monthly | SMAPE | 12.894 | **12.670** | 12.959 | 12.791 | 12.677 | 14.588 | 14.014 | 13.514 | 14.260 | 13.917 | 13.958 | 14.062 | 20.128 |
| Monthly | MASE | 0.956 | **0.933** | 0.970 | 0.969 | 0.937 | 1.368 | 1.053 | 1.037 | 1.102 | 1.097 | 1.103 | 1.141 | 2.614 |
| Monthly | OWA | 0.897 | **0.878** | 0.905 | 0.899 | 0.880 | 1.149 | 0.981 | 0.956 | 1.012 | 0.998 | 1.002 | 1.024 | 1.927 |
| Others | SMAPE | 4.940 | **4.891** | 4.952 | 5.061 | 4.925 | 7.267 | 15.880 | 6.709 | 4.954 | 6.302 | 5.485 | 24.460 | 32.491 |
| Others | MASE | 3.228 | 3.302 | 3.347 | **3.216** | 3.391 | 5.240 | 11.434 | 4.953 | 3.264 | 4.064 | 3.865 | 20.960 | 33.355 |
| Others | OWA | 1.029 | **1.035** | 1.049 | 1.040 | 1.053 | 1.591 | 3.474 | 1.487 | 1.036 | 1.304 | 1.187 | 5.879 | 8.679 |
| Average | SMAPE | 11.991 | **11.829** | 12.059 | 11.927 | 11.851 | 14.718 | 13.525 | 13.639 | 12.840 | 12.780 | 12.909 | 14.086 | 18.200 |
| Average | MASE | 1.600 | **1.585** | 1.623 | 1.613 | 1.599 | 2.408 | 2.111 | 2.095 | 1.701 | 1.756 | 1.771 | 2.718 | 4.223 |
| Average | OWA | 0.861 | **0.851** | 0.869 | 0.861 | 0.855 | 1.172 | 1.051 | 1.051 | 0.918 | 0.930 | 0.939 | 1.230 | 1.775 |