# OpenReview forum: "One Fits All: Power General Time Series Analysis by Pretrained LM"
_NeurIPS.cc/2023/Conference — NeurIPS 2023 spotlight_

### Official Review · Reviewer_qj2Q · 2023-06-12

**Soundness:** 4 excellent
**Presentation:** 2 fair
**Contribution:** 4 excellent
**Rating:** 8
**Confidence:** 3

**Summary:**

In this paper, the authors suggest leveraging pre-trained large language models, specifically utilizing a frozen Transformer backbone, for comprehensive time series analysis. The unique method involves fine-tuning only the input embeddings of time series data and the output layers. The authors substantiate their proposal with large-scale experiments across multiple benchmarks, demonstrating that their approach consistently outperforms various established baseline methods in diverse time series analysis tasks.

**Strengths:**

1. The paper's proposed method impresses with its simplicity, efficacy, and clear motivation. Despite not introducing any complex modules or additional hyperparameters to the existing model, it consistently outperforms more intricate techniques.

2. The experiments undertaken within the paper are robust and comprehensive. The authors have tested the proposed method across a broad spectrum of time series analysis aspects, such as long-term and short-term forecasting, and abnormality detection. In each scenario, the method demonstrates superior performance when compared with multiple strong baselines.

3. The insightful correlation drawn between self-attention and Principal Component Analysis (PCA) elucidates why the proposed method performs remarkably well. This strengthens the credibility of the results, making them more convincing.

**Weaknesses:**

1. The paper neglects to detail the computational cost differences between the proposed method and alternative approaches. Given that a pre-trained GPT-2 backbone is potentially substantial in size, the computational expense of the presented method might significantly outweigh that of other baseline methods.

2. Although the proposed method seems compatible with various autoregressive Transformers, the experiments rely exclusively on GPT-2. It would be beneficial to extend the experimental scope to encompass additional models such as DeBERTa, GPT-J, OPT, among others.

3. The paper's overall presentation requires improvement. Tables appear congested, and font sizes within figures may be too small for legible printing. Additionally, there is an inconsistent distribution of space on some pages, and many section titles are overly lengthy. This compromises readability and detracts from the overall impact of the findings.

**Questions:**

N/A

---

> ### Author Rebuttal · Authors · 2023-08-09
>
> We wish to express our sincere gratitude to Reviewer qj2Q for the positive assessment of our work. Your endorsement is both affirming and motivating, and it strengthens our resolve to continue our research in this field. We concur with Reviewer qj2Q's observation that the presentation of our paper could be enhanced. Specifically, we acknowledge that the tables appear cluttered and that the font sizes may be too small. This issue arose as we endeavored to comprehensively address a wide range of topics within the confines of this nine-page paper. Our goal was to deliver an exhaustive numerical analysis and provide a clear exposition of our knowledge transfer approach.
>
> We recognize that wading through all the supporting details can be challenging for reviewers, and we lament that space limitations forced us to omit some essential information in our supplementary material. It's noteworthy that our full paper, inclusive of all supporting information, spans over 30 pages. Nevertheless, we value your feedback and will make every effort to incorporate your suggestions in our revisions.
>
> ***Q1 for Reviewer qj2Q. The paper neglects to detail the computational cost differences between the proposed method and alternative approaches. Given that a pre-trained GPT-2 backbone is potentially substantial in size, the computational expense of the presented method might significantly outweigh that of other baseline methods.***
>
> Thank you for pointing that out. We concur that evaluating the computational cost is crucial, especially for large models like the one under consideration. The subsequent table presents the results.
>
> Each baseline model is offered in two configurations, with model dimensions of 32 and 768 (analogous to GPT-2). Additionally, each baseline model consists of three layers. We assessed the computational expenses using a single batch of ETTh2 (with a batch size of 128) on a solitary V100 GPU. The results indicate that GPT-2(3) offers a marked enhancement in both time efficiency and parameter count relative to the baselines with equivalent model dimensions. This substantial uptick in time efficiency can be primarily attributed to the proficient optimization techniques employed by huggingface. Furthermore, the training parameters constitute only 6.12\% for GPT-2(3) and 4.60\% for GPT-2(6).
>
>  | Model         | Training Params | Training Params Percentage(%) | Training Time for 1 Batch (s) | Inference Time for 1 Batch (s) |
>  | ------------- | --------------- | ----------------------------- | ----------------------------- | ------------------------------ |
>  | FEDformer-32  | 437,319         | 100                           | 0.889                         | 0.17                           |
>  | TimesNet-32   | 1,905,015       | 100                           | 0.747                         | 0.302                          |
>  | PatchTST-32   | 543,232         | 100                           | 0.043                         | 0.022                          |
>  | ------------- | --------------- | ----------------------------- | ----------------------------- | ------------------------------ |
>  | FEDformer-768 | 33,105,415      | 100                           | 0.208                         | 0.056                          |
>  | TimesNet-768  | 42,358,519      | 100                           | 5.723                         | 2.162                          |
>  | PatchTST-768  | 19,677,024      | 100                           | 0.457                         | 0.123                          |
>  | GPT-2(3)-768  | 3,906,912       | 6.12                          | 0.093                         | 0.032                          |
>  | ------------- | --------------- | ----------------------------- | ----------------------------- | ------------------------------ |
>  | GPT-2(6)-768  | 3,916,128       | 4.60                          | 0.104                         | 0.054                          |
>
> ***Q2 for Reviewer qj2Q. Although the proposed method seems compatible with various autoregressive Transformers, the experiments rely exclusively on GPT-2. It would be beneficial to extend the experimental scope to encompass additional models such as DeBERTa, GPT-J, OPT, among others.***
>
> We wholeheartedly concur that broadening our experimental range to include more models would augment the depth of our study. However, given the significant effort and resources required to conduct a wide array of experiments, we primarily relied on GPT-2 for our core investigations. Nevertheless, we have also utilized the CV pre-trained model BEiT, and the NLP pre-trained model BERT. Both were trained on 5\% of ETTh2 and 5\% of ETTm2, underscoring that the capability for knowledge transfer isn't exclusive to GPT-2. Detailed results related to these models are available in Appendix H.5. As we move forward, our research will delve into the performance implications of transferring pre-trained models from diverse modalities to time series analysis.
>
>
> ***Q3 for Reviewer qj2Q. The paper's overall presentation requires improvement. Tables appear congested, and font sizes within figures may be too small for legible printing. Additionally, there is an inconsistent distribution of space on some pages, and many section titles are overly lengthy. This compromises readability and detracts from the overall impact of the findings.***
>
> Thank you for your feedback. Owing to page constraints, we had to condense our figures and tables, inadvertently affecting their readability. In subsequent versions, we will refine the paper's layout, ensuring legible font sizes and optimal paragraph spacing.

---

> > ### Comment · Reviewer_qj2Q · 2023-08-11
> >
> > Thank you for your new experimental results and clarifications. I increased my review score to 8: Strong Accept

---

> > > ### Author Response · Authors · 2023-08-13
> > >
> > >  We are thrilled to hear that our new experimental results and clarifications have made a positive impact on the paper. Your review has been invaluable in improving the quality of our work. Once again, thank you for your time, effort, and support.

---

### Official Review · Reviewer_idRZ · 2023-07-01

**Soundness:** 2 fair
**Presentation:** 3 good
**Contribution:** 2 fair
**Rating:** 4
**Confidence:** 4

**Summary:**

This paper shows that fine-tuning language models along with some task specific layers, e.g. input and position embedding and lay norms, can achieve comparable and sate-of-the-art performance on various time series tasks, including forecasting, imputation, anomaly detection and classification tasks.

**Strengths:**

(1) The method for time series analysis is simple and can work for different time series analysis tasks.

(2) Results are comparable or state-of-the-art in different benchmark datasets and time series analysis tasks.


**Weaknesses:**

(1) This work mainly conduct experiments on simple benchmark datasets. Many real-word datasets have complicated dynamics and large noise and the prior knowledge within pre-train models do not necessarily align with such dynamics. If the success of using pre-train models are due to the prior knowledge pre-train models own, what are results if the time series data does not align with such prior knowledge? This paper seems not to have deep discussion regarding this matter.
(2) It is unclear why GPT-2 (6) perform better than the full-layer fine-tuning baseline. Deep layers often tend to learn global information within the data. It is not clear why authors use GPT-2 (6). Authors lack of deep discussion regarding this matter.

**Questions:**

See Weaknesses.

Missing references:

(1) LogTransformer was proposed in [1]. Results of LogTransformer was in this paper but [1] was not cited.
(2) Using pre-trained transformers to model time series is not a new idea. [2] uses pre-train LMs to conduct time series forecasting and [3] uses Vision Transformer to model irregularly sampled time series. However, this paper does not discussion this two work and add them as baselines in appropriate tasks.

[1] Li et al. Enhancing the Locality and Breaking the Memory Bottleneck of Transformer on Time Series Forecasting. NeurIPS 2019.
[2] Xue and Salim. PromptCast: A New Prompt-based Learning Paradigm for Time Series Forecasting.
[3] Li et al. Time Series as Images: Vision Transformer for Irregularly Sampled Time Series.

**Limitations:**

See Weaknesses and Questions section.

---

> ### Author Rebuttal · Authors · 2023-08-09
>
> We sincerely thank Reviewer dRZ for the favorable evaluation of our methodology's simplicity and effectiveness, as well as its state-of-the-art performance across diverse time series tasks. We deeply appreciate your detailed and perceptive feedback. Rest assured, we are committed to addressing and resolving your concerns.
>
> ***Q1 for Reviewer idRZ. This work mainly conduct experiments on simple benchmark datasets. Many real-word datasets have complicated dynamics and large noise and the prior knowledge within pre-train models do not necessarily align with such dynamics. If the success of using pre-train models are due to the prior knowledge pre-train models own, what are results if the time series data does not align with such prior knowledge? This paper seems not to have deep discussion regarding this matter.***
>
> We concur wholeheartedly with the reviewer that the absence of a comprehensive and large-scale benchmark dataset—akin to ImageNet in computer vision—is a substantial challenge for the whole community. We eagerly anticipate future advancements that will offer a more extensive and diverse benchmark suite for researchers.
>
> However, we argue that we have included most complicated benchmark datasets for time series forecasting in our study, e.g. the M4 dataset with 100,000 time series that served as the foundation for the fourth Makridakis forecasting competition—a competition widely recognized as one of the most representative in time series forecasting. The benchmark datasets used in our study have also been used by 400 papers in the last two years, with 20 papers from premier conferences, according to Semantic Scholar analysis. Although it is impossible for these datasets to showcase all the challenges faced by real applications of time series analysis, they do highlight a wide range of key challenges from real applications.
>
> We further argue that both LM and time series analysis is based on the idea of auto-regression. It is this high-level similarity that inspires us to explore frozen LM for time series analysis. When encountering a mismatch in prior assumptions between model and datasets, appropriate fine-tuning technology can always be used to fill out the gap.
>
> ***Q2 for Reviewer idRZ. It is unclear why GPT-2 (6) perform better than the full-layer fine-tuning baseline. Deep layers often tend to learn global information within the data. It is not clear why authors use GPT-2 (6). Authors lack of deep discussion regarding this matter.***
>
> One reason for GPT-2(6) to perform better than full layer fine-tuning baseline is the rank collapsing property of transformer. In other words, as we go into deeper layers, we tend to see that more and more tokens exhibit similar vectors. Since most time series analyses depend on a relatively shorter history compared to texts, it further amplifies the impact of the rank-collapsing property. As a result, important detailed information may be removed from the outputs from the deep layers, limiting the prediction accuracy.
>
> ***Q3 for Reviewer idRZ. LogTransformer was proposed in [1]. Results of LogTransformer was in this paper but [1] was not cited.***
>
> Thanks for the reminder, we will add the reference of LogTransformer.
>
> ***Q4 for Reviewer idRZ. Using pre-trained transformers to model time series is not a new idea. [2] uses pre-train LMs to conduct time series forecasting and [3] uses Vision Transformer to model irregularly sampled time series. However, this paper does not discussion this two work and add them as baselines in appropriate tasks.***
>
> Although on the very high level, using pre-trained transformer for time series analysis is not completely new, our study is clearly new and novel: we show it is possible to directly use a frozen LM to achieve the state-of-the-art performance for time series analysis. Compared to texts, time series data is more noisy and diverse, and many time series data is application dependent, which makes the cross-modality transferring learning challenging. We support this claim by both extensive empirical studies and theoretical analysis.
>
> Both papers the reviewer mentioned are interesting, and we intend to incorporate them into the related works section. However, we would like to clarify that both works are completely different and do not explore the exact same problem as us. Specifically, [2] directly uses pre-trained LMs for time series forecasting through prompt engineering, its performance for multi-step forecasting is not close to the state-of-the-art. And its emphasis is the prompt engineering. In addition, the prompt-based forecasting is limited and cannot be applied in many real-world applications. [3] leverages the VIT structure for time series analysis, and does not address the main theme of this work, i.e. cross modality transferring -- directly use frozen LM for time series analysis.

---

> ### Author Response · Authors · 2023-08-19
> **A Supplement Note**
>
> We would like to express our gratitude to reviewer idRZ for taking the time to review our paper. Although it appears that reviewer idRZ is occupied in this period and we were unable to engage in a discussion to address the concerns directly, we would like to provide additional discussion in response to the review questions raised.
>
> **Supplement to Q1: what are results if the time series data does not align with such prior knowledge for many real world applications.**
>
> This question appears to be closely linked to the fundamental inquiry of why our method is effective in this context, as well as the circumstances under which cross-modality knowledge can be leveraged successfully. From the observation, i.e. we can directly use a trained LM for time series forecasting without having to modify it model, makes us belief that the underlying model is doing something very **generic and independent from texts** despite it is trained from text data. Our analysis aims to show that part of this generic function can be related to PCA as minimizing the gradient with respect to the self attention layer seems to do something similar to PCA. And if that's the case, it can apply to the unknown real application as the reviewer mentioned.
>
> The second direction we explored is the n-gram theory. The high-level idea is **we can sample a less complex distribution from a very complex distribution**. The time series dataset available to us is notably less complex in comparison to the NLP dataset used to train GPT2. Given this, if we assume that GPT2's induction heads and FFN are capable of modeling the intricate distribution found within the text data, we can always select a subset of heads and FFN that aligns with the less complex nature of our time series analysis. Hence, if we can reasonably assume that the distribution complexity in most time series datasets is lower compared to text distribution, achieving such transfer becomes feasible.
>
>
> **Supplement to Q3: About It is not clear why authors use GPT-2 (6)**
>
> GPT-2 is merely an example; in fact, many LLMs are also applicable to time series analysis. Given the significant effort and resources required to conduct a wide array of experiments, we primarily relied on GPT-2 for our core investigations. Nevertheless, we have also utilized the CV pre-trained model BEiT, and the NLP pre-trained model BERT. Both were trained on 5% of ETTh2 and 5% of ETTm2, underscoring that the capability for knowledge transfer isn't exclusive to GPT-2. Detailed results related to these models are available in Appendix H.5. As we move forward, our research will delve into the performance implications of transferring pre-trained models from diverse modalities to time series analysis.

---

### Official Review · Reviewer_kdgy · 2023-07-07

**Soundness:** 3 good
**Presentation:** 3 good
**Contribution:** 3 good
**Rating:** 6
**Confidence:** 3

**Summary:**

The authors discuss a method by which a deep transformer model pre-trained on an NLP or CV task can be adapted to a wide variety of applications involving classification or prediction of scalar time series data. This includes an embedding scheme to represent the time series input in the same embedding space as expected by the transformer layers and a fine-tuning approach whereby the multi-head attention and feedforward components of the transformer blocks are frozen, but the embedding and layer norm components are trained. Using the GPT-2 backbone, they apply this method to classification, anomaly detection, imputation, and various forecasting tasks, showing generally good performance compared to many competitors. They

**Strengths:**

The problem of finding a broadly performant time series model is interesting and challenging, particularly in light of the diverse architectures proposed and mixed results obtained by deep neural networks for time series applications. The idea of simply using a pre-trained language model with minimal adaptation is surprising and worth discussion in the scientific community. The empirical analysis attempts to cover a very broad range of tasks and datasets, which is an appropriately high bar for a paper that proposes a “one fits all” approach to time series analysis.

**Weaknesses:**

* The time series applications I know best here are anomaly detection and long-range forecasting, and for both of these settings there are some issues with the reported results.
     - For anomaly detection, there is significant information missing. How are binary anomaly decisions made from the reconstruction error? How are precision and recall computed when anomalies are given by windows rather than points (e.g. SMAP, MSL)? These decisions can have a strong influence on the reported results.
     - While comparisons are extensive for anomaly detection, they do not seem to actually capture the state of the art. For example, the OmniAnomaly method [1], which also naturally generalizes to multivariate time series, significantly outperforms the GPT2(6) model in F1 score on all datasets for which both are evaluated (SMD, MSL, SMAP).
    - The long-range forecasting results are likewise missing comparisons to methods that significantly outperform all reported scores. For example, the S4 model [2] reports lower MSE and MAE on Weather, ETTh1, ETTh2, and ETTm1 (see Table 13, [2]).
    - Moreover, the results reported for Informer in [2] are significantly better than what is reported in Table 13 of the present paper, beating GPT2(6) on ETTh1 and ETTh2. It is unclear what the source of this discrepancy is, but it is large enough to be relevant when drawing conclusions across methods.

* The idea that a lightly-adapted NLP or CV transformer backbone is a highly competitive, highly general time series model is mildly shocking, at least to a time series practitioner. This paper largely misses the opportunity to explore *why* this might be the case, and *what* are the main drivers of this result. For example, do the pre-trained weights actually matter, or is it the architecture? Is fine-tuning required, and if so, why is fine-tuning only the embeddings and layer norm components sufficient? As it stands, the paper communicates an interesting empirical result without providing much insight as to its explanation.

* The specific adaptations of the FPT approach - i.e. the time series encoder and fine-tuning protocol - are both inadequately described. The encoder operations should be described in mathematical detail. The algorithmic details of fine-tuning should at least be provided in the supplement.

* The PCA analysis seems largely unrelated to the preceding work in the paper. It seems to be a general observation about self-attention, without any particular connection to the specific time series applications considered. Moreover, PCA itself is certainly not a one-size-fits-all solution to time series modeling, so it cannot be the basis for a convincing explanation of the model’s success.

[1] Su, Y., Zhao, Y., Niu, C., Liu, R., Sun, W., & Pei, D. (2019). Robust anomaly detection for multivariate time series through stochastic recurrent neural network. KDD.

[2] Gu, A., Goel, K., & Re, C. (2021). Efficiently modeling long sequences with structured state spaces. ICLR.

**Questions:**

Could the authors comment on the discrepancy between Informer performance reported in their Table 13 vs. in [2]? What could explain the substantial difference in metrics on the same task? I certainly don’t mean to claim that [2] must be right, but the size of the difference is enough to affect the interpretation of the results, so it merits some discussion. [Addressed in discussion period]

Could the authors share their perspective on *why* their FPT approach seems to work so well, and what are the main drivers of this result?

What unique insights for time series representation are provided by the PCA analysis in Section 7?

Why not report the results in Figure 1 as a table? They would be much easier to read and compare.

**Limitations:**

There is no direct potential for negative social impact. Potential limitations for the method and results are either discussed or covered above.

---

> ### Author Rebuttal · Authors · 2023-08-09
>
> We sincerely thank Reviewer kdgy for the positive evaluation of our work's intriguing and challenging contributions. We're deeply heartened by the sentiment that "The idea of simply using a pre-trained language model with minimal adaptation is surprising and worth discussion in the scientific community." Such feedback not only affirms our efforts but also motivates us to further our research in this domain. Thanks for your detailed and insightful comments to help us improve our work. We hope your concerns will be addressed.
>
> ***Q1:Anomaly detection.binary anomaly decisions, precision and recall computed method***
>
> To make a fair comparison, we mainly follow the binary anomaly decision methods from TimesNet[2] and using the same window size as 100 throughout the main text. Note that TimesNet[2] have performed an extensive comparison of different SOTA anomaly detection methods in the literature in this setting, which makes our empiricial studies comparable to a wide range of literatures.
> More specifically, we focus on unsupervised time series anomaly detection. Experimentally, each dataset
> includes training, validation and testing subsets. Anomalies are only labeled in the testing subset.
> We select the hyper-parameters following the Gap Statistic method (Tibshirani et al., 2001) in K-Means as described below:
>
> • After the training phase, we apply the model to the validation subset (without label) and obtain the anomaly scores(reconstruction loss) of all time points.
>
> • We count the frequency of the anomaly scores in the validation subset. It is observed that the distribution of anomaly scores is separated into two clusters. We find that the cluster with a larger anomaly score contains r time points. And for our model, r is closed to 0.1\%, 0.5\%, 1\% for SWaT, SMD and other datasets, respectively .
>
> Note that, directly setting the $\delta$ is also feasible. we can fix the $\delta$ as 0.1 for the SMD, MSL and SWaT datasets, 0.01 for the SMAP and PSM datasets, which yield a quite close performance to setting r.
>
> We understand that more complicated strategies on those issues can further improve the detection accuracy. We did not explore those choices as it departs from the main theme of this work, i.e. pointing out the possibility of leveraging frozen LM for various time series analysis tasks.
>
> ***Q2:Anomaly detection. The state of the art problem. the OmniAnomaly method [1], which also naturally generalizes to multivariate time series, significantly outperforms the GPT2(6) model in F1 score on all datasets for which both are evaluated (SMD, MSL, SMAP).***
>
> Our approach is based on Anomaly Transformer [4] by substituting the joint criterion with the reconstruction error.
>
> In the upcoming table, it is evident that integrating an additional association discrepancy via a learnable Gaussian kernel significantly bolsters the Anomaly Transformer's performance. Yet, this introduces a quandary, as the Gaussian kernel doesn't impact the primary forecasting output or the reconstruction loss. To genuinely assess the inherent capabilities of each backbone algorithm—primarily forecasting methods suited for broad time series analysis—it appears more reasonable to focus exclusively on the reconstruction error. Gven the results reported in previous study [4], which indicated that the Anomaly Transformer outperforms OmniAnomaly, we chose to exclude OmniAnomaly from our comparison. Furthermore, as per Table 1 in reference [4], OmniAnomaly exhibits a comparatively weaker performance than GPT2(6), primarily on SMD, SWaT, and PSM datasets.
>
>  | Methods | GPT2(6) | TimesNet | Anomaly.* | Anomaly | OmniAnomaly |
>  | --------| -------- | --------- | --------- | --------- |--------- |
>  | SMD  | 86.89    | 84.61     | 85.49     | 92.33     | 85.22     |
>  | MSL  | 82.45    | 81.84     | 83.31     | 93.59     | 87.67      |
>  | SMAP | 72.88    | 69.39    | 71.18     | 96.69     | 86.92     |
>  | SWaT | 94.23   | 93.02     | 83.10    | 94.07     | 82.83     |
>  | PSM  | 97.13    | 97.34     | 79.40     | 97.89     | 80.83     |
>  | Average | 86.72    | 85.24     | 80.50     | 94.91     | 84.69     |
> * We replace the joint criterion in Anomaly Transformer (2021) with reconstruction error for a fair comparison.
>
> ***Q3: Long-term forecasting. S4 comparison and discrepancy***
>
> First, the discrepancy in the reported results for Informer between [2] and our work is due to the fact that results in [2] are on univariate forecasting, whereas our study focused on multivariate forecasting. It is well recognized in previous studies [3, 5] that multi-variate forecasting is considerably more challenging than univariate forecasting as it aims to make predictions for multiple channels by a single model. Thus, most recent studies of time series analysis, e.g. [3, 5], shift to multi-variate forecasting. Without exception, our study follows the same trend, and in fact, most of the results for baseline methods come from [3]. This also explains the concern from the reviewer that in [2] S4 reports lower MSE and MAE for several datasets -- again, these numbers are based on univariate forecasting, not mult-variate forecasting. Our empirical studies did reveal significantly better results for univariate forecasting, which will be included in the appendix for the final version. The performance of S4 in univariate forecasting is markedly inferior to that of FedFormer, AutoFormer, and FiLM, as detailed in Table 10 of the FiLM appendix.
>
> ***Q4,5,6:In the global response***
>
> [3] Wu, H., Hu, T., Liu, Y., Zhou, H., Wang, J., and Long, M., TimesNet: Temporal 2D-Variation Modeling for General Time Series Analysis, ICLR, 2023.
>
> [4] Xu, J., Wu, H., Wang, J., and Long, M, Anomaly transformer: Time series anomaly detection with association discrepancy, ICLR, 2022.
>
> [5] Zhang, T., Zhang, Y., Cao, W., Bian, J., Yi, X., Zheng, S., and Li, J., Less is more: Fast multivariate time series forecasting with light sampling-oriented mlp structures

---

> > ### Comment · Area_Chair_wAFh · 2023-08-13
> > **Does the author's response address your questions ?**
> >
> > Dear reviewer kdgy,
> > One of the main issues you raised was about the quality of reported results. Does the authors rebuttal address these concerns ?
> > Thanks.

---

> > > ### Author Response · Authors · 2023-08-16
> > >
> > > Dear Chair wAFh
> > >
> > > It appears that reviewer kdgy may be occupied and unable to respond at the moment. To enhance transparency in addressing the Q3 discrepancy or quality of reported results issue, we have incorporated two concise tables summarizing the multivariate and univariate forecasting outcomes with baseline algorithms. These tables consist of average MSE values for four forecasting horizons (96, 192, 336, 720). It is evident from the tables that the univariate MSE is considerably smaller in most datasets, which explains the discrepancy.  this difference arises solely from the distinct experiment settings employed in those two works.
> > >
> > > Furthermore, it is worth noting that even in the univariate forecasting setting, S4[2] does not achieve state-of-the-art performance. Many of the baseline methods we compared in our work outperform S4[2]. Additionally, multi-variate forecasting is considerably more challenging than univariate forecasting as it aims to make predictions for multiple channels by a single model. Thus recent baseline works [3, 4, 5, 6] primarily focus on comparing multivariate forecasting results without providing a comprehensive univariate forecasting table. We have followed this trend in our work.
> > >
> > > **Multivariate**
> > > | Dataset | GPT2(6) | Dlinear | Fedformer | Autoformer |
> > >  | --------| -------- | -------- | ----------- | ----------- |
> > >  | ETTm2  |**0.266**| 0.267     | 0.305    |0.327    |
> > >  | Electricity  | **0.167**| 0.166     | 0.214    |0.227    |
> > >  | Traffic  | **0.414**| 0.434     | 0.610    |0.628    |
> > >  | Weather  | **0.237**| 0.249     | 0.309    |0.338    |
> > > | ILI  | **1.925**| 2.169     | 2.847    |3.006    |
> > >
> > > **Univariate**
> > > | Dataset |Dlinear | Fedformer | Autoformer | S4 |
> > >  | --------| -------- | -------- | ----------- | ----------- |
> > >  | ETTm2  | **0.112**| 0.118     | 0.130    |0.256    |
> > >  | Electricity  | --| **0.326**    | 0.414    |0.401    |
> > >  | Traffic  | --| **0.177**     | 0.261    |0.202    |
> > >  | Weather  | --| **0.007**     | 0.008    |0.006    |
> > > | ILI  | --| **0.694**     | 0.812    |0.808    |
> > >
> > > [2]Gu, A., Goel, K., & Re, C. (2021). Efficiently modeling long sequences with structured state spaces. ICLR.
> > >
> > > [3] Wu, H., Hu, T., Liu, Y., Zhou, H., Wang, J., and Long, M., TimesNet: Temporal 2D-Variation Modeling for General Time Series Analysis, ICLR, 2023.
> > >
> > > [4]Zhou, T., Ma, Z., Wen, Q., Wang, X., Sun, L., & Jin, R. (2022). FEDformer: Frequency Enhanced Decomposed Transformer for Long-term Series Forecasting. ICML
> > >
> > > [5]Zeng, A., Chen, M., Zhang, L., & Xu, Q. (2022). Are Transformers Effective for Time Series Forecasting? AAAI Conference on Artificial Intelligence.
> > >
> > > [6]Nie, Y., Nguyen, N.H., Sinthong, P., & Kalagnanam, J. (2022). A Time Series is Worth 64 Words: Long-term Forecasting with Transformers. ICLR.

---

> > > > ### Comment · Reviewer_kdgy · 2023-08-18
> > > > **Response to author rebuttal**
> > > >
> > > > Thanks to the authors for their engagement and extensive discussion, and apologies for my delay in continuing the conversation.
> > > >
> > > > _Regarding the anomaly detection method and results:_
> > > >
> > > > Thanks for including these details on methodology for the binary anomaly calls. It would be useful for them to be included in the supplement. For the results, the table included copies numbers from Anomaly Transformer paper, which are not themselves consistent (but consistently worse than) those reported in the Omni Anomaly paper itself. Those were the numbers I used for my original point that Omni Anomaly outperforms GPT2(6) on SMD, SMAP, and MSL. Some variation may be expected here due to stochasticity or non-convexity, but it is interesting to note that this is enough to flip the direction of certain conclusions regarding model comparisons. Regardless, with the addition of SWaT and PSM results I acknowledge that GPT2(6) shows the best performance in terms of average F1 score.
> > > >
> > > > _Regarding the forecasting results:_
> > > >
> > > > The authors are correct that I have made a mistaken comparison here, specifically between the univariate setting and results of S4 versus the multivariate setting they consider on the same data. I am grateful for their clarification.
> > > >
> > > > _Regarding the PCA analysis:_
> > > >
> > > > I remain somewhat skeptical of what we can learn from this result, and how this could function in particular as an explanation for the empirical performance of transformer models on time series tasks, but this is an overall minor reservation in the context of this work and its main contributions.
> > > >
> > > > _Conclusion:_
> > > >
> > > > The authors have addressed my major points of concern in detail, in particular surrounding the experiment details and results. I have also read the relevant discussions with other reviewers. I have increased my score to reflect my improved understanding of this work.

---

> > > > > ### Author Response · Authors · 2023-08-19
> > > > >
> > > > > Thank you for taking the time to provide a detailed and informative review and response. Your input is greatly appreciated. We are absolutely delighted to learn that our clarifications have had a positive impact on the paper. Your review has served as a motivating force for us, pushing us to enhance the overall quality of our work. Once again, we express our sincere gratitude for your dedicated time, effort, and support.

---

### Official Review · Reviewer_GLop · 2023-07-07

**Soundness:** 3 good
**Presentation:** 3 good
**Contribution:** 2 fair
**Rating:** 6
**Confidence:** 2

**Summary:**


The paper presents a unified framework for time-series tasks using pre-trained language models. The author make a united model through a fine-tuning approach that focuses on fine-tuning specific parts of the pre-trained language model, rather than the entire set of parameters. As a result of this adaptation, the proposed fine-tuned model achieves comparable performance to previous methods across various datasets. Experimental results also demonstrate the versatility of the fine-tuned model by successfully transferring knowledge from pre-trained datasets, such as images and text.



**Strengths:**

* The proposed framework surpasses existing methods.
* The paper is easily comprehensible and straightforward.
* The evaluation of various time-series tasks highlights the advantages of the proposed approach in numerous cases.


**Weaknesses:**

* Novelty: The proposed fine-tuning approach appears to be more of an incremental improvement. Fine-tuning pre-trained language models is not entirely new.
* Concerns regarding practicality: Does the architecture based on pre-trained language models require more computational resources compared to other models? It would be helpful to analyze the computation cost for inference in fair comparison settings, such as using the same number of parameters, to demonstrate the effectiveness of the proposed approach.


**Questions:**

* Why did the proposed approach fail to demonstrate good performance in the zero-shot task, unlike in the other tasks?

**Limitations:**

yes

---

> ### Author Rebuttal · Authors · 2023-08-09
>
> We deeply appreciate Reviewer GLop's positive acknowledgment of our methodology's efficacy and numerical performance. We are especially grateful for the detailed and insightful feedback provided. Rest assured, we are dedicated to addressing your concerns and enhancing our work.
>
> ***Q1 for Reviwer GLop. Novelty: The proposed fine-tuning approach appears to be more of an incremental improvement. Fine-tuning pre-trained language models is not entirely new.***
>
> Evidently, fine-tuning pre-trained language models for in-modality transfer learning is not a new concept and has been widely adopted in various domains. However, the key novelty of our work is cross-modality knowledge transfer showing that a pre-trained language model learned from texts can be successfully used for time series analysis with most of its parameters frozen. Compared to texts, time series data is more noisy and diverse, and many time series data is application dependent, which makes the cross-modality transferring learning challenging. In this paper, we support this claim both by empirical studies and the analysis of self-attention. We believe that such a transfer is "mildly shocking", as Reviewer kdgy highlighted, as there has been no previous work demonstrating that such a transfer is possible, let alone achieving state-of-the-art performance in all downstream time-series analysis tasks. Furthermore, current studies are limited to a relatively simple LM (GPT-2), we envision that a more complicated LM, such as LLAMA, can lead to more improvements in time series analysis.
>
> ***Q2 for Reviwer GLop. Concerns regarding practicality: Does the architecture based on pre-trained language models require more computational resources compared to other models? It would be helpful to analyze the computation cost for inference in fair comparison settings, such as using the same number of parameters, to demonstrate the effectiveness of the proposed approach.***
>
> We highly agree that analysis of computational cost is helpful for investigating the practicality of the LLM-based model. The results can be found in the table below. Each baseline model comes in two variants, featuring model hidden dimensions of 32 and 768, which align with GPT-2's specifications. Furthermore, the majority of the baseline models consist of three layers. We assessed the computational cost using a batch from ETTh2 (with a batch size of 128) on a 32G V100 GPU.
>
> The results indicate that GPT-2(3) has substantially enhanced time efficiency and reduced parameter quantity compared to baselines with the same model dimension. This was a surprise since we initially anticipated that this large language model might be slower. However, we surmise that the efficient optimization of huggingface's GPT model implementation primarily accounts for such a significant improvement in time costs. Furthermore, GPT-2(3) and GPT-2(6) demonstrate a mere 6.12\% and 4.60\% proportion of learnable parameters among the overall parameter size, respectively.
>
>  | Model         | Training Params | Training Params Percentage(%) | Training Time for 1 Batch (s) | Inference Time for 1 Batch (s) |
>  | ------------- | --------------- | ----------------------------- | ----------------------------- | ------------------------------ |
>  | FEDformer-32  | 437,319         | 100                           | 0.889                         | 0.17                           |
>  | TimesNet-32   | 1,905,015       | 100                           | 0.747                         | 0.302                          |
>  | PatchTST-32   | 543,232         | 100                           | 0.043                         | 0.022                          |
>  | ------------- | --------------- | ----------------------------- | ----------------------------- | ------------------------------ |
>  | FEDformer-768 | 33,105,415      | 100                           | 0.208                         | 0.056                          |
>  | TimesNet-768  | 42,358,519      | 100                           | 5.723                         | 2.162                          |
>  | PatchTST-768  | 19,677,024      | 100                           | 0.457                         | 0.123                          |
>  | GPT-2(3)-768  | 3,906,912       | 6.12                          | 0.093                         | 0.032                          |
>  | ------------- | --------------- | ----------------------------- | ----------------------------- | ------------------------------ |
>  | GPT-2(6)-768  | 3,916,128       | 4.60                          | 0.104                         | 0.054                          |
>
> ***Q3 for Reviwer GLop. Why did the proposed approach fail to demonstrate good performance in the zero-shot task, unlike in the other tasks?***
>
> For the zero-shot tasks, our goal is to verify the representation power of LLMs for time series analysis, and thus focusing on the comparison with a few recently proposed algorithms, such as DLinear, PatchTST and TimesNet. Our empirical studies show that GPT-2(6) yields similar performance as the above state-of-the-art methods designed for time series analysis. However, none of these methods is specially designed for zero-shot learning. In contrast, N-BEATS, as noted in [1,2], has an unique model design (e.g. backcasting and ensemble learning) that enables domain adaption without having to modify its weights, making it particularly suitable for zero-shot learning. That explains why N-BEATS clearly outperforms other competitors in zero-shot learning.
>
> [1] Wu, H., Hu, T., Liu, Y., Zhou, H., Wang, J., and Long, M., TimesNet: Temporal 2D-Variation Modeling for General Time Series Analysis, ICLR, 2023.
>
> [2] Oreshkin, B. N., Carpov, D., Chapados, N., and Bengio, Y, N-beats: Neural basis expansion analysis for interpretable time series forecasting, arXiv:1905.10437, 2019.

---

> > ### Comment · Reviewer_GLop · 2023-08-16
> >
> > I appreciate the author's additional experiments and clarification of your work. Thus,  I increased the score to 6.

---

> > > ### Author Response · Authors · 2023-08-16
> > >
> > > Your review has been immensely valuable in enhancing the caliber of our work. We appreciate your dedication, assistance, and encouragement. It is great to know that our latest experiments and explanations have had a beneficial effect on the paper.

---

### Official Review · Reviewer_qAFk · 2023-07-07

**Soundness:** 2 fair
**Presentation:** 3 good
**Contribution:** 3 good
**Rating:** 7
**Confidence:** 4

**Summary:**

The paper proposes to use pretrained Transformer for time series analysis.
They freeze the attention and FFN layers but fine-tune the position embeddings and add-norm modules to adapt the model to a given task.
This results demonstrate that this method leads to strong performances in time series analysis tasks.
The authors proved a theorem to explain why the text-based pretrained model could generalize well to time series analysis, by drawing insights from PCA.

**Strengths:**

The paper presents very interesting theoretical and empirical findings. Their extensive experiments show that a pretrained text-based transformer model could be adapted to time series analysis and achieve strong performances. I think the findings are worth presenting to the ML community.

Their theoretical analysis explains this by connecting self-attention mechanism with PCA, for which I have concerns/questions as expressed in Weakness.

The paper is overall clearly written.

**Weaknesses:**

I didn't give a high soundness score because the paper hasn't well justified the choice of tuning add-norm layers and positional embeddings. Line-117 claims that it "is considered a standard practice" but I am afraid it is not true.
Parameter-efficient adaptation has been a hot topic and there have been various kinds of methods, such as adding feed-forward adapters (Houlsby et al. ICML 2019) or attentional adapters (Zhao et al. EMNLP 2022), tuning bias terms (Zaken et al. ACL 2022), LoRA (Hu et al. ICLR 2022), and prefix tuning (Li and Liang ACL 2021, Qin and Eisner NAACL 2021).
There hasn't been a "standard".

I think the paper can be improved with experimental analysis on different kinds of LM adaptation methods, which will complement the current results.

More important, the choice of tuning parameters is related to the theorem in the paper: the PCA insights are drawn for the self-attention mechanism, making me wonder: does the choice of tuning parameters lead to the analysis of the self-attention? Or does the analysis of self-attention lead to the choice of tuning parameters? Or neither? Will results with any other adaptation methods give you different kinds of interpretation?

The mentioned references are:

https://arxiv.org/abs/1902.00751 (already cited)

https://arxiv.org/abs/2211.01979

https://arxiv.org/abs/2106.10199

https://openreview.net/forum?id=nZeVKeeFYf9

https://arxiv.org/abs/2101.00190

https://aclanthology.org/2021.naacl-main.410/

Another line of work that this paper should discuss is the NLP literature that also finds the "per-layer high cosine" phenomenon.

This phenomenon is first discussed for skip-gram embeddings: https://aclanthology.org/D17-1308.pdf

Then it is also found in Transformers: https://arxiv.org/pdf/1909.00512.pdf

There is also argument about why cosine metric may not mean that much: https://aclanthology.org/2022.acl-short.45/

The above is to only name a few and there are other papers that one can find by tracing their citation relations.

**Questions:**

See weakness.

**Limitations:**

See weakness.

---

> ### Author Rebuttal · Authors · 2023-08-09
>
> We sincerely thank Reviewer qAFk for the favorable evaluation of our work's theoretical and empirical contributions. It's particularly encouraging to note your belief that the findings merit presentation to the ML community. Such endorsements validate and inspire our continued dedication to this research. We deeply appreciate your detailed and perceptive feedback, and we are committed to addressing your concerns.
>
> ***Q1 for Reviewer qAFk. the choice of tuning add-norm layers and positional embeddings and following PEFT studies***
>
> We agree with the Reviewer that efforts like tuning normalization layers, position embedding, and parameter-efficient fine tuning can further improve performance.
> In fact, we have conducted experiments with PEFT and observed the improvement. However, we chose not to include these findings in our current work in order to highlight the core contribution of the cross-module knowledge transfer to time series. We focus on demonstrating that pre-trained language models can yield strong performance in general time series analysis with minimal finetuning, with cross-module knowledge transfer as our central theme. To support this, we have conducted extensive experiments with frozen FFN and attention layers, using pre-trained parameters mixed with random values to show that the pre-trained-transformer block is vital. While our initial experiment with PEFT shows encouraging results, it introduces new learnable parameters and layers that do not fully align with our main argument, and thus will be left for the future examination.
>
>  | Weather                | 96 (MSE) | 192 (MSE) | 336 (MSE) | 720 (MSE) |
>  | ---------------------- | -------- | --------- | --------- | --------- |
>  | GPT-2 (6)              | 0.162    | 0.204     | 0.254     | 0.326     |
>  | PatchTST               | 0.149    | 0.194     | 0.245     | 0.314     |
>  | GPT-2 (6) + Adapter    | 0.147    | 0.197     | 0.243     | 0.313     |
>  | GPT-2 (6) + NewAdapter | 0.143    | 0.188     | 0.239     | 0.310     |
>
>
> ***Q2 for Reviewer qAFk. The PCA insights are drawn for the self-attention mechanism, making me wonder: does the choice of tuning parameters lead to the analysis of the self-attention? Or does the analysis of self-attention lead to the choice of tuning parameters? Or neither? Will results with any other adaptation methods give you different kinds of interpretation?***
>
> On the high level, we aim to test with minimal tuning whether NLP pre-trained model parameters can be transferred to time series analysis, which was initially inspired by our empirical studies and further supported our analysis that self-attention can essentially deliver the general purposed function of PCA.
>
> ***Q3 for Reviewer qAFk. Another line of work that this paper should discuss is the NLP literature that also finds the "per-layer high cosine" phenomenon.***
>
> Thank you for the invaluable suggestion! We will incorporate the literatures on "per-layer high cosine" in the revised version.

---

> > ### Comment · Reviewer_qAFk · 2023-08-14
> > **Thank you but new discussion needed.**
> >
> > Thank you for your clarification and new results.
> >
> > However, I am afraid that my main concerns are not resolved yet.
> >
> > First, I was not suggesting tuning more things; instead, I was suggesting you give a deeper discussion on why you chose to tune certain parameters but not the others, and why you chose to use certain PEFT method but not the others.
> > You mentioned "minimal tuning", which seems relevant. But what I expect is a more in-depth discussion, hopefully supported by numbers and statistics.
> >
> > Second, I was concerned if your theoretical analysis will be affected by your choice of PEFT, which doesn't seem to be resolved either.
> >
> > So I would still like to maintain my current rating.
> >
> > But I am open to more discussion.

---

> > > ### Author Response · Authors · 2023-08-14
> > >
> > > We sincerely apologize for any misunderstanding that may have occurred regarding your question. We have conducted several experiments, although they have only been briefly presented in the main draft and SI. Below, we summarized our findings in the table.
> > >
> > > Initially, we did not have any preconceived notions about which parameter needed to be tuned for optimal performance. Nevertheless, we did hope that minimal fine-tuning would suffice since it aligns with our message and also leads to a reduction in training time. However, if other fine-tuning choices yield better results, we would prioritize reporting those as our main settings, as we always prioritize performance above all else.
> > >
> > > These choices were based on an ablation study conducted during our time series forecasting experiment. Among the settings we tested were: no-freeze pretrained weight full fine-tuning, no-pretrained Kaiming initialization full training, freeze Kaiming initialization full training, only fine-tuning the FFN layers, only fine-tuning the attention layers, and the GPT2(6) attention-FFN pretrained frozen. We found that the attention-FFN pretrained frozen setting yielded the best results, leading us to choose it as our optimal tuning setting.
> > >
> > > |                  | FFN-Att pretrain-Freeze | No Freeze-Full-Finetune | No Pretrain-Full-training | No Pretrain + Freeze | Pretrain-Finetune FFN-only | Pretrain-Finetune Attention-only |
> > > | ---------------- | ------- | --------- | ----------- | -------------------- | -------------------- | ------------------------ |
> > > | ETTh2 96(mse)  | **0.376**   | 0.440     | 0.465       | 0.540                | 0.469                | 0.443                    |
> > > | ETTh2 96(mae)  | **0.421**   | 0.449     | 0.457       | 0.497                | 0.463                | 0.446                    |
> > > | ETTh2 192(mse) | **0.418**   | 0.503     | 0.614       | 0.721                | 0.487                | 0.600                    |
> > > | ETTh2 192(mae) | **0.441**   | 0.478     | 0.536       | 0.580                | 0.470                | 0.524                    |
> > >
> > > Based on this observation, we began to explore the reasons behind this success and why the pretrained frozen attention module seems to function universally across domains. This led us to propose a theoretical connection between PCA and the attention model, as we attempt to explain the aforementioned findings.
> > >
> > > Regarding the PEFT, as stated from the primary results, it did improve our performance, but we still kept the original FFN and attention layer fixed the same as this work, adding a new layer for training while keeping the pretrained weight untouched. We believe this still matters in that case and supports our PCA analysis to some degree.
> > >
> > > We hope this revised explanation has addressed your concerns. Please do not hesitate to contact us if we have not answered your question completely. In addition, we would like to express our gratitude for requesting clarification. Your feedback has been immensely valuable in enhancing the overall quality of our work.

---

> > > > ### Comment · Reviewer_qAFk · 2023-08-14
> > > >
> > > > I am sorry that I may not be clear in prev messages.
> > > >
> > > > I was not asking for more experiments.
> > > > Instead, I was asking for more theoretical analysis or intuitions or thoughts.
> > > >
> > > > My main concern is: your theoretical conclusions rely on your technical choices.
> > > > I want to know what may change if you change your choices, and why you made the particular choices---intuitively or theoretically.
> > > >
> > > > I understand your experimental results. But those are not my concern.

---

> > > > > ### Author Response · Authors · 2023-08-14
> > > > >
> > > > > I understand your point. You are absolutely correct that our explanation is dependent on the experiment results. This explanation was inspired by the positive results we obtained from the frozen pretrained weights experiment. However, if we had found that full-finetune yielded better results or that the pretrained weights did not contribute to good results, our story would be completely different. In that case, we would find less interesting information, such as that GPT2 and other LLM's model designs are also suitable for time series analysis. There would be less explanation to provide, as it would be expected that transformer models work across different domains, and the core structure of VIT which works in CV is quite similar to that of transformer models in NLP. In that case, there would be no cross-modality knowledge transfer, only model adaptation.
> > > > >
> > > > > The reason we spent a lot of time explaining this work is that pretrained weights are incredibly helpful. This did indeed surprise us, and we came up with two approaches to explain this phenomenon, including the **PCA approach**. The core high-level idea is that classic algorithms such as PCA, SVD, and FFT exhibit broad applicability without necessitating domain-specific alterations, underscoring their universality. We believe that the pretrained attention module is somehow connected to those classic algorithms, which are the only algorithms we know of that exhibit such universality and cross-domain ability.
> > > > >
> > > > > Another idea we explored was the we believe that LLMs can learn knowledge from the data that can work across domains, similar to the N-grams combination rule. Since we tokenize the data across all domains, we can view all of them as estimating the N-grams generation probability. An N-gram can be constituted by N-1 grams and 1 token, N-2 grams and 2 grams, and so on. Such combinations are shared across any data. We conducted some theoretical analysis in the SI F section:**N-gram Explanation for Universality**. These are the two main explanations we have for cross-domain knowledge transferring.
> > > > >
> > > > > You are absolutely right. If frozen pretrained weights do not work, we may need to provide a completely different explanation for this work, or we may not need an explanation at all since there would be no cross-domain knowledge transfer.
> > > > >
> > > > > Thank you very much for your quick response and question. We hope that our response has addressed your concerns, but please do not hesitate to ask again if you require further clarification. The explanation part is indeed crucial since the phenomenon of cross-modality knowledge transfer is quite surprising.

---

> > > > > ### Author Response · Authors · 2023-08-17
> > > > >
> > > > > Thank you once again for your insightful question. We have dedicated some time and effort to delve deeper into Reviewer qAFk's query, which we find to be both profound and thought-provoking. To provide a detailed response, it is essential to have a comprehensive understanding of each component's function within the LLM model.
> > > > >
> > > > > Regarding the scenarios where attention-only-freeze, FFN-only-freeze, or full fine-tuning yield optimal results, we can offer the following explanations.
> > > > >
> > > > > The rationale behind the success of **full fine-tuning** is relatively straightforward. By starting from pretrained weights in a flat optimized area, a slight adjustment may lead to improved outcomes compared to random initial full training. This improvement can be attributed to the principles of optimization. Furthermore, overfitting is less likely to occur when using pretrained weights, particularly if we impose constraints on the learning process to minimize deviations from the initial weights. However, in such cases, it becomes challenging to clearly argue for knowledge transfer, as the weights have undergone some modifications, making it difficult to quantify the extent to which the original weights contributed to the final results.
> > > > >
> > > > > Now, let's explore the **attention-only-freeze** and **FFN-only-freeze** scenarios. Previous studies [1,2,3] have revealed that pretrained attention modules can function as induction heads, resembling a fuzzy matching mechanism:[a][b].....[a]->[b] given the existence of a historical token pair [a][b], when encountering token [a], the model predicts token [b]. This fuzzy matching paradigm involves multiple induction heads that interact with each other. On the other hand, the FFN layer serves as a memory layer that stores item concepts like tokens [a] and [b]. If we observe that either attention-only-freeze or FFN-only-freeze yields the best performance, it implies that we are leveraging specific induction head paths within the pretrained model.
> > > > >
> > > > > **The underlying idea is that we can sample a less complex distribution from a highly complex distribution.** Given that our time series dataset is significantly less complex compared to the NLP dataset used to train GPT2, and assuming that the complete set of attention modules and FFN can effectively model this complex distribution, we can selectively sample a subset of induction heads and FFN components that are better suited for our time series analysis.
> > > > >
> > > > > We acknowledge that our explanation might seem somewhat handwaving, as our understanding of language models is not yet fully comprehensive, despite their outstanding performance in numerous downstream tasks. Nevertheless, we hope that our thoughts on these three scenarios contribute to addressing your concerns.
> > > > >
> > > > > [1]A Mathematical Framework for Transformer Circuits 2021
> > > > >
> > > > > [2]Geva, M., Schuster, R., Berant, J., & Levy, O. (2021). Transformer Feed-Forward Layers Are Key-Value Memories
> > > > >
> > > > > [3]Dai, D., Dong, L., Hao, Y., Sui, Z., & Wei, F. (2022). Knowledge Neurons in Pretrained Transformers.

---

> > > > > > ### Comment · Reviewer_qAFk · 2023-08-17
> > > > > >
> > > > > > Thank you for your informative responses.
> > > > > >
> > > > > > But I guess my original questions are easier than you thought. Although you provided a lot of new discussion (which is very good though!), I am afraid that they are still off the target.
> > > > > >
> > > > > > Let me try to put it as simple as I can.
> > > > > >
> > > > > > You gave a very insightful PAC argument in the paper. But it seems to be dependent on your particular technical design.
> > > > > > (Note: not your experiment results, but your design.)
> > > > > >
> > > > > > So, what are your thoughts on how your insights can apply to other settings, e.g., I choose to tune something else in LLMs?
> > > > > >
> > > > > > Again, it is specifically about your PAC argument, and it is specifically about theory or thoughts, not experiments.

---

> > > > > > > ### Author Response · Authors · 2023-08-18
> > > > > > >
> > > > > > > We sincerely apologize for the inconvenience of having to explain this question multiple times. After discussing it among ourselves, although we haven't reached a consensus regarding our understanding of the problem, we are concerned about potentially causing any inconvenience by responding over the weekend. Therefore, we have made the decision to attempt an explanation today to see if this response adequately addresses your inquiry.
> > > > > > >
> > > > > > > In our PCA analysis, we make three assumptions. The gradient w.r.t. the parameters in self attention layer (i.e., $W_Q$ and $W_K$) is small, the temperature parameter is large enough so that the exponential function can be well approximated by the linear function and the input patterns are zero centered. Based on those those assmputions, we find that $A = W_QW_K^T$ should be aligned with the top eigen vectors of the input patterns.
> > > > > > >
> > > > > > >
> > > > > > > Therefore, if we tune the parameters in FFN layer, the input pattern will change, and on the the other hand, if we tune the parameters with-in attention, the gradient of self attention layer will also be influenced. Both may have negative impact, which echoes our numerical observations. In our current configuration, we merely use a projection layer to align the time series token with the existing word tokens. Thus as long as original pretrained GPT2 has good structure to conduct PCA task,  we may not necessary to further fine-tune the remaining modules.

---

> > > > > > > > ### Comment · Reviewer_qAFk · 2023-08-18
> > > > > > > >
> > > > > > > > Thanks for your explanation. This is close to my question.
> > > > > > > >
> > > > > > > > So, the takeaway is: this insight depends on not updating attention or FFN. Is that right? This is what I asked about.
> > > > > > > >
> > > > > > > > As for your new experiment results, now I understand why you kept showing me those results for my question. I recommend having them in the paper.
> > > > > > > >
> > > > > > > > But I am still confused with the way you phrase your argument. You said "we may not necessary to further fine-tune the remaining modules". But my questions are not about "necessity", right? They are about "possibility".
> > > > > > > > So your phrasing still makes me a little worried that my questions are not fully understood. I have been trying my best to rephrase it and clarify it, so I am not sure which part is confusing so authors may not understand what I have been asking.
> > > > > > > >
> > > > > > > > Or, maybe we can have the table turned around: what you do you think I have been asking? Instead of answering my question, can you paraphrase my question and ask in your language? Then I can check if we are on the same page.
> > > > > > > >
> > > > > > > > No worries about "weekend" for my review and discussion.

---

> > > > > > > > ### Comment · Reviewer_qAFk · 2023-08-18
> > > > > > > >
> > > > > > > > Oh. Are you saying that: using your projection layer, you can align time series with words, so there is no need to tune anything else for a good performance on time series. Therefore, it is okay to live with your param-tuning design and enjoy the neat connection between attention and PCA (even it relies on some strong assumptions)?

---

> > > > > > > > > ### Author Response · Authors · 2023-08-18
> > > > > > > > >
> > > > > > > > > Yes, you instantly get the main point! Actually, we are trying some aligning and then fine-tuning approach to be the next step after this work. Perhaps we can use the distribution alignment learning for the embedding part completely independent of GPT or any LLM. We aim to simply train an embedding layer that aligns the distribution of time series embeddings with the distribution of text embeddings. If we are right, even fewer finetuning are need and better performance could be achieved. Our ultimate objective is to eliminate the need for model training or fine-tuning, relying instead on a single pre-trained model for seamless inference with new data. While we currently lack a specific design for achieving this, it is a remarkable capability that ChatGPT or LLMs excel at.

---

> > > > > > > > > > ### Comment · Reviewer_qAFk · 2023-08-19
> > > > > > > > > >
> > > > > > > > > > Great. Sorry for my previous misunderstanding.
> > > > > > > > > >
> > > > > > > > > > Then do you have any ideas on improving the presentation or adding anything so readers like me will feel easier to appreciate it?

---

> > > > > > > > > > > ### Author Response · Authors · 2023-08-19
> > > > > > > > > > >
> > > > > > > > > > > Undoubtedly, this discussion has proven to be tremendously helpful. We greatly value your suggestions and intend to incorporate them by specifically including the discussed points in the "A Supplementary Note", and integrating the remaining ideas into the "Theoretical analysis" and "Future Work" sections, respectively. By doing so, we aim to ensure that readers can effortlessly grasp the key concepts presented.

---

> > > > > > > > > > > > ### Comment · Reviewer_qAFk · 2023-08-19
> > > > > > > > > > > >
> > > > > > > > > > > > well, honestly, it is also my bad to not fully understand your previous arguments.
> > > > > > > > > > > >
> > > > > > > > > > > > overall, my main confusion was: how your tuning design choice affects your PCA insights and how it generalizes.
> > > > > > > > > > > >
> > > > > > > > > > > > your main resolution is: you can achieve adaptation without worrying about any design that may change your PCA insights. they are somehow tied.
> > > > > > > > > > > >
> > > > > > > > > > > > This insight should be clear in your future version.
> > > > > > > > > > > >
> > > > > > > > > > > > with trust that you will improve your presentation, I increased my rating to 7.

---

> > > > > > > > > > > > > ### Author Response · Authors · 2023-08-19
> > > > > > > > > > > > >
> > > > > > > > > > > > > We sincerely appreciate your patience and the considerable time you have dedicated to rephrasing the questions and correcting our misunderstanding. It is an honor to engage in such a fruitful discussion with a knowledgeable reviewer like yourself. We express our deepest gratitude for your dedicated time, effort, and support. Rest assured, we will enhance our presentation based on the valuable discussions we have had thus far

---

> > > > > > > ### Author Response · Authors · 2023-08-18
> > > > > > > **A supplementary note**
> > > > > > >
> > > > > > > We would like to express our gratitude to reviewer qAFk for actively engaging in discussions with us. We would like to provide a supplementary note regarding the theoretical explanation of the exceptional cross-domain knowledge transfer phenomenon exhibited in this work by LLM.
> > > > > > >
> > > > > > > First, we want to be transparent with the reviewer that by no means we have a full understanding of why FPT of LM can work so well for time series. As indicated in the conclusion session of our work, we are exploring **n-gram theory** that has the potential to explain the mystery from a different direction.
> > > > > > >
> > > > > > > Second, the observation, i.e. we can directly use a trained LM for time series forecasting without having to modify its model, makes us believe that the underlying model is doing something very **generic and independent from texts** despite it being trained from text data. Our analysis aims to show that part of this generic function can be related to PCA, as minimizing the gradient with respect to the self-attention layer seems to do something similar to PCA. We don't know if PCA is the most important part of this generic function, as we indicate in the conclusion session that there is another potential generic function performed by transformer.
> > > > > > >
> > > > > > > Third, our fine-tuning experiments found that full fine-tuning failed to deliver better performance than only tuning the embedding layer and layer norm components. This is indeed not a surprise in the studies of fine-tuning. For instance, a number of studies have shown that parameter-efficient fine tuning (i.e. fine tuning a small portion of model) yields better performance compared to full fine-tuning, particularly when the number of data points used for fine-tuning is limited. When coming the question of why fine-tuning is restricted to the embedding layer and layer norm, following our hypothesis that the pre-trained LM as a whole performs something generic, **partially fine-tuning any of its components may break the generic function and lead to relatively poor performance** for time series analysis.
> > > > > > >
> > > > > > > Finally, we hope that this article helps encourage and inspire people to carry out a further deep investigation of how and why LLM works and potentially leads to discoveries of new and exciting development.

---

### Author Rebuttal · Authors · 2023-08-09

We thank the Reviewers for the insightful comments and detailed feedback. We were delighted that reviewers find our paper has the following advantages:

**Innovative Findings**: Using pretrained transformers for time series.(qAFk, kdgy, qj2Q)

**Clear Writing:** Easily comprehensible content.(qAFk, GLop, qj2Q)

**Robust Analysis**: Wide-ranging experiments across tasks/datasets.(qAFk, GLop, kdgy, qj2Q)

**Simplicity & Top-tier Results**: Outperforms complex techniques.(GLop, idRZ, qj2Q)

**Attention & PCA Insight**: Explains model's success.(qAFk, qj2Q)


We've addressed certain concerns in individual rebuttals. However, owing to space limitations and the extent of questions from Reviewer kdgy, we've moved some responses to this global reply section.

***Q4 for Reviewer kdgy. The specific adaptations of the FPT approach - i.e. the time series encoder and fine-tuning protocol - are both inadequately described. The encoder operations should be described in mathematical detail. The algorithmic details of fine-tuning should at least be provided in the supplement.***

We use the GPT-2 model [6] as our time series encoder and the detailed mathematical detail and architecture can be found in [6, 7]. Thus, with the page limit, we cannot go into further detail on this in the paper. In the fine-tuning stage, we freeze certain parameters of GPT-2 by modifying the $requires grad$ to false in PyTorch as shown in our provided code. We will provide the pseudocode of GTP-2 method in the appendix for the final version.

***Q5 for Reviewer kdgy. Could the authors comment on the discrepancy between Informer performance reported in their Table 13 vs. in [2]? What could explain the substantial difference in metrics on the same task? I certainly don’t mean to claim that [2] must be right, but the size of the difference is enough to affect the interpretation of the results, so it merits some discussion.***

The tasks referenced in Table 13 and [2] are distinct. [2] exclusively reports results for univariate forecasting, whereas our experiments focus on multivariate long-term forecasting, leading to a noticeable disparity in performance. Our experiment settings and baseline citations are consistent with previous works, such as [3]. We will provide clearer details regarding our experimental procedures in the revised version.

***Q6 for Reviewer kdgy. This paper largely misses the opportunity to explore why this might be the case, and what are the main drivers of this result. For example, do the pre-trained weights actually matter, or is it the architecture? Is fine-tuning required, and if so, why is fine-tuning only the embeddings and layer norm components sufficient? As it stands, the paper communicates an interesting empirical result without providing much insight as to its explanation. Could the authors share their perspective on why their FPT approach seems to work so well, and what are the main drivers of this result? What unique insights for time series representation are provided by the PCA analysis in Section 7?***

We have run extensive experiments for various finetuning and we did not include them in the paper due to the space limitation. Our empirical studies show that the weights learned by LM are essential to the success of cross modality transferring. In fact, full fine tuning of all the weights leads to significant degradation in performance -- that is the reason why we freeze all the weights for FFN and attention layers and only tune embedding and layer norm. To explain why FPT works so well for time series analysis, we try to understand the function of self-attention layers and find that it may be closely related to PCA, a general-purpose function independent from any domain. This domain-independent property delivered by attention layers makes us believe in the possibility of using a frozen LM for time series.

***Q7 for Reviewer kdgy. Why not report the results in Figure 1 as a table? They would be much easier to read and compare.***

Thank you for your suggestion. Due to page constraints, we consolidated the classification task results into a figure, which admittedly affected readability. We plan to reorganize the layout and include a table displaying the results either in the main body or the appendix.

Moreover, our approach mirrored that of the baseline. Since the TimesNET paper presents classification outcomes in a comparable format, we felt it prudent to maintain this consistency to facilitate easier comparison.

 |          | XGBoost | Rocket | LSTNet | LSSL | TCN  | DLinear | LightTS | TimesNet |
 | -------- | ------- | ------ | ------ | ---- | ---- | ------- | ------- | -------- |
 | Accuracy | 66.0    | 72.5   | 71.8   | 70.9 | 70.3 | 67.5    | 70.4    | 73.6     |

 |          | Transformer | Reformer | Informer | Pyraf. | Autof. | Stationf. | FEDf. | ETSf. | Flowf. | GPT2(6) |
 | -------- | ----------- | -------- | -------- | ---------- | ---------- | ------------- | --------- | --------- | ---------- | ------- |
 | Accuracy | 71.9        | 71.5     | 72.1     | 70.8       | 71.1       | 72.7          | 70.7      | 71.0      | 73.0       | 74.0    |


[2] Gu, A., Goel, K., & Re, C. (2021). Efficiently modeling long sequences with structured state spaces. ICLR.

[3] Wu, H., Hu, T., Liu, Y., Zhou, H., Wang, J., and Long, M., TimesNet: Temporal 2D-Variation Modeling for General Time Series Analysis, ICLR, 2023.

[6] Radford, A., Wu, J., Child, R., Luan, D., Amodei, D., and Sutskever, I, Language models are unsupervised multitask learners, 2019.

[7] Vaswani, A., Shazeer, N., Parmar, N., Uszkoreit, J., Jones, L., Gomez, A. N., Kaiser Lukasz, and Polosukhin, I, Attention is all you need, arXiv:1706.03762, 2017

---

### Decision · Program_Chairs · 2023-09-21

**Decision:**

Accept (spotlight)

**Comment:**

Authors show that a pre-trained language model can be a good starting point for time series modeling, which seems strange when you think about it. They show results on several different datasets and achieve compelling results on them. They provide some understanding of why pertaining would help, through the relationship of self-attention to PCA. The reviews were generally quite positive. Several concerns raised by the reviewers seem to have been addressed by the authors, which resulted in two of them unweighting the original score.

Perhaps a couple of things that could have been done better were:
1. More in depth of exploration why pertaining with LM's would help on sequence modeling.
2. Looking into other ways of fine-tuning the pretrained models.

Nevertheless, the paper makes an interesting observational contribution, and nice results, as it stands. I think the community will find the results interesting.